# GROD: Enhancing Generalization of Transformer with Out-of-Distribution Detection

## Abstract

Transformer networks excel in natural language processing (NLP) and computer vision (CV) tasks. However, they face challenges in generalizing to Out-of-Distribution (OOD) datasets, that is, data whose distribution differs from that seen during training. The OOD detection aims to distinguish data that deviates from the expected distribution, while maintaining optimal performance on in-distribution (ID) data. This paper introduces a novel approach based on OOD detection, termed the *Generate Rounded OOD Data* (GROD) algorithm, which significantly bolsters the generalization performance of transformer networks across various tasks. GROD is motivated by our new OOD detection Probably Approximately Correct (PAC) Theory for transformer. The transformer has learnability in terms of OOD detection that is, when the data is sufficient the outlier can be well represented. By penalizing the misclassification of OOD data within the loss function and generating synthetic outliers, GROD guarantees learnability and refines the decision boundaries between inlier and outlier. This strategy demonstrates robust adaptability and general applicability across different data types. Evaluated across diverse OOD detection tasks in NLP and CV, GROD achieves SOTA regardless of data format. The code is available at https://anonymous.4open.science/r/GROD-OOD-Detection-with-transformers-B70F.

## 1 Introduction

Mainstream machine learning algorithms typically assume data independence, called in-distribution (ID) data (Krizhevsky et al., 2012; He et al., 2015). However, in practical applications, data often follows the "open world" assumption (Drummond & Shearer, 2006), where outliers with different distributions can occur during inference. This real-world challenge frequently degrades the performance of AI models in prediction tasks. One remedy is to incorporate OOD detection techniques. This paper proposes a new algorithm based on OOD detection for transformer networks, which can significantly improve their performance in predicting outlier instances.

The transformer is a deep neural network architecture that leverages an attention mechanism. It is renowned for its powerful capabilities in a variety of deep learning models, such as large language models, computer vision models, and graph neural networks. OOD detection aims to identify and manage semantically distinct outliers, referred to as *OOD data*. It requires the designed algorithm to detect OOD instances and avoid making predictions on them, while maintaining robust performance on ID data. By employing OOD detection, we develop a new algorithm, which we call **G**enerate **R**ounded **O**OD **D**ata (GROD), for fine-tuning a transformer network to enhance its ability to predict the unknown distribution. By taking account of the OOD Detection in network training, we can strengthen the recognition of the in-distribution and out-distribution boundary.

We establish the OOD Detection PAC Learning Theorem (Theorem 4). It demonstrates that penalizing the misclassification of OOD data in the training loss of the transformer clarifies the decision boundary between inliers and outliers. This condition ensures that the model possesses *OOD Detection Learnability*. Moreover, we quantify the learnability by proving an error boundary regarding the transformer model's budget (the number of total trainable parameters) (Theorem 5). We define GROD following these two theorems. When the network depth is substantial, the GROD-enhanced transformer converges to the target mapping with robust generalization capabilities.

Our main contributions are summarized as follows:

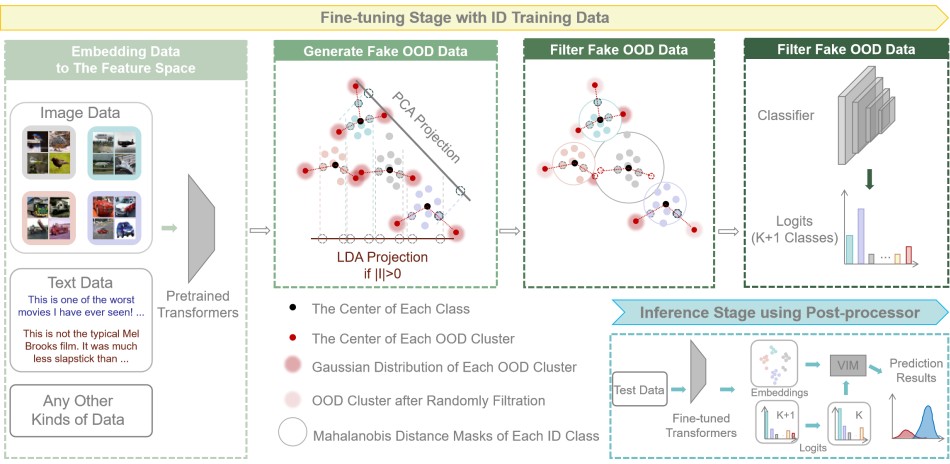

Figure 1: Overview of GROD algorithm: In the fine-tuning stage, GROD generates fake OOD data as part of the training data. GROD then guides the training by incorporating the ID-OOD classifier in the loss. In the inference stage, the features and adjusted LOGITS are input into the post-processor.

- We establish a PAC learning framework for OOD detection applied to transformers, providing necessary and sufficient conditions as well as error boundary estimates for learnability. This contribution not only bridges a theoretical gap but also supports practical decisions regarding parameter selection and model design in terms of learnability and generalizability.

- Inspired by our theoretical framework and empirical validation, we propose a novel OOD detection approach, *Generate Rounded OOD Data* (GROD). This strategy is theoretically grounded and high-quality in generating and representing features regardless of data types.

- We conduct comprehensive experiments to explore the existing limitations and the interpretability of GROD, and display the state-of-the-art (SOTA) performance of GROD on image and text datasets together with ablation studies and visualizations.

## 2 RELATED WORKS

**Methods and theory of OOD detection.** Out-of-distribution (OOD) detection has seen significant advancements in both methods and theory. Recent approaches typically combine post-processing techniques and training strategies to improve model performance. Key post-processing techniques include distance functions (Denouden et al., 2018), scoring functions (Ming et al., 2022a), and the integration of disturbance terms (Hsu et al., 2020). On the training side, strategies such as loss functions for compact representations (Tao et al., 2023) and reconstruction models for anomaly detection (Graham et al., 2023; Jiang et al., 2023) have been proposed. The transformer architecture, known for its robust feature representation capabilities, has gained popularity in OOD detection (Koner et al., 2021; Fort et al., 2021). Additionally, leveraging auxiliary outliers has emerged as a prominent strategy, with methods like Outlier Exposure (OE) using external datasets to train models for distinguishing ID from OOD samples (Hendrycks et al., 2018; Zhu et al., 2023), and generative-based methods creating synthetic OOD data through methods like VOS (Du et al., 2022) and OpenGAN (Kong & Ramanan, 2021). These generative methods help to overcome the reliance on predefined outlier datasets (Wang et al., 2023; Zheng et al., 2023). Theoretical work in OOD detection has also grown, with studies on maximum likelihood estimation (Morteza & Li, 2022), density estimation errors (Zhang et al., 2021), and PAC learning theory (Fang et al., 2022). However, a comprehensive theory for OOD detection with transformers is yet to be established (Yang et al., 2021), hindering the development of reliable OOD detection algorithms. In Appendix A, we provide a more detailed discussion of the advancements in more related fields.

**Notation.** We introduce some notations regarding OOD detection tasks. We employ subscripts following the standard notation to represent the elements of a vector or matrix. Formally, $\mathcal{X}$ and $\mathcal{Y} := \{1, 2, \cdots, K, K+1\}$ denote the whole dataset and its label space. As subsets in

$\mathcal{X}$, $\mathcal{X}_{\text{train}}$, $\mathcal{X}_{\text{test}}$ and $\mathcal{X}_I$ represents the training dataset, test dataset and ID dataset, respectively. $\mathcal{Y}_I := \{1, \cdots, K\}$ denote the ID label space. $l(\mathbf{y}_1, \mathbf{y}_2), \mathbf{y}_1, \mathbf{y}_2 \in \mathcal{Y}$ denotes the paired loss of the prediction and label of one data, and $\mathcal{L}$ denotes the total loss. We depict the basic structure of a transformer network as follows, which includes the following components: input embedding, positional encoding, an encoder, a decoder and an output layer. For OOD detection tasks, which predominantly encompass classification objectives, we directly connect an output layer subsequent to the encoder to streamline the process. For clarity and operational simplicity, we assume that the input data $\mathcal{X}$ is processed by the input embedding and positional encoding mechanisms. The encoder is an assembly of multiple attention blocks, each comprising a self-attention layer and a Feed-forward Fully Connected Network (FFCN). The self-attention layer calculates matrices of key, query, and value, to express the self-attention mechanism, where the hidden dimensions for keys and queries are $m_h$, and for values are $m_V$. Each individual data is transformed into $\tau$ tokens, with each token having a dimension $\hat{d}$. To quantify the computational overhead of a transformer block, we define the budget $m := (\hat{d}, h, m_h, m_V, r)$, representing the parameter size of one block. More details about notations and preliminaries for theoretical analysis are illustrated in Appendix B.

## 3 GROD ALGORITHM

**Framework overview.** As illustrated in Figure 1, GROD contains several pivotal steps. Firstly, a binary ID-OOD classification loss function is added to fine-tune the transformer. This adjustment aligns more closely with the transformer's learnable conditions in the proposed theory. To effectively leverage this binary classification loss, we introduce a novel strategy for synthesizing high-quality OOD data for training. To minimize computational overhead while leveraging high-quality embeddings for enhanced efficiency, GROD generates virtual OOD embeddings, rather than utilizing original data. As defined in Definition 4, GROD gains theoretical guarantee on transformers with multiple transformer layers and a classifier for OOD detection and classification tasks. So GROD has compatibility for transformers extract features like CLS tokens and inputs into the classifier, which is appliable to almost all transformer models. Notably, we focus on OOD detection without training with outlier datasets, which is another dev different from OE and also has its real application scenarios (Yang et al., 2024). Principal Component Analysis (PCA) and Linear Discriminant Analysis (LDA) projections are employed to generate global and inter-class outliers respectively, utilizing overall ID information and distinct features for each ID data category. Next, a filtering mechanism is applied to remove synthetic ID-like outliers and maintain a reasonable ratio of ID and OOD. This refined dataset together with the binary loss then serves to fine-tune the transformer under GROD framework. During the testing phase, embeddings and prediction LOGITS are extracted from the GROD-enhanced transformer. These outputs are reformulated for post-processing. The post-processor, VIM (Wang et al., 2022), is applied to get the final prediction.

**Recognize boundary ID features by PCA and LDA projections.** Let $\mathcal{X}_{\text{train}}$ denote the input to the transformer backbone, which is transformed into a feature representation $\mathcal{F} \in \mathbb{R}^{n \times s}$ in the feature space:

$$\mathcal{F} = \text{Feat} \circ \text{Block}^n(\mathcal{X}_{\text{train}}), \tag{1}$$

where $\text{Feat}(\cdot)$ is the process to obtain features. For instance, in ViT models, $\text{Feat}(\cdot)$ represents extracting CLS tokens. Subsequently, we generate synthetic OOD vectors using PCA for global outliers and LDA for inter-class distinctions. LDA is selected for its ID-separating ability, with techniques to guarantee the robustness of generated OOD, where $B$ is the batch size. Specifically, we first find data with maximum and minimum values of each dimension in projection spaces. $\mathcal{F}$ is projected by

$$\mathcal{F}_{\text{PCA}} = \text{PCA}(\mathcal{F}), \ \mathcal{F}_{\text{LDA},i} = \text{LDA}(\mathcal{F}, \mathcal{Y})|_{\mathbf{y}=i}, \ i \in \mathcal{Y}_I. \tag{2}$$

Features are mapped from $\mathbb{R}^d$ to $\mathbb{R}^{num}$, $num \leq d$. Then target vectors are acquired, denoted as $v_{\text{PCA},j}^M = \arg\max_{v \in \mathcal{F}_{\text{PCA}}} v_j$, $v_{\text{LDA},i,j}^M = \arg\max_{v \in \mathcal{F}_{\text{LDA},i}} v_j$ for maximum and $v_{\text{PCA},j}^m$, $v_{\text{LDA},i,j}^m$ for minimum, $i \in \mathcal{Y}_I$, $j = 1, \cdots s$. The sets $\hat{V}_{\text{PCA}} := \{v_{\text{PCA},j}^M \text{ and } v_{\text{PCA},j}^m, j = 1, \cdots s\}$ and $\hat{V}_{\text{LDA},i} := \{v_{\text{LDA},i,j}^M \text{ and } v_{\text{LDA},i,j}^m, j = 1, \cdots s\}, i \in \mathcal{Y}_I$ are the boundary points in the projection spaces, which are mapped back to the original feature space:

$$V_{\text{PCA}} = \text{PCA}^{-1}(\hat{V}_{\text{PCA}}), \ V_{\text{LDA},i} = \text{LDA}^{-1}(\hat{V}_{\text{LDA},i}), \ i \in \mathcal{Y}_I, \tag{3}$$

where $\text{PCA}^{-1}$ and $\text{LDA}^{-1}$ are inverse mappings of PCA and LDA according to set theory.

**Generate fake OOD data.** Boundary points, while initially within ID, are extended into OOD regions. Firstly, the centers of every training batch and category are calculated by $\mu_{\text{PCA}} = \frac{\sum_{v \in \mathcal{F}} v}{|\mathcal{F}|}$ and $\mu_{\text{LDA},i_k} = \frac{\sum_{v \in \mathcal{F}} v|_{\mathbf{y}=i_k}}{B_{i_k}}$, where $i_k \in \{i = 1, \cdots, K : |\mathcal{F}|_{\mathbf{y}=i} > 1\} := \hat{I}$. To save computation costs and control the ratio of ID and OOD, we derive a subset from $\hat{I}$ to generate fake OOD, and denote it as $I$ for simplicity:

$$\kappa = \min\{|\hat{I}|, \lceil \frac{2B}{K \cdot num} \rceil\}, \tag{4}$$

$$I := \{i \in \hat{I} : |\mathcal{F}|_{\mathbf{y}=i} \text{ is the top-}\kappa \text{ maximum for all } i\}, \tag{5}$$

where $num$ is a hyperparameter empirically set to be 1. When $n = 0$, only PCA is used. Then we generate Gaussian mixture fake OOD data with expectations $U_{\text{OOD}}$:

$$U_{\text{OOD}} = \left\{ v + \alpha \frac{v - \mu}{||v - \mu||_2 + \epsilon} : v \in V_{\text{PCA}}, \mu = \mu_{\text{PCA}} \ or \ v \in V_{\text{LDA},i_k}, \mu = \mu_{\text{LDA},i_k}, i_k \in I \right\}, \tag{6}$$

where $\epsilon = 10^{-7}$, $\alpha$ is a hyperparameter representing extension proportion of $L_2$ norm. Gaussian mixture fake OOD data are generated with distribution

$$D_{\text{OOD}} = \frac{1}{|U_{\text{OOD}}|} \sum_{\mu_{\text{OOD}} \in U_{\text{OOD}}} \mathcal{N}(\mu_{\text{OOD}}, \alpha/3 \cdot I_{\text{OOD}}), \tag{7}$$

where $I_{\text{OOD}}$ is the identity matrix. We denote the set of these fake OOD data as $\hat{\mathcal{F}}_{\text{OOD}} := \hat{\mathcal{F}}_{\text{PCA}}^{\text{OOD}} \cup (\cup_{i_k \in I} \hat{\mathcal{F}}_{\text{LDA},i_k}^{\text{OOD}})$, where $\hat{\mathcal{F}}_{\text{PCA}}^{\text{OOD}}$ and $\hat{\mathcal{F}}_{\text{LDA},i_k}^{\text{OOD}}$ are clusters consist of $num$ data points each, in the Gaussian distribution with expectations $\mu_{\text{PCA}}$ and $\mu_{\text{LDA},i_k}$ respectively.

**Filter OOD data.** To eliminate ID-like synthetic OOD data, we utilize the Mahalanobis distance, improving the generation quality of outliers. Specifically, Mahalanobis distance from a sample $\mathbf{x}$ to the distribution of mean $\mu$ and covariance $\Sigma$ is defined as $\text{Dist}(\mathbf{x}, \mu, \Sigma) = (\mathbf{x} - \mu)\Sigma^{-1}(\mathbf{x} - \mu)^T$. To ensure robust computations, the inverse matrix of $\Sigma$ is calculated with numerical techniques. Firstly, we add a regularization term with small perturbation to $\Sigma$, *i.e.* $\Sigma' = \Sigma + \epsilon_0 I_d$, where $\epsilon_0 = 10^{-4}$ and $I_d$ is the identity matrix. Given that $\Sigma'$ is symmetric and positive definite, the Cholesky decomposition technique is employed whereby $\Sigma' = L \cdot L^T$. $L$ is a lower triangular matrix, facilitating an efficient computation of the inverse $\Sigma^{-1} = (L^{-1})^T \cdot L^{-1}$. Then we filter $\hat{\mathcal{F}}_{\text{OOD}}$ by Mahalanobis distances. The average distances from ID data to their global and inter-class centers *i.e.* $\text{Dist}_{\text{PCA}}^{\text{ID}}$ and $\text{Dist}_{\text{LDA},i_k}^{\text{ID}}$ respectively are obtained by

$$\text{Dist}_{\text{PCA}}^{\text{ID}} = \frac{1}{|\mathcal{F}|} \sum_{v \in \mathcal{F}} \text{Dist}(v, \mu_{\text{PCA}}, \text{cov}(\mathcal{F})),$$

$$\text{Dist}_{\text{LDA},i_k}^{\text{ID}} = \frac{1}{|\mathcal{F}|_{\mathbf{y}=i_k}} \sum_{v \in \mathcal{F}|_{\mathbf{y}=i_k}} \text{Dist}(v, \mu_{\text{LDA},i_k}, \text{cov}(\mathcal{F}|_{\mathbf{y}=i_k})), \tag{8}$$

where $\text{cov}(\cdot)$ is the operator to calculate the covariance matrix of samples $\mathcal{F}$. In the meanwhile, Mahalanobis distances between OOD and ID are calculated:

$$\text{Dist}^{\text{OOD}}(v) = \begin{cases} \text{Dist}(v, \mu_{\text{PCA}}, \text{cov}(\mathcal{F})), & \text{if } |I| = 0, \\ \min_{i_k \in I} \text{Dist}(v, \mu_{\text{LDA},i_k}, \text{cov}(\mathcal{F}|_{\mathbf{y}=i_k})), & \text{if } |I| > 0. \end{cases} \tag{9}$$

And if $|I| > 0$, $i_0 = i_0(v) = \arg\min_{i_k \in I} \text{Dist}(v, \mu_{\text{LDA},i_k}, \text{cov}(\mathcal{F}|_{\mathbf{y}=i_k}))$ is also recorded. The set to be deleted $\mathcal{F}_D$ is

$$\mathcal{F}_D = \begin{cases} \{v \in \hat{\mathcal{F}}_{\text{OOD}} : \text{Dist}^{\text{OOD}}(v) < (1 + \Lambda)\text{Dist}_{\text{PCA}}^{\text{ID}}\}, & \text{if } |I| = 0, \\ \{v \in \hat{\mathcal{F}}_{\text{OOD}} : \text{Dist}^{\text{OOD}}(v) < (1 + \Lambda)\text{Dist}_{\text{LDA},i_0}^{\text{ID}}\}, & \text{if } |I| > 0, \end{cases} \tag{10}$$

where $\Lambda = \lambda \cdot \frac{10}{|\hat{\mathcal{F}}_{\text{OOD}}|} \sum_{v \in \hat{\mathcal{F}}_{\text{OOD}}} (\frac{\text{Dist}^{\text{OOD}}(v)}{\text{Dist}^{\text{ID}}} - 1)$, $\lambda$ is a learnable parameter with the initial value 0.1. $\text{Dist}^{\text{ID}} = \text{Dist}_{\text{PCA}}^{\text{ID}}$ if $|I| = 0$, else $\text{Dist}^{\text{ID}} = \text{Dist}_{\text{LDA},i_0(v)}^{\text{ID}}$. Additionally, we randomly filter the remaining OOD data to no more than $\lceil B/K \rceil + 2$, and the filtered set is denoted as $\mathcal{F}_{RD}$. In this way, we obtain the final generated OOD set $\mathcal{F}_{\text{OOD}} := \hat{\mathcal{F}}_{\text{OOD}} - \mathcal{F}_D - \mathcal{F}_{RD}$, with the label $\mathbf{y} = K + 1$.

**Train-time and test-time OOD detection.** During fine-tuning, training data in the feature space is denoted as $\mathcal{F}_{\text{all}} := \mathcal{F} \cup \mathcal{F}_{\text{OOD}}$, with labels $\mathbf{y} \in \mathcal{Y}$. $\mathcal{F}_{\text{all}}$ is fed into a linear classifier for $K+1$ classes. A loss function $\mathcal{L}$ that integrates a binary ID-OOD classification loss $\mathcal{L}_2$, weighted by the cross-entropy loss $\mathcal{L}_1$, to penalize OOD misclassification and improve ID classification, *i.e.*

$$\mathcal{L} = (1-\gamma)\mathcal{L}_1 + \gamma\mathcal{L}_2, \tag{11}$$

where $\hat{\Phi}$ is depicted as Eq. 61, and

$$\mathcal{L}_1(\mathbf{y}, \mathbf{x}) = -\mathbb{E}_{\mathbf{x} \in \mathcal{X}} \sum_{j=1}^{K+1} \mathbf{y}_j \log(\text{softmax}(\mathbf{f} \circ \mathbf{H}(\mathbf{x}))_j), \tag{12}$$

$$\mathcal{L}_2(\mathbf{y}, \mathbf{x}) = -\mathbb{E}_{\mathbf{x} \in \mathcal{X}} \sum_{j=1}^{2} \hat{\phi}(\mathbf{y})_j \log(\hat{\phi}(\text{softmax}(\mathbf{f} \circ \mathbf{H}(\mathbf{x})))_j). \tag{13}$$

During the test time, the feature set $\mathcal{F}_{\text{test}}$ and logit set LOGITS serve as the inputs. The post-processor VIM is utilized due to its capability to leverage both features and LOGITS effectively. To align the data formats, the first $K$ values of LOGITS are preserved and normalized using the softmax function, maintaining the original notation. We then modify LOGITS to yield the logit matrix LOGITS:

$$\text{LOGITS}_i = \begin{cases} \dfrac{1}{K}\mathbf{1}_K, & \text{if } \arg\max_{i \in \mathcal{Y}} \text{LOGITS}_i = K+1, \\ \text{LOGITS}_i, & \text{else.} \end{cases} \tag{14}$$

Nevertheless, this approach is adaptable to other OOD detection methods, provided that LOGITS is consistently adjusted for the trainer and post-processor. Formally, we also give the pseudocode of GROD displayed in Algorithm 1.

**Theoretical guarantee** A crucial aspect of using transformer networks for OOD detection is defining the limits of their OOD detection capabilities. Thus we incorporate OOD detection learning theory into transformer, including conditions for learnability (Theorem 4) and error approximation of model budgets on transformers (Theorem 5) in Appendix 3. A model is considered learnable for OOD detection if, when trained on sufficient ID data, it is capable of distinguishing OOD samples from ID samples without compromising its classification performance. Both theorems are summarized informally below:

**Theorem 1.** *(Informal Theorem 4, the equivalent conditions for OOD detection learnability on transformer networks) Given the condition $l(\mathbf{y}_2, \mathbf{y}_1) \leq l(K+1, \mathbf{y}_1)$ for any in-distribution labels $\mathbf{y}_1, \mathbf{y}_2 \in \mathcal{Y}$, and ID and OOD have no overlap, then there exists one transformer s.t. OOD detection is learnable, if and only if $|\mathcal{X}| = n < +\infty$. Furthermore, if $|\mathcal{X}| < +\infty$, $\exists \delta > 0$ and a transformer with block budget $m$ and $l$ layers, where $m = (\hat{d}_0, 2, 1, 1, 4)$ and $l = \mathcal{O}(\tau(1/\delta)^{(\hat{d}_0\tau)})$, or $m = (K+1) \cdot (2\tau(2\tau\hat{d}_0+1), 1, 1, \tau(2\tau\hat{d}_0+1), 2\tau(2\tau\hat{d}_0+1))$ and $l = 2$ s.t. OOD detection is learnable.*

**Theorem 2.** *(Informal Theorem 5, error boundary regarding the transformer's budget) Given the condition $l(\mathbf{y}_2, \mathbf{y}_1) \leq l(K+1, \mathbf{y}_1)$, for any in-distribution labels $\mathbf{y}_1, \mathbf{y}_2 \in \mathcal{Y}$, $|\mathcal{X}| = n < +\infty$ and $\tau > K+1$, and set $l = 2$ and $m = (2m_h+1, 1, m_h, 2\tau\hat{d}_0+1, r)$. Using a linear classifier $c$, the probability of OOD detection learnability regarding data distribution $\mathbf{P}$ defined in Definition 5 has a lower bound $\mathbf{P} \geq (1 - \frac{\eta}{|\mathcal{I}|\lambda_0}\tau^2 C_0(\frac{C_1}{m_h^{2\alpha-1}} + \frac{C_2}{r^\beta}(km_h)^\beta))^{(K+1)^{n+1}}$, where $C_0, C_1, C_2, \eta, \lambda_0, |\mathcal{I}|, \alpha, \beta$ can be treated as constants.*

In Theorem 1, we provide the sufficient and necessary conditions for OOD detection learnability in transformers, that is, finite data and higher penalty for OOD misclassification relative to ID misclassification errors. Moreover, we give the specific budget of the learnable transformer in limited width or depth. Theorem 2 explores the scenario where a transformer's parameter scale falls short of the requirements specified in Theorem 1. A lower bound is derived for the probability of learnability in Definition 5, based on the same conditions of penalization and finite data of Theorem 1.

In real-world scenarios, models are trained with finite data. Thus, if the condition $l(\mathbf{y}_2, \mathbf{y}_1) \leq l(K+1, \mathbf{y}_1)$ from Theorem 1 and Theorem 2 is met, optimal performance and error control in OOD

detection can be achieved with appropriate data distributions. However, traditional cross-entropy loss, effective for distinguishing ID categories, does not satisfy this condition. We also conduct experiments under ideal conditions with enough transformer budgets to ensure learnability, revealing a disparity between the reality and the theoretical ideal using cross-entropy loss only (Appendix F). To narrow this gap, we design the ID-OOD binary classification loss function $\mathcal{L}_2$ in Eq. 13, adding it to the cross-entropy loss $\mathcal{L}_1$ weighted by $\gamma$. Since training datasets without OOD cannot fully utilize $\mathcal{L}_2$, we propose a novel method to generate high-quality outliers. Therefore, we form the GROD algorithm, which enhances the generalization of transformers through fine-tuning, supported by our theoretical analysis. Notably, a trade-off between ID classification effectiveness and OOD detection capability exists associated with $\gamma$, as demonstrated in our ablation study (Section 4.3) and experiments on Gaussian mixture datasets (Appendix G). More details on the theoretical analysis and experimental validation using Gaussian mixture datasets are available in Appendix B-G.

# 4 EXPERIMENT RESULTS

In this section, we provide empirical evidence to validate the effectiveness of GROD across a range of real-world classification tasks and types of outliers, including comparison experiments with baselines on various NLP and CV tasks, and the ablation study of key parameters and modules in GROD.

## 4.1 EXPERIMENTAL SETTING

**Models.** We use GROD to strengthen the generalization capability of ViT-B-16 (Dosovitskiy et al., 2020), pre-trained on **ImageNet-1K** (Russakovsky et al., 2015), as the backbone for image classification. For text classification, we explore broader transformer architectures, as two pre-trained models *i.e.* encoder-only BERT (Devlin et al., 2018)and decoder-only GPT-2 small (Radford et al., 2019) are backbones. Details on training hyper-parameters are provided in Appendix I.1.

**Datasets.** For image classification tasks, we use four benchmark datasets *i.e* **CIFAR-10** (Krizhevsky et al., 2009), **CIFAR-100** (Krizhevsky et al., 2009), **ImageNet-200** (Deng et al., 2009), **Tiny ImageNet** (Le & Yang, 2015) and **SVHN** (Netzer et al., 2011). **CIFAR-10**, **CIFAR-100** or **ImageNet-200** serve as ID data, respectively, while three of the others are OOD data. The categories of OOD are disjoint from ID. And **SVHN** is uniquely identified as far-OOD data due to its distinct

Table 1: Image and text datasets for experiments.

| Image Datasets | | | |
| --- | --- | --- | --- |
| ID | | Near-OOD | Far-OOD |
| Classical | **CIFAR-10** **CIFAR-100** | **CIFAR-100** **CIFAR-10** | **Tiny ImageNet** **Tiny ImageNet** | **SVHN** |
| Large-scale | **ImageNet-200** | **CIFAR-10** | **CIFAR-100** | |
| Text Datasets | | | |
| | ID | | OOD |
| Semantic Shift Background Shift | **CLINC150** with intents **IMDB** | | **CLINC150** without intents **Yelp** |

tinct image contents and styles. For outlier exposure method OE and MIXOE, the auxiliary OOD datasets is **Tiny ImageNet-597** for **CIFAR-10** and **CIFAR-100** as ID, and **ImageNet-800** for **ImageNet-200** as ID (Zhang et al., 2023b; Yang et al., 2022a;b; 2021; Bitterwolf et al., 2023). For text classification, we employ datasets in Ouyang et al. (2023) to experiment with detecting semantic and background shift outliers. The semantic shift task uses the dataset **CLINC150** (Larson et al., 2019), where sentences of intents are considered ID, and those lacking intents are treated as semantic shift OOD, following Podolskiy et al. (2021). For the background shift task, the movie review dataset **IMDB** (Maas et al., 2011) serves as ID, while the business review dataset **Yelp** (Zhang et al., 2015) is used as background shift OOD, following Arora et al. (2021). We summarize information like the scale, data type, and ID-OOD similarity of datasets used in the experiment in Table 1. Detailed dataset information can be found in Appendix I.2.

**Evaluation metrics.** We evaluate our models using ID data classification accuracy (ID ACC) and three metrics for binary classification of ID and OOD data: FPR@95 (F), AUROC (A), AUPR for ID test dataset AUPR_IN (I), and AUPR for OOD test dataset AUPR_OUT (O).

## 4.2 MAIN RESULTS

Several prevalent methods are used as baselines for comparison, including MSP (Hendrycks & Gimpel, 2016), ODIN (Liang et al., 2017), VIM (Wang et al., 2022), GEN (Liu et al., 2023a), and ASH (Djurisic et al., 2022) which require only post-processing, and finetuning models G-ODIN (Hsu et al., 2020), NPOS (Tao et al., 2023), CIDER (Ming et al., 2022b), OE (Hendrycks et al., 2018) and MIXOE (Zhang et al., 2023a). All the baselines are offered in the OpenOOD benchmark (Zhang et al., 2023b; Yang et al., 2022a;b; 2021; Bitterwolf et al., 2023).

**Results for image classification.** As discussed in Section 3, GROD employs LDA projection to generate inter-class OOD only when $|I| > 0$ to ensure the stability of the synthesized OOD. To evaluate performance under both scenarios of $|I|$, we use **CIFAR-10** training set with both LDA and PCA, **CIFAR-100** in the transition, and **ImageNet-200** training sets with PCA only.

When $|I| > 0$, the inclusion of both PCA and LDA projections enriches the information in OOD, not only creating virtual OOD around ID but also synthesizing it among ID categories. Correspondingly, the experimental results presented in Table 2 show that GROD surpasses other competitors, achieving SOTA performance across all five evaluation metrics. On average, GROD reduces the FPR@95 from 9.41%, achieved by the current most competitive method, to 0.12%, while enhancing the AUROC from 97.88% to 99.98%. In transition situation *i.e.* $B < K$ but the probability $P(|I| > 0) > 0$, Table 3 shows that mainly using PCA with assistance of LDA on pat of clusters still achieve SOTA performance. When $|I| = 0$, although GROD is not as effective as the LDA-based inter-class OOD generation, it still yields competitive outcomes using only PCA, as evidenced in Tables 4. Because it relies solely on PCA without LDA, this approach falls short in capturing features of inter-class OOD data. In the special case of using **ImageNet-200** as ID and **SVHN** as OOD, the baseline model easily recognizes the difference between ID and OOD. In this situation, additional OOD detection techniques interfere with the results to varying degrees, yet GROD remains robust compared to other competitive fine-tuning methods such as NPOS and CIDER. In general, GROD achieves the best and most stable performance.

Table 2: Quantitative comparison with prevalent methods of the ID classification and OOD detection performance, where the backbone ViT-B-16 pre-trained with **ImageNet-1K** is employed. **CIFAR-10** is the ID Dataset and LDA projections are used for generating inter-class fake outliers. The red, blue and bold fonts denote Top 1,2,3 in ranking.

| OOD Datasets | - | | CIFAR-100 | | | | Tiny ImageNet | | | | SVHN | | | | Average | | |
|---|---|---|---|---|---|---|---|---|---|---|---|---|---|---|---|---|---|---|
| Evaluate Metrics (%) | ID ACC↑ | F↓ | A↑ | I↑ | O↑ | F↓ | A↑ | I↑ | O↑ | F↓ | A↑ | I↑ | O↑ | F↓ | A↑ | I↑ | O↑ |
| Baseline MSP | 96.16 | 29.31 | 91.70 | 92.70 | 90.28 | 21.21 | 94.05 | 95.54 | 92.04 | 15.39 | 95.11 | 92.72 | 97.56 | 21.97 | 93.62 | 93.65 | 93.29 |
| ODIN | | 42.96 | 91.01 | 90.69 | 91.35 | 14.59 | 97.10 | 97.39 | 96.91 | 21.49 | 94.94 | 90.88 | 97.89 | 26.35 | 94.35 | 92.99 | 95.38 |
| PostProcess VIM | 96.16 | 21.59 | 95.43 | 95.64 | 95.38 | 8.52 | 98.39 | 98.68 | 98.14 | 3.26 | 99.39 | 98.61 | 99.78 | 11.12 | 97.74 | 97.64 | 97.77 |
| GEN | | 27.24 | 93.51 | 93.72 | 93.32 | 16.99 | 96.40 | 97.02 | 95.86 | 11.16 | 97.65 | 95.50 | 99.04 | 18.46 | 95.85 | 95.41 | 96.07 |
| ASH | | 26.48 | 93.64 | 93.70 | 93.46 | 16.87 | 96.41 | 96.99 | 95.87 | 9.79 | 98.19 | 96.55 | 99.26 | 17.71 | 96.08 | 95.75 | 96.20 |
| G-ODIN | 95.56 | 82.60 | 70.76 | 68.21 | 72.86 | 64.97 | 83.05 | 83.88 | 83.58 | 62.42 | 89.48 | 68.61 | 95.81 | 70.00 | 81.10 | 73.57 | 84.08 |
| Finetuning+ NPOS | 96.75 | 21.18 | 95.63 | 95.46 | 95.68 | 15.33 | 96.85 | 97.20 | 96.47 | 3.33 | 99.18 | 98.45 | 99.60 | 13.28 | 97.22 | 97.04 | 97.25 |
| PostProcess CIDER | 96.98 | 14.13 | 96.99 | 96.98 | 96.97 | 10.19 | 97.78 | 97.95 | 97.57 | 3.91 | 98.86 | 98.17 | 99.41 | 9.41 | 97.88 | 97.70 | 97.98 |
| OE | 95.70 | 24.74 | 94.62 | 94.75 | 94.58 | 4.97 | 99.18 | 99.30 | 99.08 | 4.39 | 99.04 | 97.94 | 99.59 | 11.37 | 97.61 | 97.33 | 97.75 |
| MIXOE | 96.47 | 20.31 | 95.60 | 95.73 | 95.64 | 10.66 | 97.92 | 98.28 | 97.67 | 5.94 | 98.77 | 97.40 | 99.51 | 12.30 | 97.43 | 97.14 | 97.61 |
| **Ours** | 97.31 | 0.16 | 99.97 | 99.97 | 99.96 | 0.11 | 99.98 | 99.98 | 99.97 | 0.09 | 99.98 | 99.97 | 99.99 | 0.12 | 99.98 | 99.97 | 99.97 |

**Quantitative comparison of the computational cost.** By appropriately selecting $|I|$ in Eq. 5, we ensure an effective fine-tuning stage that minimizes time costs while maximizing performance gains. In the post-processing phase, we save the fine-tuned transformer models without adding extra parameters, highlighting their computational advantages in real-world applications. Figure 2 presents a quantitative comparison of the time costs of various OOD detection methods. Combined with the results from Tables 2, 3, and 4, it is evident that GROD achieves an optimal balance between computational expense and performance enhancement. Methods that rely solely on post-processing for OOD detection, and G-ODIN, exhibit lower fine-tuning time costs but suffer from reduced task performance. Although fine-tuning methods demonstrate competitive capabilities in image ID classification and OOD detection, they are slower than GROD in terms of fine-tuning and post-processing speed.

Table 3: Quantitative comparison with prevalent methods of the ID classification and OOD detection performance using only PCA projection and the transition mode with LDA assistance appeared in GROD algorithm for generating fake OOD data. Take CIFAR-100 as ID.

| OOD Datasets | - | CIFAR-10 | | | | Tiny ImageNet | | | | SVHN | | | | Average | | | |
|---|---|---|---|---|---|---|---|---|---|---|---|---|---|---|---|---|---|
| Evaluate Metrics (%) | ID ACC↑ | F↓ | A↑ | I↑ | O↑ | F↓ | A↑ | I↑ | O↑ | F↓ | A↑ | I↑ | O↑ | F↓ | A↑ | I↑ | O↑ |
| Baseline MSP | 84.34 | 71.11 | 77.17 | 75.37 | 77.56 | 51.34 | 84.15 | 86.55 | 78.08 | 49.58 | 82.07 | 71.41 | 91.97 | 57.34 | 81.13 | 77.78 | 82.54 |
| PostProcess ODIN | | 80.29 | 70.06 | 67.71 | 73.54 | 51.63 | 88.78 | 90.12 | 86.62 | 57.96 | 82.07 | 66.59 | 91.74 | 63.29 | 80.30 | 74.81 | 83.97 |
| VIM | 84.34 | 54.97 | 85.42 | 84.62 | 85.71 | 30.22 | 92.30 | 94.69 | 88.43 | 23.02 | 93.93 | 88.69 | 97.15 | 36.07 | 90.55 | 89.33 | 90.43 |
| GEN | | 73.77 | 80.89 | 77.28 | 82.37 | 45.00 | 89.06 | 91.44 | 84.77 | 35.83 | 90.96 | 81.97 | 96.17 | 51.53 | 86.97 | 83.56 | 87.77 |
| ASH | | 75.26 | 80.61 | 76.87 | 82.19 | 44.68 | 88.98 | 91.42 | 84.62 | 35.87 | 90.88 | 81.85 | 96.12 | 51.94 | 86.82 | 83.38 | 87.64 |
| G-ODIN | 61.40 | 89.14 | 47.52 | 51.63 | 47.76 | 74.07 | 68.87 | 77.48 | 54.99 | 30.77 | 93.15 | 95.55 | 89.40 | 64.66 | 69.85 | 74.89 | 64.05 |
| Finetuning+ NPOS | 84.76 | 43.53 | 89.63 | 89.14 | 90.42 | 33.36 | 91.72 | 94.14 | 88.38 | 38.86 | 90.62 | 81.67 | 96.04 | 38.58 | 90.66 | 88.32 | 91.61 |
| PostProcess CIDER | 84.87 | 44.47 | 89.41 | 88.74 | 90.23 | 33.08 | 91.83 | 94.18 | 88.60 | 30.36 | 93.48 | 84.46 | 97.36 | 35.97 | 91.57 | 89.13 | 92.06 |
| OE | 74.97 | 73.80 | 73.72 | 72.86 | 75.75 | 22.02 | 96.64 | 97.11 | 96.46 | 41.66 | 92.97 | 81.74 | 97.37 | 45.83 | 87.78 | 83.97 | 89.86 |
| MIXOE | 77.84 | 71.07 | 75.84 | 74.76 | 78.55 | 49.01 | 88.61 | 91.22 | 86.03 | 49.08 | 92.14 | 78.58 | 97.26 | 56.39 | 85.53 | 81.52 | 87.28 |
| Ours (PCA) | 86.21 | 43.38 | 88.00 | 88.01 | 87.94 | 38.84 | 91.44 | 93.46 | 87.91 | 23.38 | 94.59 | 87.88 | 98.63 | 35.20 | 91.34 | 89.78 | 91.49 |
| Ours | 86.10 | 38.22 | 90.45 | 90.17 | 90.88 | 27.98 | 93.32 | 95.38 | 90.52 | 22.12 | 93.70 | 88.91 | 96.59 | 29.44 | 92.49 | 91.49 | 92.66 |

Table 4: Quantitative comparison with prevalent methods of the ID classification and OOD detection performance using only PCA projection for generating fake OOD data. Take ImageNet-200 as ID.

| OOD Datasets | - | CIFAR-10 | | | | CIFAR-100 | | | | SVHN | | | | Average | | | |
|---|---|---|---|---|---|---|---|---|---|---|---|---|---|---|---|---|---|
| Evaluate Metrics (%) | ID ACC↑ | F↓ | A↑ | I↑ | O↑ | F↓ | A↑ | I↑ | O↑ | F↓ | A↑ | I↑ | O↑ | F↓ | A↑ | I↑ | O↑ |
| Baseline MSP | 89.09 | 25.28 | 92.79 | 93.05 | 91.98 | 32.09 | 93.02 | 93.18 | 92.69 | 1.01 | 99.72 | 99.54 | 99.84 | 19.46 | 95.18 | 95.26 | 94.84 |
| PostProcess ODIN | | 40.38 | 89.34 | 91.29 | 91.24 | 33.98 | 93.69 | 93.23 | 91.58 | 23.66 | 93.65 | 93.58 | 95.09 | 32.67 | 92.23 | 92.70 | 92.64 |
| VIM | 89.09 | 27.14 | 92.48 | 93.03 | 90.54 | 35.49 | 91.27 | 90.94 | 89.19 | 9.12 | 95.12 | 94.93 | 95.54 | 23.92 | 92.96 | 92.97 | 91.76 |
| GEN | | 33.79 | 83.84 | 87.94 | 75.21 | 34.40 | 85.47 | 89.04 | 76.56 | 20.86 | 86.75 | 86.97 | 88.20 | 29.68 | 85.35 | 87.98 | 79.99 |
| ASH | | 33.66 | 92.26 | 91.79 | 92.12 | 39.49 | 91.76 | 90.16 | 90.42 | 1.50 | 99.56 | 99.33 | 99.62 | 24.88 | 94.53 | 93.76 | 94.05 |
| G-ODIN | 89.28 | 84.39 | 66.20 | 63.15 | 68.85 | 84.69 | 73.99 | 69.78 | 75.80 | 22.49 | 95.98 | 89.78 | 98.53 | 63.86 | 78.72 | 74.24 | 81.06 |
| Finetuning+ NPOS | 89.96 | 14.20 | 95.32 | 96.59 | 93.77 | 26.33 | 93.28 | 93.13 | 91.91 | 13.87 | 94.59 | 92.54 | 95.01 | 18.13 | 94.40 | 94.09 | 93.56 |
| PostProcess CIDER | 90.13 | 14.51 | 95.37 | 96.23 | 93.73 | 26.01 | 93.80 | 93.81 | 91.95 | 7.39 | 96.09 | 95.83 | 96.07 | 15.97 | 95.09 | 95.29 | 93.92 |
| OE | 89.48 | 25.33 | 92.66 | 93.02 | 91.74 | 33.08 | 92.99 | 93.10 | 92.68 | 0.69 | 99.78 | 99.64 | 99.87 | 19.70 | 95.14 | 95.25 | 94.76 |
| MIXOE | 90.49 | 25.43 | 92.46 | 92.75 | 91.23 | 33.71 | 92.60 | 92.69 | 92.09 | 1.41 | 99.63 | 99.36 | 99.80 | 20.18 | 94.90 | 94.93 | 94.37 |
| Ours | 90.71 | 21.98 | 93.86 | 94.85 | 93.87 | 28.39 | 93.15 | 93.98 | 92.84 | 4.41 | 98.72 | 97.78 | 98.97 | 18.26 | 95.24 | 95.54 | 95.23 |

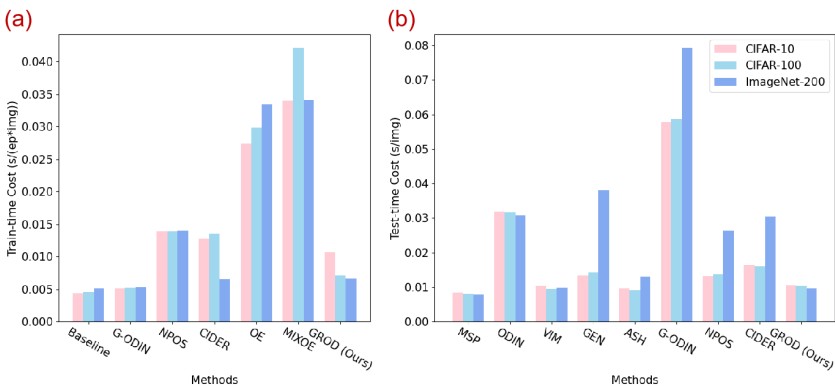

Figure 2: Quantitative comparison of the computational costs associated with various OOD detection methods on image datasets is presented, with fine-tuning and post-processing times reported in subfigures (a) and (b), respectively. Methods with only post-processing including MSP, ODIN, VIM, GEN, and ASH are used after "baseline" fine-tuning. Outlier exposure methods OE and MIXOE use MSP for post-processing.

**Results for text classification.** Table 4.2 presents the results for text classification. As two ID datasets, **IMDB** and **CLINC150** have two and ten categories respectively, with $|I| > 0$ in both cases. Hence, both PCA and LDA projections are applied to these datasets. In line with the results and analysis of image classification in Table 2, GROD outperforms other powerful OOD detection

techniques. While many popular OOD detection algorithms are rigorously tested on image datasets, their effectiveness on text datasets does not exhibit marked superiority, as Table 4.2 illustrates. In addition, methods like ODIN (Liang et al., 2017) and G-ODIN (Hsu et al., 2020), which compute data gradients, necessitate floating-point number inputs. However, the tokenizer-encoded long integers used as input tokens create data format incompatibilities when attempting to use BERT and GPT-2 alongside ODIN or G-ODIN. Given their marginal performance on image datasets, these methods are excluded from text classification tasks. For the decoder-only GPT-2 model, some methods (Baseline, GEN) are compatible with both models using CLS tokens as features and without them, as they only require logits for processing. Others are only compatible with transformers with CLS tokens since they combine features and logits. We test two modes (with/without CLS token), labeled Method-C (with CLS) and Method-L (without CLS). As shown in Table 4.2, GROD consistently improves model performance across both image and text datasets and various OOD detection tasks, highlighting its versatility and broad applicability.

Table 5: Quantitative comparison with prevalent methods of the ID classification and OOD detection performance, where the pre-trained BERT (a) and GPT-2 (b) are employed. Experimental results on two typical OOD in the text OOD detection, *i.e.* background shift OOD and semantic shift OOD are reported.

(a) BERT

| OOD Detection Type | | Background Shift | | | | | Semantic Shift | | | | |
|---|---|---|---|---|---|---|---|---|---|---|---|
| ID Datasets | | **IMDB** | | | | | **CLINC150** with Intents | | | | |
| OOD Datasets | | **Yelp** | | | | | **CLINC150** with Unknown Intents | | | | |
| Evaluate Metrics (%) | | ID ACC↑ | F↓ | A↑ | I↑ | O↑ | ID ACC↑ | F↓ | A↑ | I↑ | O↑ |
| Baseline | MSP | 91.36 | **57.72** | 74.28 | 73.28 | 74.60 | 97.78 | 37.11 | 92.31 | 97.70 | 74.66 |
| | VIM | | 64.00 | **74.61** | 70.17 | 76.05 | | 29.33 | 93.58 | 98.03 | 80.99 |
| PostProcess | GEN | 91.36 | 57.63 | 74.28 | **73.28** | 74.60 | 97.78 | **36.27** | 92.27 | 97.47 | 79.43 |
| | ASH | | 73.27 | 71.43 | 65.11 | **76.64** | | 40.67 | **92.56** | **97.60** | **79.70** |
| Finetuning+ | NPOS | 90.36 | 76.31 | 68.48 | 61.84 | 74.56 | 95.62 | 49.89 | 83.57 | 95.64 | 48.52 |
| PostProcess | CIDER | **91.28** | 59.71 | **78.10** | **75.09** | **79.07** | 95.93 | 45.04 | 86.39 | 96.44 | 55.17 |
| | **Ours** | 91.47 | 52.89 | 78.86 | 77.61 | 79.63 | **97.66** | 24.00 | 94.58 | 98.52 | 82.47 |

(b) GPT-2

| OOD Detection Type | | Background Shift | | | | | Semantic Shift | | | | |
|---|---|---|---|---|---|---|---|---|---|---|---|
| ID Datasets | | **IMDB** | | | | | **CLINC150** with Intents | | | | |
| OOD Datasets | | **Yelp** | | | | | **CLINC150** with Unknown Intents | | | | |
| Evaluate Metrics (%) | | ID ACC↑ | F↓ | A↑ | I↑ | O↑ | ID ACC↑ | F↓ | A↑ | I↑ | O↑ |
| Baseline-L | MSP-L | 88.56 | 100.0 | 59.10 | 67.81 | **70.51** | 97.09 | 41.76 | 91.81 | 97.92 | 72.86 |
| Baseline-C | MSP-C | 87.93 | 100.0 | 58.41 | **64.50** | 67.59 | 97.44 | 60.36 | 86.29 | 96.26 | 55.34 |
| | VIM | 87.93 | 84.81 | 58.55 | 51.60 | 63.95 | 97.44 | 27.53 | 93.71 | 98.21 | 79.25 |
| PostProcess | GEN-L | 88.56 | 57.80 | 75.00 | 73.55 | 75.43 | 97.08 | **33.29** | 92.46 | 97.77 | 76.76 |
| | GEN-C | 87.93 | **76.90** | **65.84** | 60.79 | 69.52 | 97.44 | 32.87 | **93.24** | **98.11** | 77.25 |
| | ASH | 87.93 | 85.41 | 60.45 | 50.97 | 68.66 | 97.44 | 41.27 | 92.73 | 97.80 | **78.21** |
| Finetuning+ | NPOS | 88.08 | 96.92 | 50.23 | 39.94 | 60.67 | 97.33 | 66.24 | 77.01 | 93.47 | 43.90 |
| PostProcess | CIDER | 87.89 | 84.46 | 59.71 | 52.03 | 62.99 | **97.43** | 57.27 | 81.40 | 95.00 | 49.16 |
| | **Ours** | **88.03** | 75.17 | 66.91 | 61.96 | 70.95 | 97.51 | 23.80 | 94.90 | 98.55 | 84.75 |

## 4.3 ABLATION STUDY

Comprehensive ablation studies are conducted to explore hyper-parameters and optimization strategies, where Figure 5 shows the ablation experiments for key parameters $\gamma$, $\alpha$, and $num$, and the ablation results of modules in GROD are displayed in Table 4.3.

**Abaltion study on hyper-parameters.** Our method introduce three hyperparameters $\alpha$, $num$ and $\gamma$. $num = 1$ is empirically an optimal choice, which is consistent with the conclusion in Fort et al. (2021) that even adding one or two OOD can raise the OOD detection performance of transformers. The ablation results regarding $\gamma$ in Fig. 5 show that $\gamma \in [0.1, 0.3]$ benefits the task performance, which is also in line with the theoretical insights and the classification (learned by $\mathcal{L}_1$) and OOD detection (learned by $\mathcal{L}_2$) goal of the task. Therefore, $num$ and $\gamma$ have their optimal solution.

As to $\alpha$, we recommend $\alpha = 10^{-3}$ if $LDA$ is used, otherwise a larger value should be taken to capture a global characteristic of outliers. We have analyzed these parameters in detail, and give an explanation from the perspective of OOD detection learning theory in Appendix J.

Table 6: Ablation experiments. The ID dataset is **CIFAR-10** and the backbone is ViT-B-16 pretrained with **ImageNet-1K**. Respectively, $\mathcal{L}_2$, $\mathcal{F}_{\text{OOD}}$, $Maha$ represent whether to use the binary loss function $\mathcal{L}_2$, fake OOD data generation and Mahalanobis distance filtration. Outliers with Gaussian distribution and randomly uniform distribution are denoted as 'G' and 'U' respectively.

| OOD Datasets | | | - | CIFAR-100 | | | | Tiny ImageNet | | | | SVHN | | | | Average | | | |
|---|---|---|---|---|---|---|---|---|---|---|---|---|---|---|---|---|---|---|---|
| \multicolumn Evaluate Metrics (%) | | | ID ACC↑ | F↓ | A↑ | I↑ | O↑ | F↓ | A↑ | I↑ | O↑ | F↓ | A↑ | I↑ | O↑ | F↓ | A↑ | I↑ | O↑ |
| $\mathcal{L}_2$ | $\mathcal{F}_{\text{OOD}}$ | $Maha$ | | | | | | | | | | | | | | | | | |
| | | | 96.16 | 21.59 | 95.43 | 95.64 | 95.38 | 8.52 | **98.39** | **98.68** | **98.14** | 3.26 | 99.39 | 98.61 | 99.78 | 11.12 | 97.74 | 97.64 | 97.77 |
| | ✓ | ✓ | **96.96** | 22.66 | 94.98 | 95.13 | 94.94 | 13.04 | 96.98 | 97.68 | 96.27 | 4.69 | 99.18 | 98.11 | 99.70 | 13.46 | 97.05 | 96.97 | 96.97 |
| ✓ | | | 97.00 | 18.02 | 96.32 | 96.32 | 96.49 | 8.78 | 98.45 | 98.70 | 98.27 | 2.76 | **99.45** | 98.58 | **99.81** | 9.85 | 98.07 | **97.87** | 98.19 |
| ✓ | ✓ | | 96.68 | 21.17 | 95.57 | 95.52 | 95.78 | 9.41 | 98.27 | 98.58 | 98.04 | 0.49 | 99.83 | 99.77 | 99.88 | **10.36** | 97.89 | 97.96 | 97.90 |
| ✓ | G | ✓ | 96.86 | 20.22 | **96.10** | **95.95** | **96.30** | 10.92 | 97.97 | 98.21 | 97.79 | **2.29** | 99.41 | **98.74** | 99.75 | 11.14 | 97.83 | 97.63 | 97.95 |
| ✓ | U | ✓ | 96.67 | **19.39** | 95.84 | 95.90 | 95.92 | 10.06 | 98.03 | 98.42 | 97.70 | 4.03 | 99.22 | 98.11 | 99.72 | 11.16 | 97.70 | 97.48 | 97.78 |
| ✓ | ✓ | ✓ | 97.31 | 0.16 | 99.97 | 99.97 | 99.96 | 0.11 | 99.98 | 99.98 | 99.97 | 0.09 | 99.98 | 99.97 | 99.99 | 0.12 | 99.98 | 99.97 | 99.97 |

**Ablation on key modules in GROD.** GROD comprises three key modules: adjusting the loss function, generating virtual OOD data, and employing the Mahalanobis distance filtering mechanism, denoted as $\mathcal{L}_2$, $\mathcal{F}_{\text{OOD}}$, and $Maha$, respectively. Table 4.3 presents the ablation studies for these modules. $\mathcal{L}_2$ alone can enhance model optimization, whereas $\mathcal{F}_{\text{OOD}}$ and $Maha$ contribute positively when integrated with $\mathcal{L}_2$. Utilizing all three strategies concurrently yields optimal performance, confirming that GROD effectively synergizes these modules to assign penalties associated with OOD and sharpen the precision of the ID-OOD decision boundary. We have also tested two simple methods to generate outliers *i.e.* outliers with Gaussian distribution and randomly uniform distribution to validate the positive utility of our synthesis strategy, denoted as 'G' and 'U' in Table 4.3, respectively. Moreover, features $\mathcal{F}_{\text{all}}$, along with the prediction LOGITS LOGITS of GROD and the baseline, are visualized under t-SNE dimensional embedding (Appendix K), which illustrate the efficiency of GROD directly.

## 5 CONCLUSION AND FUTURE WORK

In this paper, we propose GROD, a novel algorithm that enhances the generalization of transformers during fine-tuning and leverages them for OOD detection. Inspired by theoretical insights, GROD minimizes the gap between optimal generalization and practical performance with rigorous mathematical foundations, including two theorems deriving conditions and error bounds for OOD detection learnability in transformer networks. In both NLP and CV tasks involving outliers, transformers equipped with GROD show superior performance compared to standard transformers. Furthermore, its effectiveness is validated by robust ablation studies and visualizations. Our research enriches OOD detection theory by integrating with transformers, offering guidance for their use in both research of generalization and applications of transformers in OOD detection, with GROD demonstrating adaptability across multiple data formats. The proposal of a "gold standard" for measuring virtual OOD can fully utilize theoretical results and unleash the generalization potential of transformers, which is of importance in future research. We believe our findings will give insights into the generalization and reliability of transformers, and motivate further research on OOD detection and model security. In the future, we will focus on improving GROD for stable inter-class OOD data generation in multi-class tasks and more deeply explore the layer of transformer feature spaces.

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

## A   DETAILED RELATED WORKS.

**Application of OOD detection.**     The recent advancements in OOD detection models and algorithms have been significant (Sun et al., 2022; Liu et al., 2023b; Cai & Li, 2023). Typically, OOD detection methods leverage both post-processing techniques and training strategies, which can be implemented either separately or in combination (Zhang et al., 2023b). Key post-processing techniques include the use of distance functions (Denouden et al., 2018), the development of scoring functions (Ming et al., 2022a), and the integration of disturbance terms (Hsu et al., 2020), among others. Several methods introduce training strategies for OOD detection models. For instance, Tao et al. (2023) suggests loss functions to facilitate the learning of compact representations, while Graham et al. (2023); Jiang et al. (2023) innovatively employs reconstruction models to pinpoint abnormal data. In addition, the transformer architecture has gained popularity in OOD detection, prized for its robust feature representation capabilities (Koner et al., 2021; Fort et al., 2021).

**Leveraging auxiliary outliers.**     Leveraging auxiliary data for OOD detection has emerged as a prominent strategy, broadly categorized into Outlier Exposure (OE) and outlier generating methods. OE involves utilizing external datasets as outliers during training to calibrate the model's ability to distinguish ID from OOD samples (Kirchheim & Ortmeier, 2022; Chen et al., 2021). Hendrycks et al. (Hendrycks et al., 2018) first proposed OE, demonstrating the effectiveness of using extra datasets, while Zhu et al. (Zhu et al., 2023) enhanced this method by introducing diversified outlier exposure through informative extrapolation. Zhang et al. (Zhang et al., 2023a) further extended this to fine-grained environments with Mixture Outlier Exposure, emphasizing the relevance of auxiliary outliers to specific tasks. Generative-based methods, on the other hand, utilize generative models and feature modeling to create synthetic data that imitates OOD characteristics, thus enabling the generation of diverse and informative outlier samples without the need for predefined outlier datasets. VOS (Du et al., 2022) models the features as a Gaussian mixture distribution and samples out-of-distribution data in low-likelihood areas. NPOS (Tao et al., 2023) further uses KNN to generate out-of-distribution features. OpenGAN (Kong & Ramanan, 2021) pioneered this approach with GANs to generate open-set examples, and Wang et al. (Wang et al., 2023) advanced it by employing implicit outlier transformations for more diverse OOD representations. Zheng et al. (Zheng et al., 2023) addressed scenarios with noisy or unreliable auxiliary data, refining generative processes for robust outlier synthesis. Du et al. (2024) is highlighted on generating high-resolution outliers in the pixel space using diffusion models. These methods, by leveraging external or synthesized data, represent critical progress in enhancing OOD detection and improving model robustness in open-world scenarios.

**Theory of OOD detection.**     Theoretical research into OOD detection has recently intensified. Morteza & Li (2022) examines maximum likelihood on mixed Gaussian distributions and introduces a GEM log-likelihood score. Zhang et al. (2021) reveals that even minor errors in density estimation can result in OOD detection failures. Fang et al. (2022) presents the first application of Probably Approximately Correct (PAC) learning theory to OOD detection, deriving the Impossibility Theorem and exploring conditions under which OOD detection can be learned in previously unknown spaces. Moreover, Yang et al. (2021) has pioneered the concept of generalized OOD detection, noting its commonalities with anomaly detection (AD) and open set recognition (OSR) (Fang et al., 2021). To the best of our knowledge, no comprehensive theory of OOD detection for transformers has been established yet.

**Transformers and their universal approximation power**     Transformers bring inspiration and progress to OOD detection, with algorithms utilizing their self-attention mechanism achieving noteworthy results (Koner et al., 2021; Hendrycks et al., 2020; Podolskiy et al., 2021; Zhou et al., 2021). Understanding the expressivity of transformers is vital for their application in OOD detection. Current research predominantly explores two main areas: formal language theory and approximation theory (Strobl et al., 2023). The former examines transformers as recognizers of formal languages, clarifying their lower and upper bounds (Hahn, 2020; Chiang et al., 2023; Merrill & Sabharwal, 2024). Our focus, however, lies primarily in approximation theory. The universal approximation property (UAP) of transformers, characterized by fixed width and infinite depth, was initially demonstrated by Yun et al. (2019). Subsequent studies have expanded on this, exploring UAP under various

conditions and transformer architectures (Yun et al., 2020; Kratsios et al., 2021; Luo et al., 2022; Alberti et al., 2023). As another important development, Jiang & Li (2023) established the UAP for architectures with a fixed depth and infinite width and provided Jackson-type approximation rates for transformers.

# B    NOTATIONS AND PRELIMINARIES

**Notations.**    More notations for theoretical analysis can be found here. $|\cdot|$ indicates the count of elements in a set, and $||\cdot||2$ represents the $L_2$ norm in Euclidean space. The data priori-unknown distribution spaces include $\mathcal{D}_{XY}^{all}$, which is the total space including all distributions; $\mathcal{D}_{XY}^s$, the separate space with distributions that have no ID-OOD overlap; $\mathcal{D}_{XY}^{D_{XY}}$, a single-distribution space for a specific dataset distribution denoted as $D_{XY}$; $\mathcal{D}_{XY}^F$, the Finite-ID-distribution space containing distributions with a finite number of ID examples; and $\mathcal{D}_{XY}^{\mu,b}$, the density-based space characterized by distributions expressed through density functions. A superscript may be added on $D_{XY}$ to denote the number of data points in the distribution. The model hypothesis space is represented by $\mathcal{H}$, and the binary ID-OOD classifier is defined as $\Phi$. These notations, consistent with those used in Fang et al. (2022), facilitate a clear understanding of OOD detection learning theory.

Several notations related to spaces and measures of function approximation also require further clarification to enhance understanding of the theoretical framework. $\mathcal{C}$ and $C$ denote the compact function set and compact data set, respectively. Complexity measures for the self-attention blocks within transformers are denoted as $C_0(\cdot)$ and $C_1^{(\alpha)}(\cdot)$, while $C_2^{(\beta)}(\cdot)$ represents a regularity measure for the feed-forward neural networks within transformers. These measures indicate the approximation capabilities of transformers, with $\alpha$ and $\beta$ being the convergence orders for Jackson-type estimation. $\widetilde{\mathcal{C}}^{(\alpha,\beta)}$ within $\mathcal{C}$ is the function space where Jackson-type estimation is applicable. Given the complexity of the mathematical definitions and symbols involved, we aim to provide clear conceptions to facilitate a smooth understanding of our theoretical approach. These mathematical definitions regarding function approximation follow those presented by Jiang & Li (2023).

**Goal of theory.**    As an impressive work on OOD detection theory, Fang et al. (2022) defines strong learnability for OOD detection and has applied its PAC learning theory to the FCNN-based and score-based hypothesis spaces:

**Definition 1.** *(Fang et al., 2022)(Strong learnability) OOD detection is strongly learnable in $\mathcal{D}_{XY}$, if there exists an algorithm* $\mathbf{A}$: $\cup_{n=1}^{+\infty}(\mathcal{X} \times \mathcal{Y})^n \to \mathcal{H}$ *and a monotonically decreasing sequence* $\epsilon(n)$ *s.t.* $\epsilon(n) \to 0$*, as* $n \to +\infty$*, and for any domain* $D_{XY} \in \mathcal{D}_{XY}$,

$$\mathbb{E}_{S \sim D_{X_I Y_I}^n}[\mathcal{L}_D^\alpha(\mathbf{A}(S)) - inf_{h \in \mathcal{H}}\mathcal{L}_D^\alpha(h)] \le \epsilon(n), \forall \alpha \in [0,1].$$

In the data distribution spaces under our study, the equality of strong learnability and PAC learnability has been proved. So we only need to gain strong learnability to verify the proposed theorems.

**Theorem 3.** *(Fang et al., 2022) (Informal, learnability in FCNN-based and score-based hypothesis spaces)*

*If* $l(y_2, y_1) \le l(K+1, y_1)$ *for any in-distribution labels* $y_1$ *and* $y_2 \in \mathcal{Y}$*, and the hypothesis space* $\mathcal{H}$ *is FCNN-based or corresponding score-based, then OOD detection is learnable in the separate space* $\mathcal{D}_{XY}^s$ *for* $\mathcal{H}$ *if and only if* $|\mathcal{X}| < +\infty$.

Inspired by Theorem 3 and its Proof, the goal of our theory is proposed as follows:

> Goal: Given a transformer hypothesis space $\mathcal{H}_{\text{TOOD}}$, what are necessary or sufficient conditions to ensure the learnability of OOD detection? Furthermore, we try to derive the approximation rates and error bounds of OOD detection.

**The transformer hypothesis space.**    Under the goal of investigating the OOD detection learning theory on transformers, our research defines a fixed transformer hypothesis space for OOD detection $\mathcal{H}_{\text{tood}}$. A transformer block $\text{Block}(\cdot) : \mathbb{R}^{\hat{d} \times \tau} \to \mathbb{R}^{\hat{d} \times \tau}$ consists of a self-attention layer $\text{Att}(\cdot)$ and

a feed-forward layer $\text{FF}(\cdot)$, *i.e.*

$$\text{Att}(\mathbf{h}_l) = \mathbf{h}_l + \sum_{i=1}^{h} W_O^i W_V^i \mathbf{h}_l \cdot \sigma[(W_K^i \mathbf{h}_l)^T W_Q^i \mathbf{h}_l], \tag{15}$$

$$\mathbf{h}_{l+1} = \text{FF}(\mathbf{h}_l) = \text{Att}(\mathbf{h}_l) + W_2 \cdot \text{Relu}(W_1 \cdot \text{Att}(\mathbf{h}_l) + \mathbf{b_1}\mathbf{1}_n^T) + \mathbf{b_2}\mathbf{1}_n^T, \tag{16}$$

with $W_O^i \in \mathbb{R}^{\hat{d} \times m_v}$, $W_V^i \in \mathbb{R}^{m_V \times \hat{d}}$, $W_K^i, W_Q^i \in \mathbb{R}^{m_h \times \hat{d}}$, $W_1 \in \mathbb{R}^{r \times \hat{d}}$, $W_2 \in \mathbb{R}^{\hat{d} \times r}$, $b_1 \in \mathbb{R}^r$ and $b_2 \in \mathbb{R}^{\hat{d}}$. Besides, $\mathbf{h}_l \in \mathbb{R}^{\hat{d} \times \tau}$ is the hidden state of $l$-th transformer block with $\mathbf{h}_0 \in \mathbb{R}^{\hat{d} \times \tau}$ is the input data $\mathcal{X} \in \mathbb{R}^{(\hat{d}_0 \times \tau) \times n}$ (with position encoding) after a one-layer FCNN $F : \mathbb{R}^{\hat{d}_0 \times \tau} \to \mathbb{R}^{\hat{d} \times \tau}$, and $\sigma(\cdot)$ is the column-wise softmax function. We denote $d := \hat{d} \times \tau$ and $d_0 := \hat{d}_0 \times \tau$ for convenience.

The computation budget of a transformer block includes the number of heads $h$, the hidden layer size $r$ of $FF$, $m_h$, $m_V$, and $n$, denoted by $m = (\hat{d}, h, m_h, m_V, r)$. Formally, a classic transformer block with a budget of $m$ and $l$-th layer can be depicted as $\text{Block}_l^{(m)}(\cdot) = \text{FF} \circ \text{Att}(\cdot)$. Transformer is a composition of transformer blocks, by which we define transformer hypothesis space $\mathcal{H}_{\text{Trans}}$:

**Definition 2.** *(Transformer hypothesis space) The transformer hypothesis space is $\mathcal{H}_{\text{Trans}}$ is*

$$\mathcal{H}_{\text{Trans}} = \cup_l \mathcal{H}_{\text{Trans}}^{(l)} = \cup_l \cup_m \mathcal{H}_{\text{Trans}}^{(l,m)} \tag{17}$$

*where $\mathcal{H}_{\text{Trans}}^{(l)}$ is the transformer hypothesis space with $l$ layers, and $\mathcal{H}_{\text{Trans}}^{(l,m)}$ is the transformer hypothesis space with $l$ layers of $\text{Block}_i^{(m)}(\cdot)$, $i \in \{1, 2, \ldots, l\}$. More specifically,*

$$\mathcal{H}_{\text{Trans}}^{(l,m)} := \{\hat{H} : \hat{H} = \text{Block}_l^{(m)} \circ \text{Block}_{l-1}^{(m)} \circ \cdots \circ \text{Block}_1^{(m)} \circ F\}. \tag{18}$$

By the Definition 2 that $\forall \hat{H} \in \mathcal{H}_{\text{Trans}}$, $\hat{H}$ is a map from $\mathbb{R}^{d_0 \times n}$ to $\mathbb{R}^{d \times n}$. To match the OOD detection task, we insert a classifier $c : \mathbb{R}^d \to \mathcal{Y}$ applied to each data as follows:

**Definition 3.** *(Classifier ) $c : \mathbb{R}^d \to \mathcal{Y}$ is a classical classifier with forms:*

$$(\text{maximum value}) \; c(\mathbf{h}_l) = \arg\max_{k \in \mathcal{Y}} f^k(\mathbf{h}_l), \tag{19}$$

$$(\text{score-based}) \; c(\mathbf{h}_l) = \begin{cases} K + 1, & E(f(\mathbf{h}_l)) < \lambda, \\ \arg\max_{k \in \mathcal{Y}} f^k(\mathbf{h}_l), & E(f(\mathbf{h}_l)) \geq \lambda, \end{cases} \tag{20}$$

*where $f^k$ is the $k$-th coordinate of $f \in \{\hat{f} \in \mathbb{R}^d \to \mathbb{R}^{K+1}\}$, which is defined by*

$$f^k(\mathbf{h}_l) = W_{4,k}(W_{3,k}\mathbf{h}_l + b_{3,k})^T + b_{4,k}. \tag{21}$$

$W_{3,k} \in \mathbb{R}^{1 \times \hat{d}}$, $W_{4,k}, b_{3,k} \in \mathbb{R}^{1 \times \tau}$ and $b_{4,k} \in \mathbb{R}$. And $E(\cdot)$ is the scoring functions like softmax-based function (*Hendrycks & Gimpel, 2016*) and energy-based function (*Liu et al., 2020*).

Now, we can naturally derive the Definition of transformer hypothesis space for OOD detection:

**Definition 4.** *(Transformer hypothesis space for OOD detection)*

$$\mathcal{H}_{\text{tood}} := \{H \in \mathbb{R}^{d_0 \times n} \to \mathcal{Y}^n : H = c \circ \hat{H}, c \text{ is a classifier in Definition 3}, \hat{H} \in \mathcal{H}_{\text{Trans}}\} \tag{22}$$

*Similarly, we denote $\mathcal{H}_{\text{tood}}^{(l)}$ and $\mathcal{H}_{\text{tood}}^{(l,m)}$ as in Definition 2.*

## C  THEORETICAL RESULTS OF OOD DETECTION USING TRANSFORMERS

In the five priori-unknown spaces defined in Fang et al. (2022), **the total space $\mathcal{D}_{XY}^{\text{all}}$** has been thoroughly discussed. Following the impossible Theorem, OOD detection is NOT learnable due to the overlapping of datasets, even when the budget $m \to +\infty$. Consequently, we shift our focus to the learning theory of transformers within the other four spaces: $\mathcal{D}_{XY}^{D_{XY}}$, $\mathcal{D}_{XY}^s$, $\mathcal{D}_{XY}^F$, and $\mathcal{D}_{XY}^{\mu,b}$. For each space, we investigate whether OOD detection is learnable under $\mathcal{H}_{\text{tood}}$, considering the specific constraints, conditions, or assumptions. If OOD detection is found to be learnable, we then explore the approximation of rates and boundaries to further understand the generalization capabilities of transformers.

## C.1 OOD DETECTION IN THE SEPARATE SPACE

Since the overlap is a crucial factor preventing models from successfully learning OOD detection, a natural perspective is to explore the separate space $\mathcal{D}_{XY}^s$.

**Conditions for learning with transformers.** Firstly, by Theorem 10 in Fang et al. (2022) and Theorems 5, 8 in Bartlett & Maass (2003), OOD detection is NOT learnable in $\mathcal{D}_{XY}^s$. It means OOD detection also has the impossible Theorem in $\mathcal{D}_{XY}^s$ for $\mathcal{H}_{\text{tood}}$. So we enquire about the conditions for $\mathcal{H}_{\text{tood}}$ to meet the requirements of learnability, deriving Theorem 4:

**Theorem 4.** *(Necessary and sufficient condition for OOD detection learnability on transformers)*

*Given the condition $l(\mathbf{y}_2, \mathbf{y}_1) \leq l(K + 1, \mathbf{y}_1)$, for any in-distribution labels $\mathbf{y}_1, \mathbf{y}_2 \in \mathcal{Y}$, then OOD detection is learnable in the separate space $\mathcal{D}_{XY}^s$ for $\mathcal{H}_{\text{tood}}$ if and only if $|\mathcal{X}| = n < +\infty$. Furthermore, if $|\mathcal{X}| < +\infty$, $\exists \delta > 0$ and $g \in \mathcal{H}_{\text{tood}}^{(m,l)}$, where $Block(\cdot)$ budget $m = (\hat{d}_0, 2, 1, 1, 4)$ and the number of $Block(\cdot)$ layer $l = \mathcal{O}(\tau(1/\delta)^{(\hat{d}_0\tau)})$, or $m = (K + 1) \cdot (2\tau(2\tau\hat{d}_0 + 1), 1, 1, \tau(2\tau\hat{d}_0 + 1), 2\tau(2\tau\hat{d}_0 + 1))$ and $l = 2$ s.t. OOD detection is learnable with $g$.*

Theorem 4 gives the necessary and sufficient conditions for OOD detection learnability on transformers with a fixed depth or width. Detailed proof and remarks on inspection can be found in the Appendix D.

**Approximation of rates and boundaries.** To further investigate the approximation rates and boundaries as the budget $m$ grows, we formulate Jackson-type estimates for OOD detection learnability using transformer models. Before presenting the main Theorem 5, it is essential to define the probability associated with the learnability of OOD detection:

**Definition 5.** *(Probability of the OOD detection learnability) Given a domain space $\mathcal{D}_{XY}$ and the hypothesis space $\mathcal{H}_{\text{tood}}^{(m,l)}$, $D_{XY}'^n \subset D_{XY}^n \in \mathcal{D}_{XY}$ is the distribution that for any dataset $\mathcal{X} \sim D_{X_I Y_I}'^n$, OOD detection is learnable, where $D_{XY}^n$ is any distribution in $\mathcal{D}_{XY}$ with data amount $n$. The probability of the OOD detection learnability is defined by*

$$\mathbf{P} := \varliminf_{D_{XY}^n \in \mathcal{D}_{XY}} \varlimsup_{D_{XY}'^n \subset D_{XY}^n} \frac{\mu(D_{XY}'^n)}{\mu(D_{XY}^n)}, \tag{23}$$

*where $\mu$ is the Lebesgue measure in $\mathbb{R}^d$ and $n \in \mathbb{N}^*$.*

Then the main Theorem 5 of the Jackson-type approximation is formally depicted:

**Theorem 5.** *Given the condition $l(\mathbf{y}_2, \mathbf{y}_1) \leq l(K+1, \mathbf{y}_1)$, for any in-distribution labels $\mathbf{y}_1, \mathbf{y}_2 \in \mathcal{Y}$, $|\mathcal{X}| = n < +\infty$ and $\tau > K + 1$, and set $l = 2$ and $m = (2m_h + 1, 1, m_h, 2\tau\hat{d}_0 + 1, r)$. Then in $\mathcal{H}_{\text{tood}}^{(m,l)}$ restricted to maximum value classifier $c$, $\mathbf{P} \geq (1 - \frac{\eta}{|\mathcal{I}|\lambda_0}\tau^2 C_0(r_i)(\frac{C_1^{(\alpha)}(r_i)}{m_h^{2\alpha-1}} + \frac{C_2^{(\beta)}(r_i)}{r^\beta}(km_h)^\beta))^{(K+1)^{n+1}}$, and in $\mathcal{H}_{\text{tood}}^{(m,l)}$ restricted to score-based classifier $c$, $\mathbf{P} \geq (1 - \frac{\eta}{|\mathcal{I}|\lambda_0}\tau^2 C_0(r_i)(\frac{C_1^{(\alpha)}(r_i)}{m_h^{2\alpha-1}} + \frac{C_2^{(\beta)}(r_i)}{r^\beta}(km_h)^\beta))^{(K+1)^{n+1}+1}$, for any fixed $\lambda_0 > 0$ and $r_i$ defined in Lemma 5, if $\{\mathbf{v} \in \mathbb{R}^{K+1} : E(\mathbf{v}) \geq \lambda\}$ and $\{\mathbf{v} \in \mathbb{R}^{K+1} : E(\mathbf{v}) < \lambda\}$ both contain an open ball with the radius $R$, where $R > ||W_4||_2|\mathcal{I}|(\tau^2 C_0(\phi)(\frac{C_1^{(\alpha)}(\phi)}{m_h^{2\alpha-1}} + \frac{C_2^{(\beta)}(\phi)}{r^\beta}(km_h)^\beta) + \lambda_0)$, $\phi$ defined in Lemma 6 and $W_4$ is determined by $\phi$.*

The proof structure leverages the Jackson-type approximation of transformers, as detailed in Jiang & Li (2023), to fulfill one of the sufficient conditions for OOD detection learnability *i.e.* Theorem 7 in Fang et al. (2022). Notably, the Jackson-type approximation of transformers has a global error bound instead of the uniform convergence in UAP theory. Therefore, Markov's inequality is applied to get probabilistic conclusions regarding Definition 5. This approach establishes a lower bound of error and its convergent rate for OOD detection using transformers. The lower bound is not the infimum because the Jackson-type approximation is sufficient but not necessary. The specific proof and discussion about the convergent rate and insights into transformers are organized in Appendix E.

## C.2 OOD DETECTION IN OTHER PRIORI-UNKNOWN SPACES

In **the single-distribution space** $\mathcal{D}_{XY}^{D_{XY}}$, **the Finite-ID-distribution space** $\mathcal{D}_{XY}^{F}$, and **the density-based space** $\mathcal{D}_{XY}^{\mu,b}$, if there exists an overlap between ID and OOD, OOD detection is NOT learnable, which has been discussed in Fang et al. (2022); otherwise, $\mathcal{D}_{XY}^{D_{XY}} \subset \mathcal{D}_{XY}^{s}$, this situation is analyzed in the previous Appendix C.1. Additionally, in **the density-based space** $\mathcal{D}_{XY}^{\mu,b}$, Theorem 9 and Theorem 11 in Fang et al. (2022) are still established in the hypothesis space $\mathcal{H}_{\text{tood}}$, as the proof of these two theorems only need to check the finite Natarajan dimension (Shalev-Shwartz & Ben-David, 2014) of the hypothesis space, which is a weaker condition compared with the finite VC dimension.

Theorem 4 demonstrates that in $\mathcal{H}_{\text{tood}}$, models are OOD detection learnable given sufficient parameters, thereby providing a theoretical basis for employing transformers in OOD detection algorithms (Koner et al., 2021; Fort et al., 2021). Nevertheless, training models to reach their optimal state poses significant challenges. To overcome these issues, additional strategies such as incorporating extra data (Fort et al., 2021; Tao et al., 2023) and utilizing various distance metrics (Podolskiy et al., 2021) have been developed. Detailed discussions on Gaussian mixture datasets, which explore the discrepancy between theoretical performance and practical outcomes and suggest ways to bridge this gap, can be found in Appendices F and G.

## D PROOF AND REMARKS OF THEOREM 4

We first propose several Lemmas before proving the Theorem 4.

**Lemma 1.** *The FCNN-based hypothesis space* $\mathcal{H}_{\mathbf{q}}^{\text{Relu}} \subseteq \mathcal{H}_{\text{tood}}^{(m,l)}$, *where* $\mathbf{q} = (l_1, \cdots, l_g), l_1 = d_0, l_g = K+1, l_M = max\{l_1, \cdots, l_g\}, m = (l_M, 1, 1, 1, l_M),$ *and* $l = g-3, g > 2$.

**Proof.** *Given weights* $\mathbf{w}_i \in \mathbb{R}^{l_i \times l_{i-1}}$ *and bias* $\mathbf{b}_i \in \mathbb{R}^{l_i \times 1}$ *and considering* $\mathbf{x} = \mathbf{h}_0 \in \mathbb{R}^{d_0}$ *is a data in the dataset* $\mathcal{X}$, *the i-layer output of FCNN with architecture* $\mathbf{q}$ *can be written as*

$$\mathbf{f}_i(\mathbf{x}) = \text{Relu}(\mathbf{w}_i \mathbf{f}_{i-1}(\mathbf{x}) + \mathbf{b}_i), \tag{24}$$

*and that of the transformer network* $\mathbf{H} = c \circ \text{Block}_l^{(m)} \circ \text{Block}_{l-1}^{(m)} \circ \cdots \circ \text{Block}_1^{(m)} \circ F$ *in the transformer hypothesis space for OOD detection* $\mathcal{H}_{\text{tood}}$ *is depicted by Eq. equation 15 and equation 16. Particularly, set* $W_O^i = \mathbf{0}$, *and* $m = (l_M, 1, 1, 1, l_M)$, *then we get*

$$\mathbf{h}_i = \mathbf{h}_i + W_2 \cdot \text{Relu}(W_1 \cdot \mathbf{h}_{i-1} + \mathbf{b}_1) + \mathbf{b}_2, \tag{25}$$

*where* $\mathbf{h}_i, \mathbf{h}_{i-1}, \mathbf{b}_1 \in \mathcal{R}^{l_M}, W_1, W_2 \in \mathcal{R}^{l_M \times l_M}$. *Since* $\mathbf{H}$ *is composed of* $l$ *blocks and mappings* $c$ *at the bottom and* $F$ *at the top as two layers, a simple case is when* $g = 3$, *it comes that* $l = 0$, $\mathcal{H}_{\text{tood}}^{(m,l)}$ *collapse into* $\mathcal{H}_{\mathbf{q'}}^{\text{Relu}}$, *where* $\mathbf{q'} = (l_1, l_M, l_g), l_M = \max\{l_1, l_g\}$. *So* $\mathcal{H}_{\mathbf{q}}^{\text{Relu}} \subseteq \mathcal{H}_{\mathbf{q'}}^{\text{Relu}}$ *according to Lemma 10 in Fang et al. (2022).*

*When* $g > 3$, *consider* $F(\cdot) : \mathbb{R}^{d_0 \times n} \to \mathbb{R}^{l_M \times n}, F(\mathbf{x}) = \text{Relu}(W\mathbf{x} + \mathbf{b})$ *column-wise and the first layer of the FCNN-based network* $f_1 : \mathbb{R}^{l_1} \to \mathbb{R}^{l_2}, f_1(\mathbf{x}) = \text{Relu}(\omega_1 \mathbf{x} + \beta_1)$. *Since* $l_M = max\{l_1, \cdots, l_g\}, l_2 \le l_M$. *Let*

$$W = [\omega_1, \mathbf{0}]^T, \ \mathbf{b} = [\beta_1, \mathbf{0}]^T, \tag{26}$$

*then* $F(\mathbf{x}) = [f_1(\mathbf{x}), \mathbf{0}]^T$. *Similarly, we compare* $f_i = \text{Relu}(\omega_i f_{i-1}(\mathbf{x}) + \beta_i)$ *and* $\text{Block}_{i-2}$. *Suppose that* $h_{i-3} = [f_{i-1}(\mathbf{x}), \mathbf{0}]^T$, *let* $\mathbf{b}_2 = -h_{i-3}, \mathbf{b}_1 = [\beta_i, \mathbf{0}]^T, W_2 = Id_{l_M \times l_M}$ *and*

$$W_1 = \begin{bmatrix} \omega_i & \mathbf{0} \\ \mathbf{0} & \mathbf{0} \end{bmatrix}, \tag{27}$$

*then it is clear that* $h_{i-2} = [f_i(\mathbf{x}), \mathbf{0}]^T$.

*By mathematical induction, it follows that* $h_{g-3} = [f_{g-1}(\mathbf{x}), \mathbf{0}]^T$ *and* $f(h_{g-3}) = f_g(\mathbf{x})$, $f$ *is defined in Definition 3. Therefore, for any entry* $h_{\mathbf{w},\mathbf{b}} \in \mathcal{H}_{\mathbf{q}}^{\text{Relu}}$, *there exists* $\mathbf{H} \in \mathcal{H}_{\text{tood}}^{(m,l)}$, $m, l$ *defined in the Lemma s.t.* $\mathbf{H} = h_{\mathbf{w},\mathbf{b}}$. $\square$

**Lemma 2.** *For any* $\mathbf{h} \in \mathcal{C}(\mathbb{R}^d, \mathbb{R}^{K+1})$, *and any compact set* $C \in \mathbb{R}^d$, $\epsilon > 0$, *there exists a two layer transformer* $\hat{\mathbf{H}} \in \mathcal{H}_{\text{Trans}}^{(m,2)}$ *and a linear transformation* $\mathbf{f}$ *s.t.* $||\mathbf{f} \circ \hat{\mathbf{H}} - \mathbf{h}||_2 < \epsilon$ *in* $C$, *where* $m = (K+1) \cdot (2\tau(2\tau\hat{d}_0 + 1), 1, 1, \tau(2\tau\hat{d}_0 + 1), 2\tau(2\tau\hat{d}_0 + 1))$.

**Proof.** *Let* $\mathbf{h} = [h_1, \cdots, h_{K+1}]^T$. *Based on the UAP of transformers i.e. Theorem 4.1 in* Jiang & Li (2023), *for any* $\epsilon > 0$, *there exists* $\hat{h}_i = \hat{f}_i \circ \bar{H}_i$, *where* $\hat{f}_i$ *is a linear read out and* $\bar{H}_i \in \mathcal{H}_{\text{Trans}}^{(\hat{m},2)}$, $\hat{m} = 2\tau(2\tau\hat{d}_0 + 1), 1, 1, \tau(2\tau\hat{d}_0 + 1), 2\tau(2\tau\hat{d}_0 + 1)$ *s.t.*

$$\max_{\mathbf{x} \in C} ||\hat{h}_i(\mathbf{x}) - h_i(\mathbf{x})||_1 < \epsilon/\sqrt{K+1}, i = 1, 2, \cdots, K+1. \tag{28}$$

*We need to construct a transformer network* $\hat{\mathbf{H}} \in \mathcal{H}_{\text{Trans}}^{(m,2)}$ *and a linear transformation* $\mathbf{f}$ *s.t.*

$$(\mathbf{f} \circ \hat{\mathbf{H}})_i = \hat{f}_i \circ \bar{H}_i \tag{29}$$

*for all* $i \in \{1, \cdots, K+1\}$. *The following shows the process of construction:*

*Denote the one-layer FCNN in* $\bar{H}_i$ *by* $F_i : \mathcal{R}^{d_0 \times n} \to \mathcal{R}^{D \times n}$, *where* $D = 2n(2nd_0 + 1)$, *the set the one-layer FCNN in* $\hat{\mathbf{H}}$:

$$F : \mathcal{R}^{d_0 \times n} \to \mathcal{R}^{D(K+1) \times n},$$
$$F = [F_1, \cdots, F_{K+1}]^T, \tag{30}$$

*then* $\mathbf{h}_0 = [h_0^1, \cdots, h_0^{K+1}]^T$, *where* $\mathbf{h}_0$ *is the input to transformer blocks in* $\hat{\mathbf{H}}$, *and* $h_0^i$ *is that in* $\bar{H}_i, i = 1, \cdots, K+1$.

*Denote the matrices in* $\bar{H}_i$ *by* $\bar{W}_K^i$, $\bar{W}_Q^i$, $\bar{W}_V^i$ *and* $\bar{W}_O^i$ *since each block only has one head. For the i-th head in each block of transformer network* $\hat{\mathbf{H}}$, *we derive the matrix* $W_k^i \in \mathcal{R}^{(K+1)\hat{m}_h \times (K+1)D}$ *from* $\bar{W}_K^i$ *with* $\hat{m}_h = 1$:

$$W_K^i = \begin{bmatrix} \mathbf{0}_{(i-1)\hat{m}_h \times (i-1)D} & & \\ & \bar{W}_K^i & \\ & & \mathbf{0}_{(K+1-i)\hat{m}_h \times (K+1-i)D} \end{bmatrix}. \tag{31}$$

*Furthermore, we obtain* $W_Q^i$, $W_V^i$ *and* $W_O^i$ *in the same way, then independent operations can be performed on different blocks in the process of computing the matrix* $\text{Att}(\mathbf{h}_0) \in \mathcal{R}^{(K+1)D \times n}$. *So we can finally get the attention matrix in the following form:*

$$\text{Att}(\mathbf{h}_0) = [\text{Att}_1(\mathbf{h}_0), \cdots \text{Att}_{K+1}(\mathbf{h}_0)]^T, \tag{32}$$

*where* $\text{Att}_i(\mathbf{h}_0) \in \mathcal{R}^{D \times n}, i \in \mathcal{Y}_I + 1$ *are attention matrices in* $\bar{H}_i$.

*Similarly, it is easy to select* $W_1, W_2, \mathbf{b}_1, \mathbf{b}_2$ *such that* $\text{FF}(\mathbf{h}_0) = [\text{FF}_1(\mathbf{h}_0), \cdots \text{FF}_{K+1}(\mathbf{h}_0)]^T$, *i.e.* $\mathbf{h}_1 = [h_1^1, \cdots, h_1^{K+1}]^T$, *where the meaning of superscripts resembles to that of* $h_0^i$. *Repeat the process, we found that*

$$\hat{\mathbf{H}}(\mathcal{X}) = [\bar{H}_1(\mathcal{X}), \cdots \bar{H}_{K+1}(\mathcal{X})]^T. \tag{33}$$

*Denote* $\hat{f}_i(\bar{H}_i) = w_i \cdot \bar{H}_i + b_i$, *then it is natural to construct the linear transformation* $\mathbf{f}$ *by:*

$$\mathbf{f}(\hat{\mathbf{H}}) = [w_1, \cdots, w_{K+1}]^T \cdot \hat{\mathbf{H}} + [b_1, \cdots, b_{K+1}]^T, \tag{34}$$

*which satisfies Eq. equation 29.*

*By Eq. equation 28, for any* $\epsilon > 0$, *there exists* $\hat{\mathbf{H}} \in \mathcal{H}_{\text{Trans}}^{(m,2)}$ *and the linear transformation* $\mathbf{f}$ *s.t.*

$$\max_{\mathbf{x} \in C} ||\mathbf{f} \circ \hat{\mathbf{H}} - \mathbf{h}||_2 \leq \sqrt{\sum_{i=1}^{K+1} (\max_{\mathbf{x} \in C} ||\hat{h}_i(\mathbf{x}) - h_i(\mathbf{x})||_1)^2}$$
$$< \sqrt{\sum_{i=1}^{K+1} (\epsilon/\sqrt{K+1})^2} = \epsilon, \tag{35}$$

*where* $m = (K+1) \cdot \hat{m}$.

*We have completed this Proof.* $\qquad\square$

Then we prove the proposed Theorem 4.

**Proof.** *First*, *we prove the sufficiency. By the proposed Lemma 1 and Theorem 10 in* Fang et al. *(2022), the sufficiency of Theorem 4 is obvious.*

***Furthermore***, *according to the Proof of Theorem 10 in* Fang et al. *(2022), to replace the FCNN-based or score-based hypothesis space by the transformer hypothesis space for OOD detection $\mathcal{H}_{\text{tood}}$, the only thing we need to do is to investigate the UAP of transformer networks s.t. the UAP of FCNN network i.e. Lemma 12 in* Fang et al. *(2022) can be replaced by that of transformers. Moreover, it is easy to check Lemmas 13Ĩ6 in* Fang et al. *(2022) still holds for $\mathcal{H}_{\text{tood}}$. So following the Proof of Theorem 10 in* Fang et al. *(2022), by Theorem 3 in* Yun et al. *(2019) and the proposed Lemma 2, we can obtain the needed layers $l$ and specific budget $m$ which meet the conditions of the learnability for OOD detection tasks.*

***Second***, *we prove the necessity. Assume that $|\mathcal{X}| = +\infty$. By Theorems 5, 8 in* Bartlett & Maass *(2003), $VCdim(\Phi \circ \mathcal{H}_{\text{tood}}^{(m,l)}) < +\infty$ for any $m, l$, where $\Phi$ maps ID data to $1$ and maps OOD data to $2$. Additionally, $\sup_{h \in \mathcal{H}_{\text{tood}}^{(m,l)}} |\{\mathbf{x} \in \mathcal{X} : h(\mathbf{x}) \in \mathcal{Y}\}| = +\infty$ given $|\mathcal{X}| = +\infty$ for any $m, l$. By the impossibility Theorem 5 for separate space in* Fang et al. *(2022), OOD detection is NOT learnable for any finite $m$, $l$.* □

**Remark 1.** Yun et al. *(2019) and* Jiang & Li *(2023) provide two perspectives of the capacity of transformer networks. The former gives the learning conditions of OOD detection with limited width (or budget of each block) and any depth of networks, and the letter develops the learning conditions with limited depth.*

**Remark 2.** *Define a partial order for the budget $m$: for $m = (d, h, m_h, m_V, r)$ and $m' = (d', h', m'_h, m'_V, r')$, $m' < m$ if every element in $m'$ is less than the corresponding element in $m$. $m' \leq m$ if if every element in $m'$ is not greater than the corresponding element in $m$. So it easily comes to a corollary: $\forall m'$ satisfies $m \leq m'$ and $l \leq l'$, if transformer hypothesis space $\mathcal{H}_{\text{tood}}^{(m,l)}$ is OOD detection learnable, then $\mathcal{H}_{\text{tood}}^{(m',l')}$ is OOD detection learnable.*

**Remark 3.** *It is notable that when $m = +\infty$ or $l = +\infty$, $VCdim(\Phi \circ \mathcal{H}_{\text{tood}}^{(m,l)})$ may equal to $+\infty$. This suggests the possibility of achieving learnability in OOD detection without the constraint of $|\mathcal{X}| < +\infty$. Although an infinitely capacitated transformer network does not exist in reality, exploring whether the error asymptotically approaches zero as capacity increases remains a valuable theoretical inquiry.*

# E   PROOF AND REMARKS OF THEOREM 5

To derive the Theorem 5, we need to figure out some Lemmas.

**Lemma 3.** *For any $\mathbf{h} \in \widetilde{\mathcal{C}}^{(\alpha,\beta)}$, and any compact set $C \in \mathbb{R}^d$, there exists a two layer transformer $\hat{\mathbf{H}} \in \mathcal{H}_{\text{Trans}}^{(m,2)}$ and a linear read out $\mathbf{c} : \mathbb{R}^{\hat{d} \times \tau} \to \mathbb{R}^{1 \times \tau}$ s.t. the inequality equation 39 is established, where $m = (2m_h + 1, 1, m_h, 2\tau \hat{d}_0 + 1, r)$.*

**Proof.** *According to Theorem 4.2 in* Jiang & Li *(2023), for any $\mathbf{h} \in \widetilde{\mathcal{C}}^{(\alpha,\beta)}$, there exists $\mathbf{H} \in \mathcal{H}_{\text{Trans}}^{(m,2)}$ and a linear read out $\mathbf{c}$ s.t.*

$$\int_{\mathcal{I}} \sum_{t=1}^{\tau} |\mathbf{c} \circ \mathbf{H}_t(\mathbf{x}) - \mathbf{h}_t(\mathbf{x})| d\mathbf{x} \leq \tau^2 C_0(\mathbf{h}) \left( \frac{C_1^{(\alpha)}(\mathbf{h})}{m_h^{2\alpha-1}} + \frac{C_2^{(\beta)}(\mathbf{h})}{m_{\text{FF}}^{\beta}} (m_h)^{\beta} \right). \quad (36)$$

*Based on Chebyshev's Inequality,*

$$P\left( \sum_{t=1}^{\tau} |\mathbf{c} \circ \mathbf{H}_t(\mathbf{x})_i - \mathbf{h}_t(\mathbf{x})_i| / |\mathcal{I}| > \text{RHS in Eq. equation 36} \right) + \lambda_0 \right) \leq \frac{\text{RHS in Eq. equation 36}}{\lambda_0 |\mathcal{I}|} \quad (37)$$

*for any $\lambda_0 > 0$. Additionally,*

$$||\mathbf{c} \circ \mathbf{H}(\mathbf{x}) - \mathbf{h}(\mathbf{x})||_2 = \sqrt{\sum_{t=1}^{\tau} |\mathbf{c} \circ \mathbf{H}_t(\mathbf{x})_i - \mathbf{h}_t(\mathbf{x})_i|^2}$$

$$\leq \sum_{t=1}^{\tau} |\mathbf{c} \circ \mathbf{H}_t(\mathbf{x})_i - \mathbf{h}_t(\mathbf{x})_i|. \quad (38)$$

*So we get*

$$P(||\mathbf{c} \circ \mathbf{H}(\mathbf{x}) - \mathbf{h}(\mathbf{x})||_2 > |\mathcal{I}|(\textit{RHS in Eq. equation } 36 + \lambda_0) \leq \frac{\textit{RHS in Eq. equation } 36}{\lambda_0 |\mathcal{I}|} \quad (39)$$

*where $m_{\mathrm{FF}}$ is usually determined by its number of neurons and layers. As the number of layers in FF is fixed, the budget $m_{\mathrm{FF}}$ and $r$ are proportional:*

$$r = k \cdot m_{\mathrm{FF}}. \quad (40)$$

*So the right side of the equation equation 36 can be written as*

$$\text{RHS} = \tau^2 C_0(\mathbf{h})\left(\frac{C_1^{(\alpha)}(\mathbf{h})}{m_h^{2\alpha-1}} + \frac{C_2^{(\beta)}(\mathbf{h})}{r^\beta}(km_h)^\beta\right). \quad (41)$$

*We have completed this Proof of the Lemma 3.* $\qquad\square$

Given any finite $\delta$ hypothesis functions $h_1, \cdots, h_\delta \in \{\mathcal{X} \to \mathcal{Y}\}$, for each $h_i$, we introduce a correspongding $\mathbf{g}_i$ (defined over $\mathcal{X}$) satisfying that for any $\mathbf{x} \in \mathcal{X}$, $\mathbf{g}_i(\mathbf{x}) = \mathbf{y}_k$ and $W_4 \mathbf{g}_i^T + b_4 = \mathbf{z}_k$ if and only if $h_i(\mathbf{x}) = k$, where $\mathbf{z}_k \in \mathbb{R}^{K+1}$ is the one-hot vector corresponding to the label $k$ with value $N$. Clearly, $\mathbf{g}_i$ is a continuous mapping in $\mathcal{X}$, because $\mathcal{X}$ is a discrete set. Tietze Extension Theorem (Urysohn, 1925) implies that $\mathbf{g}_i$ can be extended to a continuous function in $\mathbb{R}^d$. If $\tau \geq K + 1$, we can find such $\mathbf{g}_i, W_4, b_4$.

**Lemma 4.** *For any introduced $\mathbf{g}_i$ mentioned above, there exists $\hat{\mathbf{g}}_i$ satisfies $\hat{\mathbf{g}}_i \in \widetilde{\mathcal{C}}^{(\alpha,\beta)}$ and $||\hat{\mathbf{g}}_i - \mathbf{g}_i||_2 < \epsilon$.*

**Proof.** *Based on Theorem 7.4 in DeVore et al. (2021), set $G \equiv 0$ and $\rho \equiv 0$, then $\hat{\mathbf{g}}_i \in \widetilde{\mathcal{C}}^{(\alpha,\beta)}$, and there exists a constant $C$, s.t. $||\hat{\mathbf{g}}_i - \mathbf{g}_i||_2 < \frac{C}{(r+1)^\beta}$.*

*Choose $r$ which is great enough, the proof is completed.* $\qquad\square$

**Remark 4.** *Note that we can also prove the same result if $\mathbf{g}_i$ is any continuous function from $\mathbb{R}^{\hat{d}}$ to $\mathbb{R}$ with compact support.*

**Lemma 5.** *Let $|\mathcal{X}| = n < +\infty$, $\tau > K + 1$ and $\sigma$ be the Relu function. Given any finite $\delta$ hypothesis functions $h_1, \cdots, h_\delta \in \{\mathcal{X} \to \{1, \cdots, K+1\}\}$, then for any $m_h, r > 0$, $m = (2m_h + 1, 1, m_h, 2\tau\hat{d}_0 + 1, r)$, $P(h_1, \cdots, h_\delta \in \mathcal{H}_{\mathrm{tood}}^{(m,2)}) \geq (1 - \frac{m\textit{RHS in Eq. equation } 36}{|\mathcal{I}|\lambda_0})^{(K+1)\delta}$ for any $\eta > 1$.*

**Proof.** *Since $\mathcal{X}$ is a compact set, then Lemma 4 implies that there exists $\hat{\mathbf{g}}_i \in \widetilde{\mathcal{C}}^{(\alpha,\beta)}$ s.t.*

$$||\mathbf{g}_i - \hat{\mathbf{g}}_i||_2 < \epsilon/||W_4||_2. \quad (42)$$

*Denote $r_i = W_4 \mathbf{g}_i^T + b_4$ and $\hat{r}_i = W_4 \hat{\mathbf{g}}_i^T + b_4$, So we get*

$$||r_i - \hat{r}_i||_2 = ||W_4(\mathbf{g}_i - \hat{\mathbf{g}}_i)^T||_2 \leq \epsilon. \quad (43)$$

*Then by Lemma 3, there exists $\hat{\mathbf{H}} \in \mathcal{H}_{\mathrm{Trans}}^{(m,2)}$ and a linear read out $\mathbf{c}$ s.t.*

$$P(||\mathbf{c} \circ \mathbf{H}(\mathbf{x}) - \mathbf{h}(\mathbf{x})||_2 \leq |\mathcal{I}|(\textit{RHS in Eq. equation } 36 + \lambda_0) \geq 1 - \frac{\textit{RHS in Eq. equation } 36}{\lambda_0 |\mathcal{I}|}. \quad (44)$$

*Thus we get if $h_i(\mathbf{x}) = k$, which is equal to $\mathbf{g}_i(\mathbf{x}) = \mathbf{y}_k$ or $r_i(\mathbf{x}) = \mathbf{z}_k$:*

*Firstly, denote $\mathbf{f} = W_4 \mathbf{c} \circ \mathbf{H}^T + b_4$, and let $\mathbf{h} = \hat{\mathbf{g}}_i$, then*

$$P(||\mathbf{f}(\mathbf{x}) - \hat{r}_i(\mathbf{x})||_2 \leq ||W_4||_2 |\mathcal{I}|(\textit{RHS in Eq. equation } 36 + \lambda_0) \geq 1 - \frac{\textit{RHS in Eq. equation } 36}{\lambda_0 |\mathcal{I}|}. \quad (45)$$

*So we obtain that*

$$
\begin{aligned}
P(|\mathbf{f}_k - N| &\le ||W_4||_2 |\mathcal{I}| (RHS \text{ in Eq. equation } 36 + \lambda_0) \\
&\ge P(|\mathbf{f}_k - \hat{r}_{i,k}| + |\hat{r}_{i,k} - r_{i,k}| \le ||W_4||_2 |\mathcal{I}| (RHS \text{ in Eq. equation } 36 + \lambda_0)) \\
&\ge P(||\mathbf{f} - \hat{r}_i||_2 + ||\hat{r}_i - r_i||_2 \le ||W_4||_2 |\mathcal{I}| (RHS \text{ in Eq. equation } 36 + \lambda_0)) \\
&\ge P(||\mathbf{f} - \hat{r}_i||_2 + \epsilon \le ||W_4||_2 |\mathcal{I}| (RHS \text{ in Eq. equation } 36 + \lambda_0)) \\
&= P(||\mathbf{f} - \hat{r}_i||_2 \le ||W_4||_2 |\mathcal{I}| (RHS \text{ in Eq. equation } 36 + (\lambda_0 - \frac{\epsilon}{|\mathcal{I}|}))) \\
&\ge 1 - \frac{RHS \text{ in Eq. equation } 36}{|\mathcal{I}|(\lambda_0 - \frac{\epsilon}{|\mathcal{I}|})} \\
&= 1 - \frac{RHS \text{ in Eq. equation } 36}{|\mathcal{I}|\lambda_0 - \epsilon}.
\end{aligned}
\tag{46}
$$

*Similarly, for any $j \ne k$, we can also obtain that*

$$
P(|\mathbf{f}_k| \le ||W_4||_2 |\mathcal{I}| (RHS \text{ in Eq. equation } 36 + \lambda_0) \ge 1 - \frac{RHS \text{ in Eq. equation } 36}{|\mathcal{I}|\lambda_0 - \epsilon}.
\tag{47}
$$

*Therefore, $P(\arg\max_{k \in \mathcal{Y}} \mathbf{f}_k(\mathbf{x}) = h_i(\mathbf{x})) \ge (1 - \frac{\eta RHS \text{ in Eq. equation } 36}{|\mathcal{I}|\lambda_0})^{K+1}$ for any $\mathbf{x}$, if*

$$
N > 2||W_4||_2 |\mathcal{I}| (RHS \text{ in Eq. equation } 36 + \lambda_0)
\tag{48}
$$

*for any $\eta > 1$, i.e.*

$$
P(h_1, \cdots, h_\delta \in \mathcal{H}_{\text{tood}}^{(m,2)}) \ge (1 - \frac{\eta RHS \text{ in Eq. equation } 36}{|\mathcal{I}|\lambda_0})^{(K+1)\delta},
\tag{49}
$$

*if*

$$
N > 2||W_4||_2 |\mathcal{I}| (RHS \text{ in Eq. equation } 36 + \lambda_0)
\tag{50}
$$

*for any $\eta > 1$. Since $N$ is arbitrary, we can find such $N$.* □

**Lemma 6.** *Let the activation function $\sigma$ be the Relu function. Suppose that $|\mathcal{X}| < +\infty$, and $\tau > K + 1$. If $\{\mathbf{v} \in \mathbb{R}^{K+1} : E(\mathbf{v}) \ge \lambda\}$ and $\{\mathbf{v} \in \mathbb{R}^{K+1} : E(\mathbf{v}) < \lambda\}$ both contain an open ball with the radius $R > ||W_4||_2 |\mathcal{I}| (RHS \text{ in Eq. equation } 36(\phi) + \lambda_0)$, the probability of introduced binary classifier hypothesis space $\mathcal{H}_{\text{tood},E}^{(m,2),\lambda}$ consisting of all binary classifiers $P > (1 - \frac{\eta RHS \text{ in Eq. equation } 36}{|\mathcal{I}|\lambda_0})^{(K+1)\delta+1}$, where $m = (2m_h + 1, 1, m_h, 2\tau \hat{d}_0 + 1, r)$ and $\phi(\mathbf{x})$ is determined by centers of balls, specifically defined in the proof and $W_4$ is determined by $\phi(\mathbf{x})$.*

**Proof.** *Since $\{\mathbf{v} \in \mathbb{R}^{K+1} : E(\mathbf{v}) \ge \lambda\}$ and $\{\mathbf{v} \in \mathbb{R}^{K+1} : E(\mathbf{v}) < \lambda\}$ both contain an open ball with the radius $R \ge ||W_4||_2 |\mathcal{I}| (RHS \text{ in Eq. equation } 36 + \lambda_0)$, we can find $\mathbf{v}_1 \in \{\mathbf{v} \in \mathbb{R}^{K+1} : E(\mathbf{v}) \ge \lambda\}$, $\mathbf{v}_2 \in \{\mathbf{v} \in \mathbb{R}^{K+1} : E(\mathbf{v}) < \lambda\}$ s.t. $B_R(\mathbf{v}_1) \subset \{\mathbf{v} \in \mathbb{R}^{K+1} : E(\mathbf{v}) \ge \lambda\}$ and $B_R(\mathbf{v}_2) \subset \{\mathbf{v} \in \mathbb{R}^{K+1} : E(\mathbf{v}) < \lambda\}$, where $B_R(\mathbf{v}_1) := \{\mathbf{v} : ||\mathbf{v} - \mathbf{v}_1||_2 < R\}$ and $B_R(\mathbf{v}_2) := \{\mathbf{v} : ||\mathbf{v} - \mathbf{v}_2||_2 < R\}$.*

*For any binary classifier $h$ over $\mathcal{X}$, we can induce a vector-valued function as follows. For any $\mathbf{x} \in \mathcal{X}$,*

$$
\phi(\mathbf{x}) = \begin{cases} \mathbf{v}_1, & \text{if } h(\mathbf{x}) = 1, \\ \mathbf{v}_2, & \text{if } h(\mathbf{x}) = 2. \end{cases}
\tag{51}
$$

*Since $\mathcal{X}$ is a finite set, the Tietze Extension Theorem implies that $\phi$ can be extended to a continuous function in $\mathbb{R}^d$. Since $\mathcal{X}$ is a compact set, then Lemma 3 and Lemma 4 implies that there exists a two layer transformer $\mathbf{H} \in \mathcal{H}_{\text{Trans}}^{(m,2)}$ and $f$ defined in 3 s.t for any $\eta > 1$,*

$$
P(||\mathbf{f} \circ \mathbf{H}(\mathbf{x}) - \phi(\mathbf{x})||_2 \le ||W_4||_2 |\mathcal{I}| (RHS \text{ in Eq. equation } 36 + \lambda_0) \ge 1 - \frac{RHS \text{ in Eq. equation } 36}{|\mathcal{I}|\lambda_0 - \epsilon}
\tag{52}
$$

*Therefore, for any $\mathbf{x} \in \mathcal{X}$, it is easy to check that $E(\mathbf{f} \circ \mathbf{H}(\mathbf{x})) \ge \lambda$ if and only if $h(\mathbf{x}) = 1$, and $E(\mathbf{f} \circ \mathbf{H}(\mathbf{x})) < \lambda$ if and only if $h(\mathbf{x}) = 2$ if the condition in $P(\cdot)$ is established.*

*Since $|X| < +\infty$, only finite binary classifiers are defined over $\mathcal{X}$. By Lemma 5, we get*

$$P(\mathcal{H}_{\text{all}}^b = \mathcal{H}_{\text{tood},E}^{(m,2),\lambda}) \geq (1 - \frac{\eta RHS \text{ in Eq. equation } 36}{|\mathcal{I}|\lambda_0})^{(K+1)\delta+1} \qquad (53)$$

*The proof is completed.* □

Now we prove one of the main conclusions *i.e.* Theorem 5, which provides a sufficient Jackson-type condition for learning of OOD detection in $\mathcal{H}_{\text{tood}}$.

**Proof.** *First, we consider the case that $c$ is a maximum value classifier. Since $|\mathcal{X}| < +\infty$, it is clear that $|\mathcal{H}_{\text{all}}| < +\infty$, where $\mathcal{H}_{\text{all}}$ consists of all hypothesis functions from $\mathcal{X}$ to $\mathcal{Y}$. For $|\mathcal{X}| < +\infty$ and $\tau > K+1$, according to Lemma 5, $P(\mathcal{H}_{\text{all}} \subset \mathcal{H}_{\text{tood}}^{(m,2)}) \geq (1 - \frac{\eta RHS \text{ in Eq. equation } 36}{|\mathcal{I}|\lambda_0})^{(K+1)\delta}$ for any $\eta > 1$, where $m = (2m_h + 1, 1, m_h, 2nd + 1, r)$ and $\delta = (K+1)^n$.*

*Consistent with the proof of Lemma 13 in Fang et al. (2022), we can prove the correspondence Lemma 13 in the transformer hypothesis space for OOD detection if $\mathcal{H}_{\text{all}} \subset \mathcal{H}_{\text{tood}}^{(m,2)}$, which implies that there exist $\mathcal{H}^{\text{in}}$ and $\mathcal{H}^b$ s.t. $\mathcal{H}_{\text{tood}}^{(m,2)} \subset \mathcal{H}^{\text{in}} \circ \mathcal{H}^b$, where $\mathcal{H}^{\text{in}}$ is for ID classification and $\mathcal{H}^b$ for ID-OOD binary classification. So it follows that $\mathcal{H}_{\text{all}} = \mathcal{H}_{\text{tood}}^{(m,2)} = \mathcal{H}^{\text{in}} \circ \mathcal{H}^b$. Therefore, $\mathcal{H}_b$ contains all binary classifiers from $\mathcal{X}$ to $\{1, 2\}$. According to Theorem 7 in (Fang et al., 2022), OOD detection is learnable in $\mathcal{D}_{XY}^s$ for $\mathcal{H}_{\text{tood}}^{(m,2)}$.*

***Second, we consider the case that $c$ is a score-based classifier.*** *It is easy to figure out the probability of which OOD detection is learnable based on Lemma 6 and Theorem 7 in Fang et al. (2022).*

*The proof of Theorem 5 is completed.* □

**Remark 5.** *Approximation of $\alpha$: First of all, it is definitely that $\alpha > \frac{1}{2}$ to maintain the conditions in Theorem 4.2 of Jiang & Li (2023). Then, analyze the process of our proof, because of the powerful expressivity of* Relu*, we only need $G \equiv 0$ to bridge from $\mathcal{C}$ to $\widetilde{\mathcal{C}}^{(\alpha,\beta)}$. So with regard to $\mathcal{H}_{\text{tood}}$, any $\alpha > \frac{1}{2}$ satisfies all conditions. But $C_1^\alpha$ can increase dramatically when $\alpha$ get greater.*

**Remark 6.** *Approximation of $\beta$: We denote $\beta \in (0, \beta_{\max}]$. According to Theorem 7.4 in DeVore et al. (2021), $\beta_{\max} \in [1, 2]$.*

**Remark 7.** *By the approximation of $\alpha$ and $\beta$, we discuss the trade-off of expressivity and the budget of transformer models. Firstly, the learnability probability $P \to 1$ if and only if $m_h \to +\infty$ and $\frac{r}{m_h} \to +\infty$. For a fixed $r$, there exists a $m_h$ which achieves the best trade-off. For a fixed $m_h$, the greater $r$ is, the more powerful the expressivity of transformer models is.*

**Remark 8.** *Different scoring functions $E$ have different ranges. For example, $\max_{k \in \{1, \cdots K\}} \frac{e^{v^k}}{\sum_{c=1}^{K+1} e^{v^c}}$ and $T \log \sum_{c=1}^K e^{(\frac{v^c}{T})}$ have ranges contain $(\frac{1}{K+1}, 1)$ and $(0, +\infty)$, respectively. The Theorem 5 gives the insight that the domain and range of scoring functions should be considered when dealing with OOD detection tasks using transformers.*

**Remark 9.** *It can be seen from Theorem 5 that the complexity of the data increases, and the scale of the model must also increase accordingly to ensure the same generalization performance from the perspective of OOD detection. Increasing the category $K$ of data may exponentially reduce the learnable probability of OOD detection, while increasing the amount of data $n$ reduces the learnable probability much more dramatically. Using Taylor expansion for estimation,*

$$(1 - \frac{\eta}{|\mathcal{I}|\lambda_0}\tau^2 C_0(r_i)(\frac{C_1^{(\alpha)}(r_i)}{m_h^{2\alpha-1}} + \frac{C_2^{(\beta)}(r_i)}{r^\beta}(km_h)^\beta))^{(K+1)^{n+1}}$$

$$= 1 - (K+1)^{n+1}\frac{\eta}{|\mathcal{I}|\lambda_0}\tau^2 C_0(r_i)(\frac{C_1^{(\alpha)}(r_i)}{m_h^{2\alpha-1}} + \frac{C_2^{(\beta)}(r_i)}{r^\beta}(km_h)^\beta) \qquad (54)$$

$$+ \mathcal{O}((\frac{\eta}{|\mathcal{I}|\lambda_0}\tau^2 C_0(r_i)(\frac{C_1^{(\alpha)}(r_i)}{m_h^{2\alpha-1}} + \frac{C_2^{(\beta)}(r_i)}{r^\beta}(km_h)^\beta))^2)$$

*for any $\frac{\eta}{|\mathcal{I}|\lambda_0}\tau^2 C_0(r_i)(\frac{C_1^{(\alpha)}(r_i)}{m_h^{2\alpha-1}} + \frac{C_2^{(\beta)}(r_i)}{r^\beta}(km_h)^\beta) < 1$. To ensure generalization, increasing the data category $K$ requires a polynomial increase of model parameters; while increasing the amount*

*of data $n$ requires an exponential increase of model parameters. The data with positional coding $\mathcal{X}$ is contained in $\mathcal{I}$. The greater $\mathcal{I}$ is, the more possibility transformers have of OOD detection learnability. Nevertheless, the scoring function needs to meet a stronger condition of $R$. Theorem 5 indicates that large models are guaranteed to gain superior generalization performance.*

**Remark 10.** *This theorem has limitations for not determining the exact optimal convergence order and the infimum of the error. More research on function approximation theory would be helpful to develop it in-depth.*

# F  THE GAP BETWEEN THEORETICAL EXISTENCE AND TRAINING OOD DETECTION LEARNABLE MODELS

We first show the key problems that intrigue the gap by conducting experiments on generated datasets. The specific experiments are described as follows.

## F.1  BASIC DATASET GENERATION

We generated Gaussian mixture datasets consisting of two-dimensional Gaussian distributions. The expectations $\mu^i$ and the covariance matrices $\Sigma^i$ are randomly generated respectively, $i = 1, 2$ *i.e.* $K = 2$:

$$
\begin{aligned}
\mu^i &= \frac{i}{10}[|\mathcal{N}(0,1)|, |\mathcal{N}(0,1)|]^T, \\
\Sigma^i &= \begin{bmatrix} \sigma_1^i & 0 \\ 0 & \sigma_2^i \end{bmatrix}, \text{ where } \sigma_j^i = \frac{i}{10}|\mathcal{N}(0,1)| + 0.1, \ j = 1, 2,
\end{aligned}
\tag{55}
$$

and the data whose Euclidean distance from the expectation is greater than $3\sigma$ is filtered to construct the separate space. Further, we generated another two-dimensional Gaussian distribution dataset, and also performed outlier filtering operations as OOD data with the expectation $\mu^O$ and the covariance matrix $\Sigma^O$ as

$$
\begin{aligned}
\mu^O &= \frac{1}{2}[-|\mathcal{N}(0,1)|, -|\mathcal{N}(0,1)|]^T, \\
\Sigma^O &= \begin{bmatrix} \sigma_1^O & 0 \\ 0 & \sigma_2^O \end{bmatrix}, \text{ where } \sigma_j^O = 0.2|\mathcal{N}(0,1)| + 0.1.
\end{aligned}
\tag{56}
$$

Formally, the distribution of the generated dataset can be depicted by

$$
D_X = \frac{1}{3}(\mathcal{N}(\mu^1, \Sigma^1) + \mathcal{N}(\mu^2, \Sigma^2) + \mathcal{N}(\mu^O, \Sigma^O))
\tag{57}
$$

as the quantity of each type of data is almost the same. A visualization of the dataset with a fixed random seed is shown in Figure 3(a).

## F.2  MODEL CONSTRUCTION AND GAP ILLUSTRATION

We constructed the transformer models strictly following the hypothesis space definition 4, where $\hat{d}_0 = \hat{d} = 2$ and $\tau = 1$. Our experimental results are shown in Figure 5(b). According to Theorem 4, in $\mathcal{H}_{\text{tood}}^{(m,l)}$, where $m = (2, 2, 1, 1, 4)$ and $l$ is sufficiently large, or $l = 2$, $m = (2w, 1, 1, w, 2w)$, where $w := \tau(2\tau\hat{d}_0 + 1) = 15$, OOD detection can be learned. Since Theorem 4 does not give a specific value for $l$, so we choose a wide range of $l$ for experiments. Figure 5(b) shows that even for a very simple Gaussian mixture distribution dataset, transformer models without additional algorithm design can classify ID data with high accuracy in most cases, but can not correctly classify OOD data, showing severe overfitting and strong bias to classify OOD data into ID categories. By chance, transformers with some $l$ just converge to a learnable state. We have also selected the scoring function $E(f(\mathbf{h}_l)) = max_{k \in \{1, \cdots K\}} \frac{e^{f(\mathbf{h}_l)^k}}{\sum_{c=1}^{K+1} e^{f(\mathbf{h}_l)^c}}$ and visualized the scoring function values for every category by the trained models. It can be seen that in a model that cannot identify OOD data, using the score-based classifier $c$ also can not distinguish the OOD data.

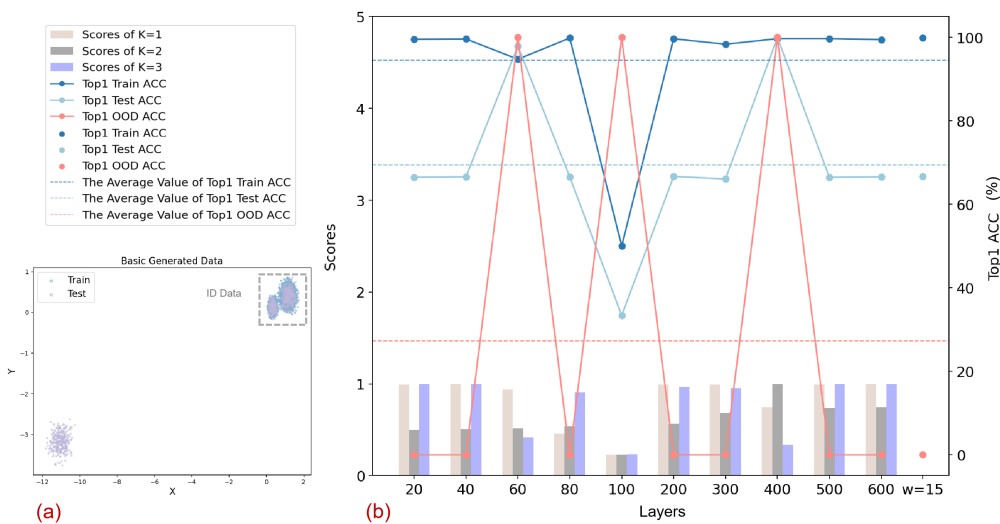

(a)          (b)

Figure 3: (a) The visualization of the generated two-dimensional Gaussian mixture dataset. (b) Curves show the classification accuracy and OOD detection accuracy of the training stage and test stage with different model budgets. And likelihood score bars demonstrate that the model with the theoretical support is disabled to learn OOD characters, leading to the failure of OOD detection.

## G   DETAILS OF OPTIMIZATION AND VALIDATION ON GENERATED DATASETS

In this section, we analyze the causes of training failures and introduce an algorithm designed to address these challenges. We used five different random seeds for data generation for each dataset type discussed later. The experimental outcomes are illustrated in Figure 4.

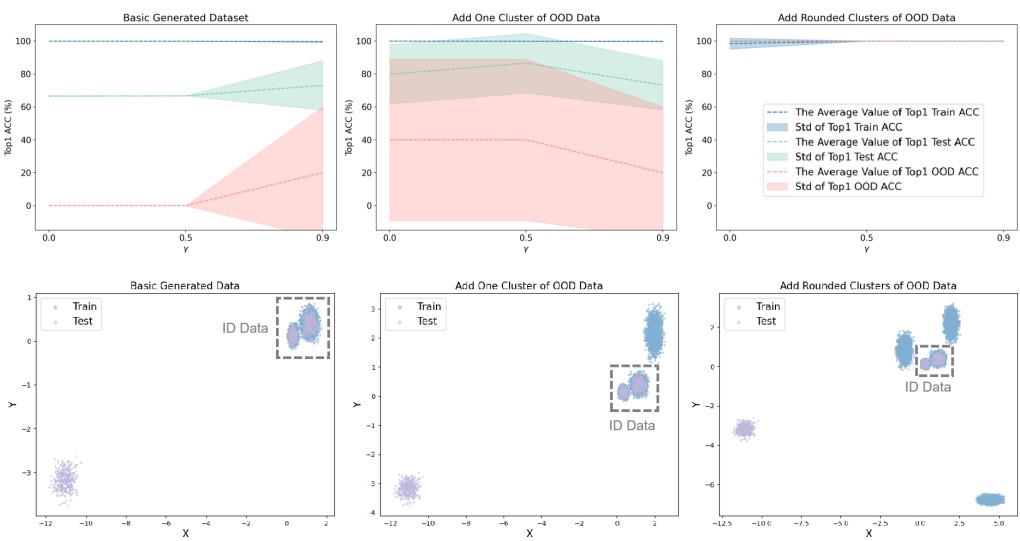

Figure 4: The classification and OOD detection results in the optimization process. The first row of subfigures is the results of experiments under different OOD data distributions. The scAtter plots below show the corresponding training and test set data. The trade-off of loss function $\mathcal{L}$ is shown when picking different $\gamma$, and the power of adding rounded OOD data is illustrated with perfect performance in the third column.

### G.1 OPTIMIZATION 1

First of all, considering that the classical cross-entropy loss $\mathcal{L}_1$ does not satisfy the condition $l(\mathbf{y}_2, \mathbf{y}_1) \leq l(K + 1, \mathbf{y}_1)$, for any in-distribution labels $\mathbf{y}_1, \mathbf{y}_2 \in \mathcal{Y}$, and there is no instruction for model to learn recognizing OOD data, an additional loss $\mathcal{L}_2$ is added in the loss function:

$$\mathcal{L} = (1 - \gamma)\mathcal{L}_1 + \gamma\mathcal{L}_2, \tag{58}$$

$$\mathcal{L}_1(\mathbf{y}, \mathbf{x}) = -\mathbb{E}_{\mathbf{x} \in \mathcal{X}} \sum_{j=1}^{K+1} \mathbf{y}_j \log(\text{softmax}(\mathbf{f} \circ \mathbf{H}(\mathbf{x}))_j), \tag{59}$$

$$\mathcal{L}_2(\mathbf{y}, \mathbf{x}) = -\mathbb{E}_{\mathbf{x} \in \mathcal{X}} \sum_{j=1}^{2} \hat{\phi}(\mathbf{y})_j \log(\hat{\phi}(\text{softmax}(\mathbf{f} \circ \mathbf{H}(\mathbf{x})))_j), \tag{60}$$

where $\mathbf{H} \in \mathcal{H}_{\text{Trans}}$, $\mathbf{y}$ is the one-hot label vector, $\hat{\phi} : \mathbb{R}^{K+1} \to \mathbb{R}^2$ is depicted as follows:

$$\hat{\phi}(\mathbf{y}) = \begin{bmatrix} \sum_{i=1}^{K} \mathbf{y}_i \\ \mathbf{y}_{K+1} \end{bmatrix}. \tag{61}$$

When the condition is satisfied, the classification loss sensitivity of ID data classification decreases, affecting the classification performance of ID data. Therefore, it is qualitatively evident that the value of $\gamma$ has a trade-off between the performance of ID data classification and OOD data recognition.

### G.2 OPTIMIZATION 2

Selecting $\gamma = 0.0, 0.5, 0.9$, we observe a nuanced trade-off illustrated in the basic generated dataset (see the first column of Figure 4), as the model will classify ID data randomly, achieving only 50% accuracy in both training and testing phases, if $\gamma = 1$. Modifying the loss function merely increases the probability that the model can learn from OOD data but does not ensure stable training for achieving high-performance OOD detection. This limitation arises because when the model accurately classifies ID data, the value of $\mathbf{f} \circ \mathbf{H}(\mathbf{x})_{K+1}$ remains small, rendering $\mathcal{L}_2$ almost ineffective during training and impeding the model's ability to distinguish between ID and OOD. Without OOD data in the training set, the model tends to classify all test set data as ID. To address these issues, we explore the generation of virtual OOD data. Our experiments, shown in the middle column of Figure 4, indicate that creating a single cluster of virtual OOD data markedly enhances the OOD detection capabilities of transformers, while also illustrating the trade-offs associated with the parameter $\gamma$ as analysized in Section 3. However, challenges persist in situations where the model correctly classifies ID data but fails to identify OOD data during training. To further enhance performance, we generate three clusters of OOD data surrounding the ID data. As demonstrated in the right column of Figure 4, enriching the content of virtual OOD data enables the model to consistently learn ID classification and extend its generalization to OOD data. Adding rounded clusters of OOD data significantly diminishes the influence of $\mathcal{L}_2$, emphasizing the importance of generating high-quality fake OOD data. Considering the high dimensionality of most datasets and the challenges of delineating high-dimensional ID data boundaries in Euclidean space due to the curse of dimensionality, we retain the binary loss $\mathcal{L}_2$ in our algorithm.

Our experimental results are also consistent with recent research. For example, Fort et al. (2021) shows that incorporating outlier exposure significantly improves the OOD detection performance of transformers, and Tao et al. (2023) has presented a method for synthesizing OOD data using boundary data from KNN clusters.

## H THE PSEUDOCODE OF GROD

The pseudocode of GROD is shown in Alg. 1.

---

**Algorithm 1** GROD

---

**Require:** The training dataset and labels $\mathcal{X}_{\text{train}}$, $\mathcal{Y}$, the testing dataset and labels $\mathcal{X}_{\text{test}}$, $\mathcal{Y}_{\text{test}}$, the learnable parameter $\alpha$, fixed parameters $\gamma$ and the number of each cluster of OOD data $num$, batch size $B$, number of ID classes $K$

**Ensure:** Trained model $M$, classification results $\hat{\mathcal{Y}}_{\text{test}}$ for ID data and OOD detection

{**Fine-tuning Stage**}

**for** ep in training epochs **do**

    **for** each batch $\mathcal{X}$ in $\mathcal{X}_{\text{train}}$ **do**

        $\mathcal{F} \leftarrow \text{NET}(\mathcal{X})$ {Obtain features by Eq. equation 1}

        $\mathcal{F}_{\text{PCA}} \leftarrow \text{PCA}(\mathcal{F}, num)$ {PCA projction}

        $V_{\text{PCA}} \leftarrow \text{Boundary}(\mathcal{F}_{\text{PCA}})$ {Obtain boundary ID data}

        $\mu_{\text{PCA}} \leftarrow \text{MEAN}(\mathcal{F})$ {Obtain centers of ID data}

        $\Sigma_{\text{PCA}} \leftarrow \text{COV}(\mathcal{F})$

        $\text{Dist}_{\text{PCA}}^{ID} \leftarrow \text{MEAN}(\text{DIST}(\mathcal{F}, \mu_{\text{PCA}}, \Sigma_{\text{PCA}}))$ {Obtain average distances of ID by Eq. equation 8 (the former one)}

        $\hat{\mathcal{F}}_{\text{PCA}}^{\text{OOD}} \leftarrow \text{GENERATE}(V_{\text{PCA}}, \mu_{\text{PCA}}, \alpha, num)$ {Generate fake OOD data by Eq. equation 6-Eq. equation 7}

        $\mathcal{F}_{\text{LDA}} := \cup_{i=1}^{K} \mathcal{F}_{\text{LDA},i} \leftarrow \text{LDA}(\mathcal{F}, \mathcal{Y}, num)$ {Generate inter-class fake OOD data and calculate Mahalanobis distances similar to the above process}

        $V_{\text{LDA},i} \leftarrow \text{Boundary}(\mathcal{F}_{\text{LDA},i})$

        **for** $i_k \in I$ **do**

            $\mu_{\text{LDA},i_k} \leftarrow \text{MEAN}(\mathcal{F}|_{\mathbf{y}=i_k})$

            $\Sigma_{\text{LDA},i_k} \leftarrow \text{COV}(\mathcal{F}|_{\mathbf{y}=i_k})$

            $\mu_{\text{LDA},i_k}^{\text{ID}} \leftarrow \text{MEAN}(\text{DIST}(\mathcal{F}|_{\mathbf{y}=i_k}, \mu_{\text{PCA}}, \Sigma_{\text{LDA},i_k}))$

            $\hat{\mathcal{F}}_{\text{LDA},i_k}^{\text{OOD}} \leftarrow \text{GENERATE}(V_{\text{LDA},i_k}, \mu_{\text{LDA},i_k}, \alpha, num)$ {$I$ is derived by Eq. 5}

        **end for**

        {Mahalanobis distance filtering mechanism by Eq. equation 9-Eq. equation 10}

        **if** $|I| > 0$ **then**

            $\hat{\mathcal{F}}^{\text{OOD}} \leftarrow \hat{\mathcal{F}}_{\text{PCA}}^{\text{OOD}} \cup (\cup_{i_k} \hat{\mathcal{F}}_{\text{LDA},i_k}^{\text{OOD}})$

            $\text{Dist}^{\text{OOD}} \leftarrow \min_{i_k} \text{DIST}(\hat{\mathcal{F}}^{\text{OOD}}, \mu_{\text{LDA},i_k}, \Sigma_{\text{LDA},i_k})$

            $I_0 \leftarrow \arg\min_{i_k} \text{DIST}(\hat{\mathcal{F}}^{\text{OOD}}, \mu_{\text{LDA},i_k}, \Sigma_{\text{LDA},i_k})$

            $\Lambda \leftarrow \text{LAMBDA}(\lambda, \hat{\mathcal{F}}^{\text{OOD}}, \text{Dist}^{\text{OOD}}, \text{Dist}_{\text{LDA},I_0}^{ID})$

            $mask = \text{Dist}^{\text{OOD}} \geq (1 + \Lambda)\text{Dist}_{\text{LDA},I_0}^{ID}$

        **else**

            $\hat{\mathcal{F}}^{\text{OOD}} \leftarrow \hat{\mathcal{F}}_{\text{PCA}}^{\text{OOD}}$

            $\text{Dist}^{\text{OOD}} \leftarrow \text{DIST}(\hat{\mathcal{F}}^{\text{OOD}}, \mu_{\text{PCA}}, \Sigma_{\text{PCA}})$

            $\Lambda \leftarrow \text{LAMBDA}(\lambda, \hat{\mathcal{F}}^{\text{OOD}}, \text{Dist}^{\text{OOD}}, \text{Dist}_{\text{PCA}}^{ID})$

            $mask = \text{Dist}^{\text{OOD}} \geq (1 + \Lambda)\text{Dist}_{\text{PCA}}^{ID}$

        **end if**

        $\mathcal{F}^{\text{OOD}} \leftarrow \hat{\mathcal{F}}^{\text{OOD}}[\text{mask}]$

        **if** $|\mathcal{F}^{\text{OOD}}| > B/K + 2$ **then**

            $\mathcal{F}^{\text{OOD}} \leftarrow \mathcal{F}^{\text{OOD}}[\text{random mask}]$ {Random filtering mechanism}

            $\mathcal{F}_{\text{all}} \leftarrow \mathcal{F} \cup \mathcal{F}^{\text{OOD}}$

            $\mathcal{Y}_{\text{all}} \leftarrow \text{STACK}(\mathcal{Y}, (K+1)\mathbf{1}_{|\mathcal{F}^{\text{OOD}}|})$

            $\hat{\mathcal{Y}}_{\text{all}}, \text{LOGITS} \leftarrow \text{CLASSIFIER}(\mathcal{F}_{\text{all}})$

            Iterate the model parameters according to the loss function $\mathcal{L}$ in Eq. equation 58-equation 60.

        **end if**

    **end for**

    Save model $M$ with the best performance.

**end for**

**return** $M$

{**Inference Stage**}

$\mathcal{F}_{\text{test}}, \text{LOGITS}_{\text{test}} \leftarrow M(\mathcal{F}_{\text{test}})$

$\text{LOGITS}_{\text{test}} \leftarrow \text{ADJUST}(\text{LOGITS}_{\text{test}})$ {Adjust LOGITS by Eq. equation 14}

$\hat{\mathcal{Y}}_{\text{test}} \leftarrow \text{PostProcessor}(\mathcal{F}_{\text{test}}, \text{LOGITS}_{\text{test}})$ {Obtain prediction results after post-processing}

**return** $\hat{\mathcal{Y}}_{\text{test}}$

---

# I   IMPLEMENTATION DETAILS

## I.1   SETTINGS FOR THE FINE-TUNING STAGE.

For image classification, we finetune the ViT backbone and GROD model with hyper-parameters as follows: epoch number = 20, batch size = 64, and the default initial learning rate = $1 \times 10^{-4}$. We set parameters $\alpha = 1 \times 10^{-3}$ for PCA and LDA projection and $\alpha = 0.1$ for PCA projection, $num = 1$, and $\gamma = 0.1$. An AdamW (Kingma & Ba, 2014; Loshchilov & Hutter, 2017) optimizer with the weight decay rate $5 \times 10^{-2}$ is used when training with one Intel(R) Xeon(R) Platinum 8352V CPU @ 2.10GHz and one NVIDIA GeForce RTX 4090 GPU with 24GiB memory. For other OOD detection methods, we adopt the same values of common training hyperparameters for fair comparison, and the parameter selection and scanning strategy provided by OpenOOD (Zhang et al., 2023b; Yang et al., 2022a;b; 2021; Bitterwolf et al., 2023) for some special parameters. For text classification, we employ the pre-trained BERT base model and GPT-2 small. We modify the default initial learning rate to $2 \times 10^{-5}$ and the weight decay rate to $1 \times 10^{-3}$ for BERT, and initial learning rate to $5 \times 10^{-5}$ and the weight decay rate to $1 \times 10^{-1}$ for GPT-2, other hyperparameters maintain the same as in image classification tasks.

GROD has three hyperparameters *i.e.* $num$, $\gamma$ and $\alpha$. $num = 1$ is empirically an optimal choice, which is consistent with the conclusion in Fort et al. (2021) that even adding one or two OOD can raise the OOD detection performance of transformers, and is coordinated for the ratio of ID and fake OOD. The ablation results regarding $\gamma$ in Fig. 5 show that $\gamma \in [0.1, 0.3]$ benefits the task performance, which is also in line with the theoretical insights and the classification (learned by $\mathcal{L}_1$) and OOD detection (learned by $\mathcal{L}_2$) goal of the task. As for $\alpha$, empirically, if $LDA$ is used, we recommend $\alpha = 10^{-3}$, otherwise $10^{-1}$ should be taken.

We preserve the finetuned model with the highest ID data classification accuracy on the validation dataset and evaluate its performance with test datasets. The training and validation process is conducted without any OOD exposure.

## I.2   DATASET DETAILS

We provide details of the datasets as follows:

**Image datasets.**

- **CIFAR-10** (Krizhevsky et al., 2009): This dataset contains $60,000$ images of 32x32 pixels each, distributed across 10 diverse categories (airplane, automobile, bird, cat, deer, dog, frog, horse, ship, and truck). Each category includes $6,000$ images, split into $50,000$ for training and $10,000$ for testing. It is a standard benchmark for image classification tasks.

- **CIFAR-100** (Krizhevsky et al., 2009): Building on the structure of **CIFAR-10**, **CIFAR-100** offers greater variety with 100 categories, each containing 600 images. This dataset serves as an extension of **CIFAR-10**, providing a deeper pool of images for more complex machine-learning models.

- **ImageNet-200** (Deng et al., 2009): **ImageNet-200** is images selected from **ImageNet-1k** with 200 categories. The detailed data list is following the OpenOOD benchmark (Zhang et al., 2023b; Yang et al., 2022a;b; 2021; Bitterwolf et al., 2023).

- **Tiny ImageNet** (Le & Yang, 2015): **Tiny ImageNet** comprises $100,000$ images resized to $64 \times 64$ pixels, spread across 200 categories, with each category featuring 500 training samples, and 50 samples each for validation and testing. This dataset offers a broad spectrum of challenges in a format similar to the CIFAR datasets but on a larger scale.

- **SVHN** (Netzer et al., 2011): The Street View House Numbers (**SVHN**) dataset, extracted from Google Street View images, focuses on number recognition with 10 classes corresponding to the digits $0, 1, \cdots, 9$. This dataset is particularly suited for developing machine learning techniques as it simplifies preprocessing steps.

- **Tiny ImageNet-597** (Zhang et al., 2023b; Yang et al., 2022a;b; 2021; Bitterwolf et al., 2023): Firstly filter out many categories from **ImageNet-1K** to avoid overlap with test

OOD data, resulting in 597 categories left. Then apply the same processing as getting **Tiny ImageNet** from **ImageNet** to create this dataset.

- **ImageNet-800**(Zhang et al., 2023b; Yang et al., 2022a;b; 2021; Bitterwolf et al., 2023): The 800-class subset of **ImageNet-1K** that is disjoint with **ImageNet-200**.

**Text datasets.**

- Semantic shift: Following the approach in Podolskiy et al. (2021), we use the CLINC150 dataset (Larson et al., 2019), which consists of phrases used in voice assistants, representing various intents. The OOD data is set to be phrases with unidentified intents, serving as "out-of-scope" inquiries not aligned with any predefined categories. This dataset is ideal for testing the robustness of intent classification systems against unexpected queries and includes both in-scope and out-of-scope data.

- Background shift: We follow (Arora et al., 2021) to choose the long movie review dataset **IMDB** (Maas et al., 2011) as the ID dataset and a business review dataset **Yelp** (Zhang et al., 2015) as the OOD dataset. The **IMDB** dataset consists of $50,000$ movie reviews, tailored for binary sentiment classification to discern positive and negative critiques. The **Yelp** dataset, which includes a variety of business, review, and user data, represents a shift in the background context and is treated as OOD data, providing a different commercial background from the movie reviews of the **IMDB** dataset.

## J  ABLATION STUDY ON HYPER-PARAMETERS.

**Ablation on the loss weight** $\gamma$**.**  Figure 5(a) examines variations in $\gamma$ within the loss function as detailed in Eq. equation 11-equation 13. As outlined in Section 3, changes in $\gamma$ show the trade-off within the loss function $\mathcal{L}$. When the value of $\gamma$ ranges from 0 to 1, the performance under each evaluation metric initially increases and then decreases. When $\gamma = 1$, the model fails to classify ID data. Intriguingly, $\mathcal{L}_2$ and the fake OOD slightly enhance the ID classification performance, surpassing the $10\%$ accuracy threshold of randomness, which explains how GROD simultaneously improves ID data classification and OOD detection performance, as illustrated in Section 4.2. The efficiency of $\mathcal{L}_2$ also indicates that OOD generated by GROD closely mimics OOD from real datasets.

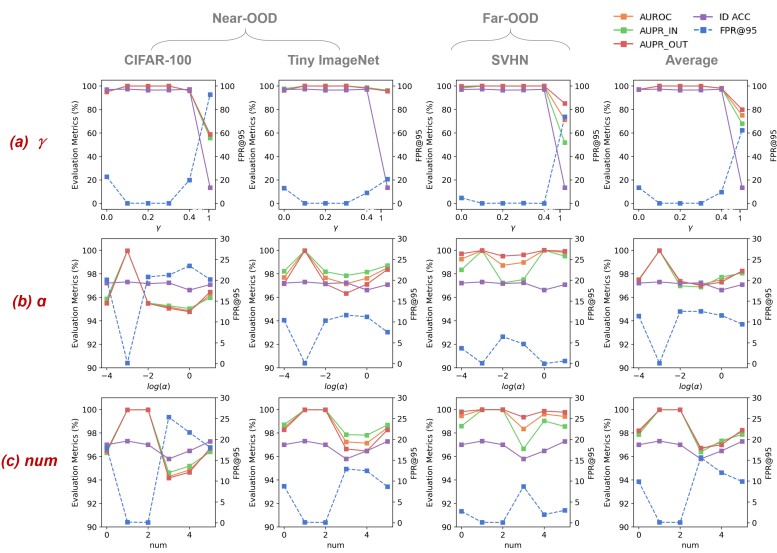

Figure 5: Ablation study on extra hyperparameters in GROD. (a) The weight $\gamma$ in $\mathcal{L}$. (b) The parameter $\alpha$ adjusts the extending distance of generated OOD data. (c) The number of every OOD cluster $num$. The ID dataset is **CIFAR-10** and the backbone is the pre-trained ViT-B-16.

**Ablation on $\alpha$ in adjusting the ID-OOD distance.**    In Figure 5(b), the value of $\alpha$ is adjusted, demonstrating that a larger $\alpha$ increases the Mahalanobis distance between ID and synthetic OOD. Empirical results indicate that an $\alpha$ value of $1 \times 10^{-3}$ achieves optimal performance when using LDA projection. If $\alpha$ is reduced, causing ID and OOD data to be too closely aligned in Mahalanobis distance, the model tends to overfit and fails to discern their differences. Conversely, if $\alpha$ is too high, most inter-class OOD data either become global OOD around ID data or resemble ID from other classes, thus being excluded by the Mahalanobis distance condition in Eq. equation 9. At this time, inter-class OOD is similar to global OOD typically generated only by PCA, leading to a significant drop in near-OOD detection performance, while far-OOD detection remains consistent. The performance curves of near-OOD detection also indicate that if only PCA projections are used, we can set $\alpha$ in a larger value, as the performance increases after dropping from the top.

**Ablation on $num$ in the number of outliers.**    Figure 5(c) explores how the dimension parameter $num$ influences performance. The model demonstrates superior performance when $num$ is set to 1 or 2, as PCA and LDA effectively retain characteristics of the original data and distinguish clusters of each category. Increasing the dimensions of PCA and LDA projections often results in the selection of less representative features in our filtering mechanism. Besides, maintaining $num$ at 1 or 2 usually ensures a balanced ratio of generated OOD data to ID data. Overall, the model consistently delivers competitive outcomes, affirming the efficiency of GROD in various settings.

## K    VISUALIZATION FOR FAKE OOD DATA AND PREDICTION LIKELIHOOD

**Feature visualization.**    As shown in Figure 6, we use the t-SNE dimensionality reduction method to visualize the two-dimensional dataset embeddings in the feature space. All the subfigures are derived from the same fine-tuned ViT-B-16 model.

The ID dataset, the test set of **CIFAR-10**, displays ten distinct clusters after embedding, each clearly separated. Consistent with our inference on GROD, the LDA projection generates fake OOD around each ID data cluster. Despite the high-dimensional feature space where OOD data typically lies outside ID clusters due to GROD's generation and filtering mechanisms, the two-dimensional visualization occasionally shows virtual OOD data within the dense regions of ID. This occurs because the projection from high dimensions to two-dimensional space inevitably results in some loss of feature expression, despite efforts to maintain the integrity of the data distribution.

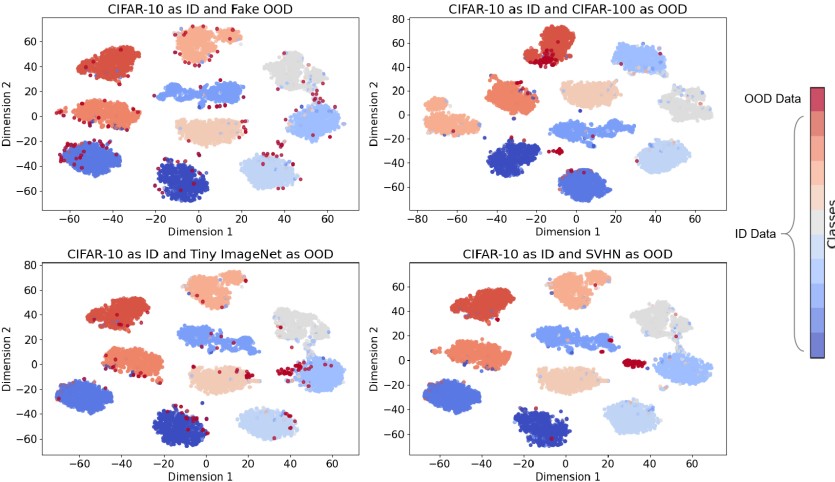

Figure 6: t-SNE visualization of the generated OOD data and test sets in the feature space.

We also visualize real OOD features from near-OOD datasets **CIFAR-100** and **Tiny ImageNet**, and the far-OOD dataset **SVHN**. To distinctively compare the distribution characteristics of fake and real OOD data, we plot an equal number of real and synthetic OOD samples selected randomly. Near-OOD data resembles our synthetic OOD, both exhibiting inter-class surrounding characteristics, while far-OOD data from **SVHN** displays a different pattern, mostly clustering far from the ID

clusters. Although far-OOD data diverges from synthetic OOD data, the latter contains a richer array of OOD features, facilitating easier detection of far-OOD scenarios. Thus, GROD maintains robust performance in detecting far-OOD instances as well. The visualization results in Figure 6 confirm that GROD can generate high-quality fake OOD data effectively, overcoming the limitation discussed in He et al. (2022) that OOD generated by some methods can not represent real outliers.

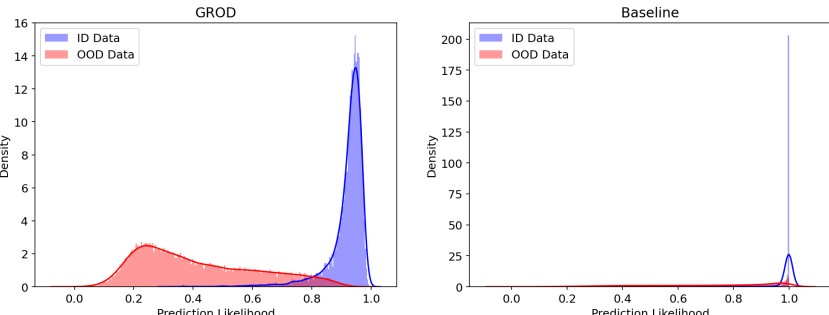

Figure 7: The distribution histograms and probability density curves of prediction likelihoods of ID and OOD test data. Results derived by GROD and the baseline MSP are visualized, with **CIFAR-10** as ID and **SVHN** as OOD.

**Likelihood visualization.** The process of OOD detection and model performance evaluation follows a standardized protocol, where classification predictions and their likelihood scores are generated and subsequently analyzed. The likelihood scores for OOD data are typically lower than those for ID data, as OOD samples do not fit into any ID category, resulting in a bimodal distribution of likelihood scores of all test data. In this distribution, ID and OOD form distinct high-frequency areas, separated by a zone of lower frequency. A broader likelihood range in this low-frequency zone with minimal overlap between the ID and OOD data signifies the model is more effective for OOD detection.

Comparing the likelihood distributions of the baseline MSP model with GROD as shown in Figure 7, it is evident that GROD significantly enhances the distinction in classification likelihood between ID and OOD, thereby improving OOD detection performance. The enhancements are quantitatively supported by the performance metrics reported in Table 2, where GROD surpasses the baseline by $15.30\%$ in FPR@95 and $4.87\%$ in AUROC on datasets **CIFAR-10** and **SVHN**.

