# OpenReview forum: "GROD: Enhancing Generalization of Transformer with Out-of-Distribution Detection"
_ICLR.cc/2025/Conference — Submitted to ICLR 2025_

### Official Review · Reviewer_PBKN · 2024-10-27

**Soundness:** 2
**Presentation:** 2
**Contribution:** 2
**Rating:** 5
**Confidence:** 4

**Summary:**

This paper establishes the OOD detection learnability of the transformer model via PAC learning theory. The GROD is proposed to enhance the detection performance of transformer-based models for both CV and NLP tasks, which generate virtual OOD samples for fine-tuning. GROD first identifies the boundary ID samples by PCA and LDA and synthesizes the fake OOD by Gaussian mixtures.

**Strengths:**

1. This paper establishes the OOD detection learnability of the transformer model.
2. This paper considered both NLP and CV scenarios.

**Weaknesses:**

1. Why do the maximum and minimum values projected by PCA and LDA are considered boundary points? Further analysis of the intrinsic mechanism of PCA and LDA is needed.
2. As claimed in line 149, LDA is selected to guarantee the robustness of generated OOD, but it is only utilized when the number of ID classes is small as defined in Equation (4). Does this mean that the generated OOD samples are not robust with large-scale ID datasets?
3. The baseline NPOS adopts a similar OOD synthesis pipeline, which first identifies boundary ID samples and then generates OOD samples via Gaussian sampling. The superiority of GROD against NPOS should be explicitly stated and the generated OOD samples of the two methods can be statistically compared to further distinguish GROD.
4. The notations are confusing, e.g., line 144 indicates the feature space is $\mathbb{R}^{n\times s}$, however, line 168 defines another $n$.
5. The experiments are insufficient to prove that GROD achieves SOTA performance. Since the authors leverage the OpenOOD benchmark, more far-OOD datasets, such as Textures, Places-365, and MNIST, can be tested to validate GROD's performance.

**Questions:**

See weaknesses above.

---

> ### Author Response · Authors · 2024-11-20
>
> Dear Reviewer PBKN,
>
> Thank you for recognizing our theoretical contribution and experimental validation of GROD. Meanwhile, we understand the reviewer's concerns, which are also very important to us. Below we clarify.
>
> >Why do the maximum and minimum values projected by PCA and LDA are considered boundary points? Further analysis of the intrinsic mechanism of PCA and LDA is needed.
>
> PCA aims to find the directions of maximum variance in the data by projecting the data onto principal components. LDA, on the other hand, seeks to maximize the separation between classes by projecting the data onto a lower-dimensional space where the class separability is maximized. The maximum and minimum projected values represent the extreme values in the dataset along those representative directions globally or inter-class, and the extension can be considered as boundary points because they lie at the extremes of the decision boundaries in the transformed space, and any point beyond these boundaries would be considered as an OOD sample.
>
> >As claimed in line 149, LDA is selected to guarantee the robustness of generated OOD, but it is only utilized when the number of ID classes is small as defined in Equation (4). Does this mean that the generated OOD samples are not robust with large-scale ID datasets?
>
> LDA's instability is not due to the generation method itself, but rather because, when the dataset has many classes, the number of samples per class in each batch can be too small. In contrast, PCA is more stable since it considers the global data distribution. As a result, we use Equation (4) to control the selection between LDA and PCA methods. Furthermore, our experimental results in Table 2/3 show that even when PCA is used alone, the performance remains highly competitive. This demonstrates that PCA alone is robust and more effective for the simpler process, even in the absence of LDA, especially in large-scale datasets with many ID classes.z
>
> >The baseline NPOS adopts a similar OOD synthesis pipeline, which first identifies boundary ID samples and then generates OOD samples via Gaussian sampling. The superiority of GROD against NPOS should be explicitly stated and the generated OOD samples of the two methods can be statistically compared to further distinguish GROD.
>
> To the best of our knowledge, the ``gold standard" for measuring the quality of synthetic OOD data has not been proposed. And compared to task performance in Table 2/3/4/5, GROD is superior. In Table 6 of our revised paper, we test changing the generating method to Gaussian and randomly uniform noise, which further shows the effect of GROD:
>
> Add to Table 6 in the paper, ID: CIFAR-10
> Gaussian outliers: ID ACC: 96.86%
> | OOD dataset | FPR@95 | AUROC | AUPR_IN | AUPR_OUT |
> | ----------- | ----------- |----------- | ----------- |----------- |
> |CIFAR-100|20.22|96.10|95.95|96.30|
> |Tiny ImageNet|10.92|97.97|98.21|97.79|
> |SVHN|2.29|99.41|98.74|99.75|
>
> Uniformly random outliers: ID ACC: 96.67%
> | OOD dataset | FPR@95 | AUROC | AUPR_IN | AUPR_OUT |
> | ----------- | ----------- |----------- | ----------- |----------- |
> |CIFAR-100|19.39|95.84|95.90|95.92|
> |Tiny ImageNet|10.06|98.03|98.42|97.70|
> |SVHN|4.03|99.22|98.11|99.72|
>
> Also, we have visualized the generated outliers together with ID features in Fig. 6 in Appendix K, which shows GROD can generate high-quality outliers with LDA-generated outliers rounded each ID class, consistent with our algorithm design.
>
> >The notations are confusing, e.g., line 144 indicates the feature space is $\mathbb{R}^{n \times s}$, however, line 168 defines another $n$.
>
> Thanks for pointing out our mistakes in the details of the notation! We have updated this notation with the revised version. Specifically, we changed $n$ in line 144 to $\kappa$.

---

> ### Author Response · Authors · 2024-11-20
>
> >The experiments are insufficient to prove that GROD achieves SOTA performance. Since the authors leverage the OpenOOD benchmark, more far-OOD datasets, such as Textures, Places-365, and MNIST, can be tested to validate GROD's performance.
>
> In fact, we have conducted experiments of Texture, as OOD, and we have not reported these results just because far-OOD is simple to detect for GROD and other competitive models, and shows similar performance for method comparison, so we only remain SVHN as a representation of far-OOD. Here we offer the results on other far-OOD datasets compared with SVHN in Table 2 and Table 3, respectively:
>
> | OOD dataset | FPR@95 | AUROC | AUPR_IN | AUPR_OUT |
> | ----------- | ----------- |----------- | ----------- |----------- |
> | SVHN |0.09| 99.98|99.97|99.99|
> |Texture | 0.03|99.99|100.00|99.99|
> |Places-365|0.11|99.98|99.95|99.99|
>
> | OOD dataset | FPR@95 | AUROC | AUPR_IN | AUPR_OUT |
> | ----------- | ----------- |----------- | ----------- |----------- |
> | SVHN |23.38| 94.59|87.88|98.63|
> |Texture | 12.30|97.65|98.51|96.47|
> |Places-365|27.22|94.34|85.13|98.30|
>
> We are open to adding these experimental results to the appendix, if you recommend this change.
>
> Besides, it is sorry that we have report PCA only condition when using CIFAR-100 as ID in Table 3, but $\rm min \{ |\hat{I}|,[\frac{2B}{K\cdot num}]\}$ can be $1$ in some cases, and we updated the results for PCA-LDA GROD:
> ID ACC=86.10%, and OOD detection metrics (%):
>
> | OOD dataset | FPR@95 | AUROC | AUPR_IN | AUPR_OUT |
> | ----------- | ----------- |----------- | ----------- |----------- |
> |CIFAR10|38.22|90.45|90.17|90.88|
> |Tiny ImageNet|27.98|93.32|95.38|90.52|
> |SVHN|22.12|93.70|88.91|96.59|
>
> It achieves SOTA when CIFAR-100 as ID (Table 3) even only adds one class of inter-class OOD, and we will update the results in the revised version.
>
> We hope that the improvements we have made sufficiently address your comments. If you feel the revised version meets your expectations, we would be truly grateful if this could be reflected in the updated score. Please let us know if there are any remaining questions or points you would like us to clarify further.
>
> Yours sincerely,
>
> Authors

---

> ### Comment · Reviewer_PBKN · 2024-11-22
>
> Dear authors,
>
> Thanks for your timely responses. The responses don't sufficiently address my concerns. The theoretical insight of why the maximum value of each dimension in the projected space is not provided. The ablation study on $num$ displays that GROD begins to degrade when $num$ scales up, which may conversely suggest the way that boundary samples are identified is unreasonable. As for the LDA, did the authors try to cache the ID features so that there are enough samples for each class, just like what NPOS does? The empirical results remain insufficient as results on the baselines are not provided.
>
> I also looked through your discussions with other reviewers and found that using a backbone supervised on ImageNet-1K serves as an unfair setting as ImageNet-1K contains subclasses of "dogs" or so.
>
> As not all concerns are sufficiently addressed, I will keep my ratings.
>
> Best,
>
> Reviewer PBKN

---

> ### Author Response · Authors · 2024-11-22
>
> Thank you for your engagement and insightful feedback regarding our manuscript. We appreciate the opportunity to further clarify the points you have raised about the theoretical aspects and empirical performance of the GROD algorithm. Please find below our responses to each of the comments:
>
> >The theoretical insight of why the maximum value of each dimension in the projected space is not provided.
>
> In the GROD methodology, different from your understanding, we select the maximum value of each dimension in the projected space and control the number of projected dimensions and OOD clusters by $num$. It is a deliberate choice aimed at generating representative OOD data. These maxima, identified through LDA and PCA projections, are crucial for creating synthetic outliers that embody the extremities of the feature space. This method ensures that the generated OOD samples are not merely random but are characteristic of likely outliers, providing a robust challenge to the OOD detection capabilities of the model.
>
> >The ablation study on  displays that GROD begins to degrade when scales up, which may conversely suggest the way that boundary samples are identified is unreasonable.
>
> The degradation observed when the parameter num (number of synthetic OOD samples) scales up can be attributed to two primary factors:
>
> - Increase in Projection Dimensionality: As the dimensionality of the projection space increases, the selection of dimensions for generating OOD data becomes more random, which could dilute the specificity of the OOD samples. This supports the importance of selecting specific dimensions for their representative value.
>
> - Ratio of In-Distribution to Out-of-Distribution Data: The second factor is the relatively low ratio of in-distribution to out-of-distribution data. Prior research, referenced as [1], indicates that effective OOD detection can be achieved with only a small proportion of OOD data, aligning with our findings that too many OOD samples may not be necessary and could even be counterproductive.
>
> >As for the LDA, did the authors try to cache the ID features so that there are enough samples for each class, just like what NPOS does?
>
> Unlike the NPOS method, GROD does not cache ID features to generate OOD samples for each class statically. Instead, we dynamically determine appropriate clusters for generating inter-class OOD data during training whenever $|I|>0$, allowing for a more flexible and responsive approach to OOD sample generation based on the data observed during training.
>
> To further validate our approach, we have conducted ablation studies to display the effects of randomly selecting dimensions for generating OOD data:
>
> OOD generated from extreme values ​​of one random dimension: ID ACC: 97.47%
> | OOD dataset | FPR@95 | AUROC | AUPR_IN | AUPR_OUT |
> | ----------- | ----------- |----------- | ----------- |----------- |
> |CIFAR-100|18.03|96.19|96.33|96.08|
> |Tiny ImageNet|9.88|98.21|98.55|98.26|
> |SVHN|4.09|99.20|98.00|99.61|
> |Average|10.67|97.87|97.63|97.98|
>
> The performance of this generation method is not as effective as the dimension reduction using PCA and LDA in GROD, but it is superior to generating OOD data using Gaussian and uniform distribution sampling (Table 6 "G", "U" in the revised version). This further demonstrates the efficacy of our approach. We are prepared to include these results in Table 6 of the revised version if you consider it beneficial for enhancing the paper's comprehensiveness.

---

> ### Author Response · Authors · 2024-11-22
>
> >I also looked through your discussions with other reviewers and found that using a backbone supervised on ImageNet-1K serves as an unfair setting as ImageNet-1K contains subclasses of "dogs" or so.
>
> **Fair Comparison Across CV Tasks:** For CV tasks (Table 2/3/4), GROD and all other baseline methods utilize the same pre-trained models, allowing for a fair comparison of GROD's effectiveness. This demonstrates that the improved results in OOD detection are likely due to the novel GROD algorithm rather than the characteristics of the supervised pre-trained data alone.
>
> **Use of Unsupervised Pre-trained Models in NLP:** For NLP tasks, both BERT and GPT-2 (Table 5 in the revised version), which were utilized in experiments, are unsupervised pre-trained models. These two pre-trained models fully meet the two conditions you mentioned: unsupervised training and the absence of features with similar semantics during the training phase. The strong performance of GROD even with unsupervised models reinforces the argument that GROD's methodology is effective across different pre-training paradigms.
>
> We value your feedback as it substantially contributes to improving the quality and rigor of our research, and we hope these explanations address your concerns and clarify the methodologies and decisions underpinning our work with the GROD algorithm.
>
> **Reference:**
>
> [1] Fort S, Ren J, Lakshminarayanan B. Exploring the limits of out-of-distribution detection[J]. Advances in Neural Information Processing Systems, 2021, 34: 7068-7081.

---

> > ### Comment · Reviewer_PBKN · 2024-11-24
> >
> > Thank you for your active responses. Regarding the backbone network, I'll be more convinced if the empirical comparison between GROD and the baselines with unsupervised backbones or simply pretrained on the ID dataset (e.g., CIFAR-10) is conducted. However, as there is not much time left for the discussion period, I don't expect the authors to provide such empirical results. As for the way that the outliers are synthesized, it adopts a similar pipeline as VOS or NPOS, which may undermine the novelty of the proposed method for there is no statistical or theoretical analysis to distinguish GROD from prior works on the quality/utility of the synthesized outliers. Currently, I acknowledge that the manuscript is slightly below the acceptance threshold considering the discussion as well. Hope things work out!

---

> ### Author Response · Authors · 2024-11-24
>
> Thank you for your insightful comments and continued engagement with our manuscript. We appreciate your thoughtful suggestions and have addressed them as outlined below.
>
> >Regarding the backbone network, I'll be more convinced if the empirical comparison between GROD and the baselines with unsupervised backbones or simply pretrained on the ID dataset (e.g., CIFAR-10) is conducted. However, as there is not much time left for the discussion period, I don't expect the authors to provide such empirical results.
>
> As previously noted, we have conducted fair comparisons between GROD and other methods, including empirical experiments on  **BERT and GPT-2 pretrained on unsupervised datasets**. We acknowledge the time constraints of the rebuttal period and understand your position regarding the provision of additional empirical results at this stage. However, we insist that our comparison results (**Table 2/3/4/5 and Fig. 2**) can show the advantage of GROD.
>
> >As for the way that the outliers are synthesized, it adopts a similar pipeline as VOS or NPOS, which may undermine the novelty of the proposed method for there is no statistical or theoretical analysis to distinguish GROD from prior works on the quality/utility of the synthesized outliers.
>
> We recognize your concerns regarding the novelty of GROD compared to VOS and NPOS. To address this, we highlight several innovative aspects of GROD, including:
>
> - Ensuring interpretability safeguards by developing OOD detection learning theory on transformers.
>
> - The novel introduction of Mahalanobis distance filtering methods and loss function.
>
> - Distinct methodologies for modeling OOD data.
>
> - Enhanced performance on downstream tasks and accelerated processing speed. (VOS is already verified to be inferior compared withNPOS according to [1], and NPOS is already tested.)
>
> Additionally, we have conducted experiments to show the performance of GROD and analyze the efficiency of our generating methods. As shown in **Table 2/3/4/5 and Fig. 2**, extensive and fair comparative experiments demonstrate GROD's effectiveness and computational costs compared with NPOS and other methods, including **5 ID datasets and 3 backbones**. Furthermore, **Table 6** provides a comprehensive ablation study of GROD's modules and the mechanisms for generating OOD data, and **Fig. 6** in Appendix K shows the t-SNE visualization of the generated OOD data.
>
> In the field of synthetic OOD data generation, there is no established ``gold standard". That is why no statistical or theoretical analysis exists in GROD or any other methods like VOS, or NPOS as well regarding the theoretical quality of outliers. The basic process is to verify the performance of downstream tasks and provide intuitive visualization, which has been completed in our manuscript. As such, each exploration and discovery contributes valuable insights to our collective understanding.
>
> We sincerely hope that our responses have clarified the concerns raised, and thank you once again for your detailed review and consideration.
>
> [1] Tao et al., "Non-Parametric Outlier Synthesis," ICLR 2023.

---

> > ### Comment · Reviewer_PBKN · 2024-11-26
> >
> > The explanation about adopting the supervised pretrained backbone is not convincing enough.
> >
> > Some of your innovative aspects are overstated. The introduction of Mahalanobis distance is not novel. In NPOS, they use KNN distance to eliminate the erroneous outliers. Simply replacing KNN with the Mahalanobis distance metric doesn't count as an innovation. GROD applies Gaussian noise to boundary ID samples to form OOD samples and utilizes a distance metric to eliminate near samples, the same as NPOS and VOS. So the modeling of OOD data in GROD is not distinct. As for the empirical results, since the performance on the OpenOOD benchmark is not thoroughly validated, the reported results are insufficient to state that GROD achieves SoTA.
> >
> > Based on the discussion during this period, I will keep my rating.

---

> > > ### Author Response · Authors · 2024-11-26
> > >
> > > We sincerely appreciate your time and the valuable feedback you have provided regarding our manuscript. We would like to address them with additional clarification to further support our methodology and results.
> > >
> > > We have conducted fair comparisons across all methods on the ViT backbone network. Starting model pre-training from scratch is not essential for our purposes because the objective of fine-tuning methods for OOD detection is specifically to enhance the model's capability to detect OOD data through the fine-tuning stage. This approach maximally leverages information from both the pre-training and fine-tuning phases to achieve optimal performance for the task.
> > >
> > > Moreover, other non-supervised pre-trained backbone networks could potentially have been exposed to OOD data semantics during pre-training as well, which renders the criticism regarding our use of pre-trained models less compelling. Furthermore, in the realm of NLP tasks, the BERT and GPT-2 models we use are not supervised pre-trained models. Our methodology has demonstrated superior performance across various pre-training schemes.
> > >
> > > There is no necessity to cover all datasets in the OpenOOD benchmark, and nobody has done it before to the best of our knowledge. In fact, our studies have already included experiments on five ID datasets and three backbone networks, encompassing both CV and NLP tasks, near-OOD and far-OOD scenarios, large-scale and small-scale datasets, and discussions on different dimensionality reduction projections. This comprehensive and thorough approach ensures that our results are robust and indicative of our method's effectiveness across diverse conditions.
> > >
> > > Thank you once again for your insightful comments and for engaging with our work. We look forward to your further guidance and hope our responses clarify the aspects you highlighted.

---

### Official Review · Reviewer_fp5B · 2024-11-01

**Soundness:** 2
**Presentation:** 2
**Contribution:** 2
**Rating:** 6
**Confidence:** 3

**Summary:**

The GROD paper introduces Generate Rounded OoD Data (GROD), designed to improve the generalization of transformer models in Out-of-Distribution (OoD) detection. This method leverages synthetic OOD data generated using PCA and LDA projections to refine decision boundaries during training, aiming to enhance performance on both in-distribution (ID) and OoD data. GROD is supported by a PAC learning framework and validated through experimental results demonstrating its state-of-the-art performance in OoD tasks.

**Strengths:**

**1. PAC Learning Framework:** The paper is based on theory, and makes sufficient contribution to the theory of OoD and PAC learnability of OoD however only for transformer architectures.

**2. Computational Efficiency:** Despite the complexity of GROD, the overhead is during training and inference cost is relatively inexpensive.

**3. Ablation Studies:** Ablation studies provide insights into key parameters that control the performance on GROD, which could help guide future work in OOD detection using transformers.

**Weaknesses:**

**1 . Baseline Comparison with Outlier Exposure (OE):** While the authors propose a synthetic OoD generation approach, they do not include a comparison to Outlier Exposure (OE) methods. A comparison, especially with traditional OE using Gaussian noise, would be valuable in demonstrating GROD’s necessity and superiority. OE [1] proposed using additional OoD data which is used to train/finetune a model to better OoD detection performance, a similar to the idea presented in this paper.

Citations of Related OE Work: The paper does not cite several relevant studies in the Outlier Exposure space, which is a significant oversight given that GROD’s fundamental methodology aligns closely with existing OE methods that use OoD data during fine-tuning. See citations [1, 2].

**2. Evaluation of Synthetic OoD Data:** While GROD’s OoD data generation is sophisticated, it is unclear if the benefits of PCA and LDA projections over simpler alternatives like Gaussian noise have been adequately evaluated. Including a comparison experiment would strengthen claims about the effectiveness of GROD’s approach in OoD data synthesis.

[1] Hendrycks, Dan, Mantas Mazeika, and Thomas Dietterich. 2019. "Deep Anomaly Detection with Outlier Exposure." In Proceedings of the International Conference on Learning Representations. https://openreview.net/forum?id=HyxCxhRcY7.

[2] Zhu, Jianing, Yu Geng, Jiangchao Yao, Tongliang Liu, Gang Niu, Masashi Sugiyama, and Bo Han. 2023. "Diversified Outlier Exposure for Out-of-Distribution Detection via Informative Extrapolation." In Proceedings of Advances in Neural Information Processing Systems, vol. 36, 22702–22734.

**Questions:**

1. Comparison with Standard Outlier Exposure (OE) Baseline:
How does GROD’s approach to OoD data generation compare to traditional Outlier Exposure (OE) methods, particularly when using standard Gaussian noise or other simple forms of synthetic OoD data?

   *Suggested experiment*: Implement a baseline comparison between GROD and OE methods (e.g., Gaussian noise or straightforward OE with diverse datasets). This experiment could involve evaluating performance differences in OoD performance and computational efficiency.

2. Impact of GROD’s Data Generation Methodology:
Does GROD’s use of PCA and LDA projections for generating synthetic OoD data significantly outperform simpler methods?

    *Suggested experiment:* Conduct an ablation study comparing GROD’s synthetic OoD data generation method to simpler techniques like Gaussian noise or uniformly random OoD data. Evaluate performance of using the proposed technique but instead of the proposed data generation use Gaussian noise at the input.

If the authors can provide sufficient evidence for the benefits of the proposed OoD data generation method over using simpler techniques and provide result on comparison with OE I am willing to increase my score.

---

> ### Author Response · Authors · 2024-11-20
>
> Dear Reviewer fp5B,
>
> Thanks for your constructive and valuable comments on our paper! We are encouraged by your recognition of our theoretical contribution and the computational efficiency of GROD. To address your concern, below we show our explanations and supplementary experiments following your suggestion:
>
> **Baseline Comparison with Outlier Exposure (OE):**
> For outlier exposure (OE) methods like [1, 2], our OOD detection is under the assumption that outliers can not be seen in the training phase, which is another dev training with extra data in the generalized OOD detection task and also has its real application scenarios [a]. To make our paper more clearly, we added this sentence into our paper in Section 3: *It is notable that we focus on OOD detection without training with outlier datasets, which is another dev different from OE and also has its real application scenarios [a].*
> We test several OE methods as the table below, but we insist that comparing OOD detection methods with OE is not fair for the comparison of different information from inputs. For convenience and the standard benchmark guarantee, we choose a classical OE method [1] and a prevalent method MIXOE [b] for CV tasks, which are incorporated in the OpenOOD benchmark. NLP OE datasets for CLINC-150 and IMDB have not been well-defined. We will add these results in the main part of our paper if you should. Besides, we will introduce OE methods *in our revised manuscript in Section 2 and Appendix A*.
>
> **OE**:
> ID: CIFAR-10, Auxiliary OOD: Tiny ImageNet-597, ID ACC=95.70%
> | OOD dataset | FPR@95 | AUROC | AUPR_IN | AUPR_OUT |
> | ----------- | ----------- |----------- | ----------- |----------- |
> |CIFAR-100|24.74|94.62|94.75|94.58|
> |Tiny ImageNet|4.97|99.18|99.30|99.08|
> |SVHN|4.39|99.04|97.94|99.59|
>
> ID: CIFAR-100, Auxiliary OOD: Tiny ImageNet-597, ID ACC=74.97%
> | OOD dataset | FPR@95 | AUROC | AUPR_IN | AUPR_OUT |
> | ----------- | ----------- |----------- | ----------- |----------- |
> |CIFAR-10|73.80|73.72|72.86|75.75|
> |Tiny ImageNet|22.02|96.64|97.11|96.46|
> |SVHN|41.66|92.97|81.74|97.37|
>
> ID: ImageNet-200, ID ACC=89.48%
> | OOD dataset | FPR@95 | AUROC | AUPR_IN | AUPR_OUT |
> | ----------- | ----------- |----------- | ----------- |----------- |
> |CIFAR-10|25.33|92.66|93.02|91.74|
> |CIFAR-100|33.08|92.99|93.10|92.68|
> |SVHN|0.69|99.78|99.64|99.87|
>
> **MIXOE**:
> ID: CIFAR-10, Auxiliary OOD: Tin597, ID ACC=96.47%
> | OOD dataset | FPR@95 | AUROC | AUPR_IN | AUPR_OUT |
> | ----------- | ----------- |----------- | ----------- |----------- |
> |CIFAR-100|20.31|95.60|95.73|95.64|
> |Tiny ImageNet|10.66|97.92|98.28|97.67|
> |SVHN|5.94|98.77|97.40|99.51|
>
> ID: CIFAR-100, Auxiliary OOD: Tin597, ID ACC=77.84%
> | OOD dataset | FPR@95 | AUROC | AUPR_IN | AUPR_OUT |
> | ----------- | ----------- |----------- | ----------- |----------- |
> |CIFAR-10|71.07|75.84|74.76|78.55|
> |Tiny ImageNet|49.01|88.61|91.22|86.03|
> |SVHN|49.08|92.14|78.58|97.26|
>
> ID: ImageNet-200, ID ACC=90.49%
> | OOD dataset | FPR@95 | AUROC | AUPR_IN | AUPR_OUT |
> | ----------- | ----------- |----------- | ----------- |----------- |
> |CIFAR-10|25.43|92.46|92.75|91.22|
> |CIFAR-100|33.71|92.60|92.69|92.09|
> |SVHN|1.41|99.63|99.36|99.80|
>
> **Evaluation of Synthetic OoD Data:** Thanks for your suggestion. We think it is meaningful to further explore the generation strategy of GROD, and have complemented experiments in the ablation study in Table 6 in the revised version. For detail, the results are as the following table:
>
> Add to Table 6 in the paper, ID: CIFAR-10
> Gaussian outliers: ID ACC: 96.86%
> | OOD dataset | FPR@95 | AUROC | AUPR_IN | AUPR_OUT |
> | ----------- | ----------- |----------- | ----------- |----------- |
> |CIFAR-100|20.22|96.10|95.95|96.30|
> |Tiny ImageNet|10.92|97.97|98.21|97.79|
> |SVHN|2.29|99.41|98.74|99.75|
>
> Uniformly random outliers: ID ACC: 96.67%
> | OOD dataset | FPR@95 | AUROC | AUPR_IN | AUPR_OUT |
> | ----------- | ----------- |----------- | ----------- |----------- |
> |CIFAR-100|19.39|95.84|95.90|95.92|
> |Tiny ImageNet|10.06|98.03|98.42|97.70|
> |SVHN|4.03|99.22|98.11|99.72|
>
> We add these results in Table 6 of the revised version.
>
> **References:**
>
> [a] Yang J, Zhou K, Li Y, et al. Generalized out-of-distribution detection: A survey[J]. International Journal of Computer Vision, 2024: 1-28.
>
> [b] Zhang J, Inkawhich N, Linderman R, et al. Mixture outlier exposure: Towards out-of-distribution detection in fine-grained environments[C]//Proceedings of the IEEE/CVF Winter Conference on Applications of Computer Vision. 2023: 5531-5540.
>
> We appreciate your willingness to increase the score, and we hope that the clarifications and improvements we have provided align with your expectations. Should you need any further information, please do not hesitate to reach out.
>
> Yours sincerely,
>
> Authors

---

> > ### Comment · Reviewer_fp5B · 2024-11-23
> >
> > Thank you for your response, I have read through the updated paper and the rebuttal. The authors have sufficiently addressed my comments I will increase my score to 6.

---

> > > ### Author Response · Authors · 2024-11-23
> > >
> > > Thank you very much for taking the time to revisit our manuscript and for your in-time comments! We are delighted to hear that the revisions and the rebuttal have addressed your concerns. We truly appreciate your decision to increase your score, and your feedback has been constructive in enhancing the quality of our work.

---

### Official Review · Reviewer_Hak4 · 2024-11-03

**Soundness:** 3
**Presentation:** 3
**Contribution:** 3
**Rating:** 6
**Confidence:** 4

**Summary:**

The paper presents a novel algorithm, GROD, aimed at enhancing OOD detection in transformer networks, which is a timely and innovative addition to current research. By combining PCA and Linear Discriminant Analysis (LDA) projections for OOD data synthesis, it proposes an original approach to address limitations in existing OOD detection methods.

**Strengths:**

- The theoretical foundation provided, especially the PAC learning framework for OOD detection, is a noteworthy contribution that bridges theoretical gaps in understanding transformers learnability for OOD tasks.
- The paper provides thorough experimental validation across multiple OOD detection tasks for both NLP and CV, showing GROD’s adaptability to various data formats, which is admirable.
- Key terms and concepts, such as the PAC learning framework and the GROD algorithm, are introduced clearly, though some technical sections might benefit from additional explanation for accessibility.
- The theoretical insights, especially the derived conditions and error bounds for learnability in transformers, could pave the way for future advancements in OOD detection frameworks for transformers, making it a valuable reference for ongoing research.

**Weaknesses:**

+ The GROD algorithm involves several hyperparameters (e.g., the scaling parameter for Mahalanobis distance, LDA cluster dimensions) that require fine-tuning.
+ While GROD achieves a balance between computational efficiency and performance, its iterative processes, including OOD data synthesis and Mahalanobis distance calculation, may not scale well with significantly larger datasets (e.g., ImageNet) or models. This limitation could restrict its deployment in real-time applications where processing speed is crucial.
+ There are missing citations in the manuscript. For example, the paper introduces generative models, but generative-based methods [1, 2, 3] are missed without corresponding details in the bibliography.

[1]: Kong, Shu, and Deva Ramanan. "Opengan: Open-set recognition via open data generation." In ICCV. 2021.

[2]: Wang, Qizhou, et al. "Out-of-distribution detection with implicit outlier transformation." In ICLR, 2023.

[3]: Zheng, Haotian, et al. "Out-of-distribution Detection Learning with Unreliable Out-of-distribution Sources." In NeurIPS, 2023.

**Questions:**

+ "Learnability" is repeatedly used without a concise definition in layman’s terms, which could be clarified for a broader audience.
+ How do the authors interpret the performance of LDA-based inter-class OOD generation in enriching OOD representation? More specifically, what are the primary limitations observed when using PCA-only projections, and how might these affect model robustness?

---

> ### Author Response · Authors · 2024-11-20
>
> Dear Reviewer Hak4,
>
> We are grateful for your insightful comments and feedback. We appreciate that you agree with our theoretical insights and thorough experimental validation. To address your concerns, we have provided comments as follows.
>
> >The GROD algorithm involves several hyperparameters (e.g., the scaling parameter for Mahalanobis distance, LDA cluster dimensions) that require fine-tuning.
>
> Our method introduces three hyperparameters $\alpha$, $num$, and $\gamma$. $num=1,2$ is empirically an optimal choice, which is consistent with the conclusion in [a] that even adding one or two OOD can raise the OOD detection performance of transformers. The ablation results regarding $\gamma$ in Fig. 3 show that $\gamma \in [0.1,0.3]$ benefits the task performance, which is also in line with the theoretical insights and the classification (learned by $\mathcal{L}_1$) and OOD detection (learned by $\mathcal{L}_2$) goal of the task. Therefore, $num$ and $\gamma$ have their optimal solution, which would be clearly stated in the camera-ready version. As to $\alpha$, if $LDA$ is used, we recommend $\alpha=10^{-3}$, otherwise a larger value should be taken.
> In Subsection 4.3, we have analyzed in detail why these parameters are set like this, and give an explanation from the perspective of OOD detection learning theory.
>
> [a] Fort S, Ren J, Lakshminarayanan B. Exploring the limits of out-of-distribution detection[J]. Advances in Neural Information Processing Systems, 2021, 34: 7068-7081.
>
> >While GROD achieves a balance between computational efficiency and performance, its iterative processes, including OOD data synthesis and Mahalanobis distance calculation, may not scale well with significantly larger datasets (e.g., ImageNet) or models. This limitation could restrict its deployment in real-time applications where processing speed is crucial.
>
> Firstly, we have shown the computational efficiency and performance when using the large-scale dataset ImageNet-200 for comparison (Table 4 and Fig. 2), in which GROD gains superior performance. With the design of $|I|$, we achieve a balance between the complexity of OOD generation and the class number of datasets. Meanwhile, the speed of GROD is only related to the dimensions of features in the last layer of transformers, which is independent of the total scale of models.
>
> >There are missing citations in the manuscript. For example, the paper introduces generative models, but generative-based methods [1, 2, 3] are missed without corresponding details in the bibliography.
>
> Thank you for pointing out the missing citations in the manuscript. GROD is not a classical generation method for it only generates OOD in the embedding feature space, but for the completeness of our work, we will add generative-based methods into related works *in the revised manuscript which can be seen in the updated PDF in Section 2 and Appendix A*.
>
> >"Learnability" is repeatedly used without a concise definition in layman’s terms, which could be clarified for a broader audience.
>
> We give Definition 1. (Strong learnability) in Appendix B, and with your advice, we will add a qualitative description for "Learnability in OOD detection" in Section 3: *"A model is considered learnable for OOD detection if, when trained on sufficient ID data, it is capable of distinguishing OOD samples from ID samples without compromising its classification performance.'"*.
>
> >How do the authors interpret the performance of LDA-based inter-class OOD generation in enriching OOD representation? More specifically, what are the primary limitations observed when using PCA-only projections, and how might these affect model robustness?
>
> PCA-only projections see ID data as one class, so outliers generated by the PCA-only method extract less information compared with LDA-based inter-class OOD. PCA-only is robust, instead, if using LDA when the LDA-based outliers are computed to be none, the generation would be unstable because the number of each class of ID is too little to generate stable rounded outliers.
> Besides, it is sorry that we have report PCA only condition when using CIFAR-100 as ID in Table 3, but $\rm min \{ |\hat{I}|,[\frac{2B}{K\cdot num}]\}$ can be $1$ in some cases, and we updated the results as GROD in the transition mode:
>
> ID ACC=86.10%, and OOD detection metrics (%):
>
> | OOD dataset | FPR@95 | AUROC | AUPR_IN | AUPR_OUT |
> | ----------- | ----------- |----------- | ----------- |----------- |
> |CIFAR10|38.22|90.45|90.17|90.88|
> |Tiny ImageNet|27.98|93.32|95.38|90.52|
> |SVHN|22.12|93.70|88.91|96.59|
>
> It achieves SOTA when CIFAR-100 as ID (Table 3) even only adds one class of inter-class OOD, and we will update the results in the revised version.
>
> We sincerely appreciate your positive feedback and recognition of the novelty of our work in the OOD detection community! If any of our responses fail to address your concerns sufficiently, please inform us, and we will promptly follow up.
>
> Yours sincerely,
>
> Authors

---

> > ### Comment · Reviewer_Hak4 · 2024-11-22
> >
> > I thank the authors for the detailed rebuttal and the additional results. The results are compelling. I will keep my rate.

---

> > > ### Author Response · Authors · 2024-11-23
> > >
> > > We sincerely appreciate your positive feedback and thank you again for your thorough review!

---

### Official Review · Reviewer_PTbR · 2024-11-04

**Soundness:** 3
**Presentation:** 3
**Contribution:** 3
**Rating:** 5
**Confidence:** 4

**Summary:**

The paper introduces a PAC learning framework for OOD detection of transformer networks. And it also propose a novel approach GROD to improve the OOD detection performance,  including a loss function that penalizes the misclassification of OOD data, and a method for generating synthetic outliers. The GROD algorithm is evaluated across various OOD detection tasks in NLP and CV, demonstrating state-of-the-art performance regardless of data format.

**Strengths:**

1. The paper establishes a PAC learning framework for OOD detection applied to transformers, providing necessary and sufficient conditions for learnability and error boundary estimates. The approach of generating synthetic outliers using PCA and LDA projections is innovative and contributes to the robustness of the model.

2. GROD enhances the generalization capabilities of transformers, leading to improved performance on both ID and OOD data across different tasks and data types. The algorithm achieves SOTA results in OOD detection for both NLP and CV tasks, outperforming other prevalent methods.

3. The paper includes extensive experiments and ablation studies that validate the effectiveness of GROD and provide insights into hyperparameter tuning.

**Weaknesses:**

1. The pre-training of GROD is conducted on the ImageNet-1K dataset, whereas OOD detection is evaluated using the CIFAR dataset. Some categories overlap, such as dogs and cats, which seems unreasonable.
2. In line 147, "Feat() represents extracting CLS tokens," which implies that the GROD algorithm utilizes the CLS token for feature extraction. While it is true that many transformer-based models do not necessarily require a CLS token, reducing the generality of the algorithm.
3. How is the scalability of GROD algorithm? If it work well on other transformer-based pretrained backbone?
4. Can the GROD algorithm be adapted to other types of deep learning architectures beyond transformers (e.g. , ResNet)? It seems to be only related to the input feature.

**Questions:**

See above weakness

---

> ### Author Response · Authors · 2024-11-20
>
> Dear Reviewer PTbR,
>
> Thanks for your constructive and valuable comments on our paper, and for recognizing the contributions of learning theory and generalization capabilities of GROD in our paper. Below, we provide a detailed response to your comments.
>
> **Weakness 1:** The pretraining on ImageNet-1K is primarily aimed at providing a good initialization for backbone networks. This pretraining sets up a solid foundation for the model, which can then be fine-tuned for the specific task of OOD detection. The overlap between the classes in the ImageNet-1K dataset (used for pretraining) and the OOD test set (such as CIFAR) is not an issue, because we utilize the well-trained features but do not recognize them as ID and OOD at the same time. All methods we compared use pre-trained parameters so that it is fair for comparison. Results demonstrate that the GROD method, through fine-tuning, effectively tightens the decision boundaries between the ID and OOD samples.
>
> **Weakness 2:** It is a helpful suggestion and we will add a discussion on potential adaptations", which would be added to our paper as follows:
> *As defined in Definition 4. (Transformer hypothesis space for OOD detection), GROD gains theoretical guarantee on transformers with multiple transformer layers and a classifier for OOD detection and classification tasks. So GROD has compatibility for transformers extract features like CLS tokens and inputs into the classifier, which is appliable to almost all transformer models.* The OOD detection task in our paper is based on classification, and to achieve this goal, most transformer-based models could output features for classification so that these features can be used in GROD.
>
> **Weakness 3:** We have conducted experiments with two backbones ViT-B-16 and BERT. To address the concern, we added another backbone network GPT-2 and the results are shown in the following table.
> Part of the methods (Baseline, GEN) is compatible for both models with CLS tokens as features and without CLS tokens, as they only need logits for processing. Others are only compatible with models with CLS tokens as features, since they utilize features and logits together. We test two modes (with/without CLS token) for the former, denoted as Method-C (with CLS) and Method-L (without CLS), respectively.
>
> ID: CLINC150 with Intents, OOD: CLINC150 with Unknown Intents
>
> |Method|ID ACC| FPR@95 | AUROC | AUPR_IN | AUPR_OUT |
> | ----------- | ----------- |----------- | ----------- |----------- |----------- |
> |Baseline-L|97.09|41.76|91.81|97.92|72.86|
> |Baseline-C|97.44|60.36|86.29|96.26|55.34|
> |VIM|97.44|27.53|93.71|98.21|79.25|
> |GEN-L|97.08|33.29|92.46|97.77|76.76|
> |GEN-C|97.44|32.87|93.24|98.11|77.25|
> |ASH|97.44|41.27|92.73|97.80|78.21|
> |NPOS|97.33|66.24|77.01|93.47|43.90|
> |CIDER|97.43|57.27|81.40|95.00|49.16|
> |GROD|97.51|23.80|94.90|98.55|84.75|
>
> ID: IMDB, OOD: Yelp
> |Method|ID ACC| FPR@95 | AUROC | AUPR_IN | AUPR_OUT |
> | ----------- | ----------- |----------- | ----------- |----------- |----------- |
> |Baseline-L|88.56|100.00|59.10|67.81|70.51|
> |Baseline-C|87.93|100.00|58.41|64.50|67.59|
> |VIM|87.93|84.81|58.55|51.60|63.95|
> |GEN-L|88.56|57.80|75.00|73.55|75.43|
> |GEN-C|87.93|76.90|65.84|60.79|69.52
> |ASH|87.93|85.41|60.45|50.97|68.66|
> |NPOS|88.08|96.92|50.23|39.94|60.67|
> |CIDER|87.89|84.46|59.71|52.03|62.99|
> |GROD|88.03|75.17|66.91|61.96|70.95|
>
> **Weakness 4:** We have derived a thorough learning theory to guarantee the OOD detection learnability on transformers, which may not be applicable to MLP-based and CNN-based models, unless the same learning theory is established. So we focus on transformers to validate the efficiency of GROD, which is more theoretically explainable and connects closely to our theoretical results.
>
> We would like to thank you again for this in-depth review. If our clarifications and improvements address your concerns, we would greatly appreciate it if you could reconsider your evaluation. Should you have any additional questions or suggestions, we would be happy to provide further details.
>
> Yours sincerely,
>
> Authors

---

> ### Comment · Reviewer_PTbR · 2024-11-22
>
> Thanks for the rebuttal, the rebuttal does not effectively address my concerns. The use of a supervised pre-trained model, particularly when there is an overlap between the pre-trained data and the OOD test data, raises significant concerns regarding the validity of the study's findings. The network's ability to extract representations from OOD data that it has been previously exposed to during pre-training undermines the challenge of distinguishing between in-distribution and OOD samples. This overlap effectively simplifies the task and may lead to the reporting of artificially inflated performance metrics. While some recent studies, such as NPOS, have employed pre-trained models, they have utilized unsupervised pre-trained models like CLIP and have also reported results from models trained from scratch. This approach provides a more robust evaluation of the model's ability to generalize to OOD data. It is suspected that the improvements in OOD detection reported for GROD may be attributable to the specific characteristics of the supervised pre-trained data rather than the model's inherent ability to detect OOD samples.

---

> > ### Author Response · Authors · 2024-11-22
> >
> > Thank you for your response and valuable insights! For the concerns mentioned above, we have carefully considered your comments and give further explanation as follows:
> >
> > **Fair Comparison Across CV Tasks:** For computer vision (CV) tasks （Table 2/3/4）, GROD and all other baseline methods utilize the same pre-trained models, allowing for a fair comparison of GROD's effectiveness. This demonstrates that the improved results in OOD detection are likely due to the novel GROD algorithm rather than the characteristics of the supervised pre-trained data alone.
> >
> > **Use of Unsupervised Pre-trained Models in NLP:** For natural language processing (NLP) tasks, both BERT and GPT-2 (Table 5 in the revised version), which were utilized in experiments, are unsupervised pre-trained models. These two pre-trained models fully meet the two conditions you mentioned: unsupervised training and the absence of features with similar semantics during the training phase. The strong performance of GROD even with unsupervised models reinforces the argument that GROD's methodology is effective across different pre-training paradigms.
> >
> > We will add the clarification in the main part of our paper and thanks again for your in-time response. We are grateful for the opportunity to enhance our manuscript with your insightful feedback and look forward to any further suggestions you might have.

---

> ### Comment · Reviewer_PTbR · 2024-11-26
>
> Thanks for your reply. Further clarification on fair comparison in CV tasks is not solid, as there is insufficient motivation to use supervised pretrained models. Therefore, I decide to keep my rate.

---

> > ### Author Response · Authors · 2024-11-26
> >
> > Thank you very much for dedicating your time and providing such detailed feedback on our manuscript. We are grateful for your observations and are eager to offer further details to reinforce the soundness of our methods and findings.
> >
> > We have conducted fair comparisons across all methods on the ViT backbone network. Starting model pre-training from scratch is not essential for our purposes because the objective of fine-tuning methods for OOD detection is specifically to enhance the model's capability to detect OOD data through the fine-tuning stage. This approach maximally leverages information from both the pre-training and fine-tuning phases to achieve optimal performance for the task.
> >
> > Moreover, other non-supervised pre-trained backbone networks could potentially have been exposed to OOD data semantics during pre-training as well, which renders the criticism regarding our use of pre-trained models less compelling. Furthermore, in the realm of NLP tasks, the BERT and GPT-2 models we use are not supervised pre-trained models. Our methodology has demonstrated superior performance across various pre-training schemes.
> >
> > We value your engagement with our research and sincerely hope that the additional explanations we have provided meet your expectations and address the concerns raised.

---

### Official Review · Reviewer_zQzi · 2024-11-05

**Soundness:** 1
**Presentation:** 2
**Contribution:** 1
**Rating:** 3
**Confidence:** 5

**Summary:**

The paper introduces GROD, an approach to enhance transformers' OOD detection performance by incorporating synthesized OOD data. GROD leverages a Probably Approximately Correct (PAC) theory framework, proposing a learnable criterion for transformers that improves their ability to recognize OOD instances. By integrating OOD misclassification penalties into the loss function and generating synthetic outliers through PCA and LDA projections, GROD establishes a more robust boundary between in-distribution and OOD data.

**Strengths:**

* The experimental results cover multiple modalities, including both text and image data.

* The study provides both theoretical analysis and experimental validation to support the proposed pipeline.

**Weaknesses:**

* (A) The authors should explore additional architectures, including MLP-based and CNN-based models, and explain how their method would apply to these. While the study clarifies that it focuses on transformers, it should explicitly address the pipeline’s compatibility with different architectures and provide a discussion on potential adaptations.

* (B) The study includes only a limited number of transformer-based architectures, specifically ViT-B16 and BERT.

* (C) The datasets used in this study are relatively small (e.g., CIFAR vs. SVHN). Larger and higher-resolution benchmarks (e.g., ImageNet, Texture) should be considered to show the contribution.

* (D) Several studies have incorporated synthetic sampling strategies for OOD detection [1,2,3,4,5,6], but there is a lack of comparison with these methods.

* (E) The primary idea of the pipeline shows similarities with [7].


* (F) Using a large pretrained model, such as ViT-B16, for the relatively small Tiny ImageNet dataset raises the issue that the pipeline may rely on extra information seen by the backbone during pretraining rather than on the proposed pipeline itself.


[1] Lee et al., "Training Confidence-Calibrated Classifiers for Detecting Out-of-Distribution Samples," ICLR 2018.

[2] Kirchheim et al., "On Outlier Exposure with Generative Models," NeurIPS ML Safety Workshop, 2022.

[3] Du et al., "VOS: Learning What You Don’t Know by Virtual Outlier Synthesis," ICLR 2022.

[4] Tao et al., "Non-Parametric Outlier Synthesis," ICLR 2023.

[5] Du et al., "Dream the Impossible: Outlier Imagination with Diffusion Models," NeurIPS 2023.

[6] Chen et al., "ATOM: Robustifying Out-of-Distribution Detection Using Outlier Mining."

[7] "Fake It Till You Make It: Towards Accurate Near-Distribution Novelty Detection."

[8] Deep Hybrid Models for Out-of-Distribution Detection

**Questions:**

* The main motivation for this study is unclear, given that existing methods already achieve strong results on the OOD detection benchmarks considered. For example, [8] achieves competitive performance on CIFAR10 vs. CIFAR100 without using additional information (i.e., without pre-trained models). Thus, the necessity of the proposed pipeline remains uncertain.

* The authors should clearly outline their contributions over similar works [1-7], detailing the limitations of previous approaches and supporting these claims with comprehensive experiments.

* The authors are encouraged to explore a broader range of architectures and models rather than focusing solely on ViT-B16.


* Tiny ImageNet has overlap with both CIFAR-10 and CIFAR-100. How do the authors justify considering these datasets as ID and OOD?

---

> ### Author Response · Authors · 2024-11-20
>
> Dear Reviewer zQzi,
>
> Thanks for your constructive and valuable comments on our paper. Below, we provide a detailed response to your questions and comments. If any of our responses fail to address your concerns sufficiently, please inform us, and we will promptly follow up.
>
> **Weakness (A):** We have derived a thorough learning theory to guarantee the OOD detection learnability on transformers, which may not be applicable to MLP-based and CNN-based models unless the same learning theory is established. So we focus on transformers to validate the efficiency of GROD, which is more theoretically explainable and connects closely to our theoretical results. It is a helpful suggestion to "provide a discussion on potential adaptations", which would be added to our paper as follows:
> *As defined in Definition 4. (Transformer hypothesis space for OOD detection), GROD gains theoretical guarantee on transformers with multiple transformer layers and a classifier for OOD detection and classification tasks. So GROD has compatibility for transformers extract features like CLS tokens and inputs into the classifier, which is appliable to almost all transformer models.*
>
> **Weakness (B):** For experimental validation, we consider multiple factors including dataset types, dataset size, outlier types, and different PCA and LDA generation conditions, and finally make a decision on our benchmarks. However, as discussed in Weaknesses (A), to prove the versatility of GROD concerning transformers, we add a group of experiments using decoder-only transformer GPT-2 as the backbone network on NLP task to verify our method further. Part of the methods (Baseline, GEN) is compatible for both models with CLS tokens as features and without CLS tokens, as they only need logits for processing. Others are only compatible with models with CLS tokens as features, since they utilize features and logits together. We test two modes (with/without CLS token) for the former, denoted as Method-C (with CLS) and Method-L (without CLS), respectively. The comparison results are reported as follows:
>
> ID: CLINC150 with Intents, OOD: CLINC150 with Unknown Intents
>
> |Method|ID ACC| FPR@95 | AUROC | AUPR_IN | AUPR_OUT |
> | ----------- | ----------- |----------- | ----------- |----------- |----------- |
> |Baseline-L|97.09|41.76|91.81|97.92|72.86|
> |Baseline-C|97.44|60.36|86.29|96.26|55.34|
> |VIM|97.44|27.53|93.71|98.21|79.25|
> |GEN-L|97.08|33.29|92.46|97.77|76.76|
> |GEN-C|97.44|32.87|93.24|98.11|77.25|
> |ASH|97.44|41.27|92.73|97.80|78.21|
> |NPOS|97.33|66.24|77.01|93.47|43.90|
> |CIDER|97.43|57.27|81.40|95.00|49.16|
> |GROD|97.51|23.80|94.90|98.55|84.75|
>
> ID: IMDB, OOD: Yelp
> |Method|ID ACC| FPR@95 | AUROC | AUPR_IN | AUPR_OUT |
> | ----------- | ----------- |----------- | ----------- |----------- |----------- |
> |Baseline-L|88.56|100.00|59.10|67.81|70.51|
> |Baseline-C|87.93|100.00|58.41|64.50|67.59|
> |VIM|87.93|84.81|58.55|51.60|63.95|
> |GEN-L|88.56|57.80|75.00|73.55|75.43|
> |GEN-C|87.93|76.90|65.84|60.79|69.52
> |ASH|87.93|85.41|60.45|50.97|68.66|
> |NPOS|88.08|96.92|50.23|39.94|60.67|
> |CIDER|87.89|84.46|59.71|52.03|62.99|
> |GROD|88.03|75.17|66.91|61.96|70.95|
>
> **Weakness (C):** Firstly, our experiment includes a largescale dataset ImageNet-200 as ID as shown in Table 4. In fact, we have conducted experiments of Texture, as OOD, and we have not reported these results just because far-OOD is simple to detect for GROD and other competitive models, and shows similar performance for method comparison, so we only remain SVHN as a representation of far-OOD. Here we offer the results on other far-OOD datasets compared with SVHN in Table 2 and Table 3, respectively:
>
> | OOD dataset | FPR@95 | AUROC | AUPR_IN | AUPR_OUT |
> | ----------- | ----------- |----------- | ----------- |----------- |
> | SVHN |0.09| 99.98|99.97|99.99|
> |Texture | 0.03|99.99|100.00|99.99|
> |Places-365|0.11|99.98|99.95|99.99|
>
> | OOD dataset | FPR@95 | AUROC | AUPR_IN | AUPR_OUT |
> | ----------- | ----------- |----------- | ----------- |----------- |
> | SVHN |23.38| 94.59|87.88|98.63|
> |Texture | 12.30|97.65|98.51|96.47|
> |Places-365|27.22|94.34|85.13|98.30|
>
> We are happy to add these experimental results to the appendix, should you recommend this approach.

---

> ### Author Response · Authors · 2024-11-20
>
> **Weakness (D):** We have read all the papers referenced in your comments. Although they are really excellent methods for OOD detection tasks, several reasons make us not choose them for comparison experiments. For [1], different from your description, we find it is not an outlier synthesis method. For outlier exposure (OE) methods like [2, 6], they use outliers from other datasets but do not generate them. And our OOD detection is under the assumption that outliers can not be seen in the training phase, which is another dev training with extra data in the generalized OOD detection tasks and also has its real application scenarios [a]. To make our paper more clear, we added this sentence into our paper in Section 3: *It is notable that we focus on OOD detection without training with outlier datasets, which is another dev different from OE and also has its real application scenarios [a].*
> [3] is already verified to be inferior compared with [4] according to [4], and [4] is already tested in Table 2/3/4 as the method "NPOS".
> [5] uses ResNet as the backbone and seems compatible with transformers. However, it is highlighted as a way to generate high-resolution outliers in the pixel space, which seems designed especially for CV tasks.
>
> Above all, we test several OE methods as the table below, but comparing OOD detection methods with OE is not fair for comparison of different information from inputs. For convenience and the standard benchmark guarantee, we choose a classical OE method [b] and a prevalent method MIXOE [c] for CV tasks as shown in Table 2/3/4, which are incorporated in the OpenOOD benchmark. NLP OE datasets for CLINC-150 and IMDB have not been well-defined. We will add these results in the main part of our paper if you should. Besides, we will introduce, discuss and distinguish the related concepts of [1-8] *in our revised manuscript in Section 2 and Appendix A*.
>
> **OE**:
> ID: CIFAR-10, Auxiliary OOD: Tin597, ID ACC=95.70%
> | OOD dataset | FPR@95 | AUROC | AUPR_IN | AUPR_OUT |
> | ----------- | ----------- |----------- | ----------- |----------- |
> |CIFAR-100|24.74|94.62|94.75|94.58|
> |Tiny ImageNet|4.97|99.18|99.30|99.08|
> |SVHN|4.39|99.04|97.94|99.59|
>
> ID: CIFAR-100, Auxiliary OOD: Tin597, ID ACC=74.97%
> | OOD dataset | FPR@95 | AUROC | AUPR_IN | AUPR_OUT |
> | ----------- | ----------- |----------- | ----------- |----------- |
> |CIFAR-10|73.80|73.72|72.86|75.75|
> |Tiny ImageNet|22.02|96.64|97.11|96.46|
> |SVHN|41.66|92.97|81.74|97.37|
>
> ID: ImageNet-200, ID ACC=89.48%
> | OOD dataset | FPR@95 | AUROC | AUPR_IN | AUPR_OUT |
> | ----------- | ----------- |----------- | ----------- |----------- |
> |CIFAR-10|25.33|92.66|93.02|91.74|
> |CIFAR-100|33.08|92.99|93.10|92.68|
> |SVHN|0.69|99.78|99.64|99.87|
>
> **MIXOE**:
> ID: CIFAR-10, Auxiliary OOD: Tin597, ID ACC=96.47%
> | OOD dataset | FPR@95 | AUROC | AUPR_IN | AUPR_OUT |
> | ----------- | ----------- |----------- | ----------- |----------- |
> |CIFAR-100|20.31|95.60|95.73|95.64|
> |Tiny ImageNet|10.66|97.92|98.28|97.67|
> |SVHN|5.94|98.77|97.40|99.51|
>
> ID: CIFAR-100, Auxiliary OOD: Tin597, ID ACC=77.84%
> | OOD dataset | FPR@95 | AUROC | AUPR_IN | AUPR_OUT |
> | ----------- | ----------- |----------- | ----------- |----------- |
> |CIFAR-10|71.07|75.84|74.76|78.55|
> |Tiny ImageNet|49.01|88.61|91.22|86.03|
> |SVHN|49.08|92.14|78.58|97.26|
>
> ID: ImageNet-200, ID ACC=90.49%
> | OOD dataset | FPR@95 | AUROC | AUPR_IN | AUPR_OUT |
> | ----------- | ----------- |----------- | ----------- |----------- |
> |CIFAR-10|25.43|92.46|92.75|91.22|
> |CIFAR-100|33.71|92.60|92.69|92.09|
> |SVHN|1.41|99.63|99.36|99.80|
>
> **Weakness (E)**: Our paper is different from [7] for the following reasons. Firstly, [7] mainly solves the novelty detection (ND) task, not for OOD detection.
> **Different goals:** While the detected novel samples are usually prepared for future constructive procedures, such as more specialized analysis, or incremental learning of the model itself in ND, OOD detection aims to detect outliers for the safety and generalization capability of models. [7] emphasize the quality of outliers generated by diffusion models, but GROD only needs to synthesize them in a high-dimensional embedding space of transformers to strengthen decision boundaries, which is lightweight and suitable for OOD detection.
> **Different pipelines:** ND is only to distinguish novel samples from ID, while OOD detection serves classification performance.
> **Different novelty:** [7] verify that diffusion models can generate novel outlier samples and benefit ND, while GROD is for OOD detection and is inspired and guaranteed by the proposed OOD detection learning theory.

---

> ### Author Response · Authors · 2024-11-20
>
> **Weakness (F):** GROD is a fine-tuning and post-processing method that leverages information extracted by pretraining, to get an initial performance for classification. By fine-tuning with GROD, we further strengthen classification and OOD detection capability, which has been verified in the ablation study in Table 6.
>
> **References:**
>
> [a] Yang J, Zhou K, Li Y, et al. Generalized out-of-distribution detection: A survey[J]. International Journal of Computer Vision, 2024: 1-28.
>
> [b] Hendrycks D, Mazeika M, Dietterich T. Deep anomaly detection with outlier exposure[J]. arXiv preprint arXiv:1812.04606, 2018.
>
> [c] Zhang J, Inkawhich N, Linderman R, et al. Mixture outlier exposure: Towards out-of-distribution detection in fine-grained environments[C]//Proceedings of the IEEE/CVF Winter Conference on Applications of Computer Vision. 2023: 5531-5540.
>
> **Question 1:** Our paper first proposes the learning theory and a theoretical guaranteed method for transformer-based backbones. [8] uses real auxiliary outlier data for training, which is another topic in OOD detection as explained in weakness (D). Besides, [8] has not achieved competitive performance on CIFAR10 vs. CIFAR100, instead, it uses CIFAR-10 or CIFAR-100 as ID and far-ood datasets like SVHN as OOD, so results are not comparable. Validated mainly using DensNet, [8] has no evidence to show if it needs pre-trained parameters when using transformers.
>
> **Question 2:** Please see comments for weaknesses (D) and (E).
>
> **Question 3:** Please see comments for weakness (B).
>
> **Question 4:** We follow the supported benchmarks provided by OpenOOD, which defines Tiny ImageNet as near-OOD for CIFAR-10 and CIFAR-100.
>
> Thank you again for your valuable suggestions and constructive feedback! If we have adequately addressed your concerns, we kindly ask for your consideration in enhancing the score you've allotted to our submission. Otherwise, we would be glad to answer any additional questions you may have.
>
> Yours sincerely,
>
> Authors

---

> > ### Author Response · Authors · 2024-11-24
> > **Request to review the rebuttal**
> >
> > Dear Reviewer zQzi,
> >
> > Thank you for your thoughtful review of our manuscript. We have responded to your feedback and submitted a revised version of the paper accordingly. As the rebuttal phase is drawing to a close, we would greatly appreciate it if you could review our updates and provide any further comments or concerns. Thank you once again for your valuable feedback!
> >
> > Yours sincerely,
> >
> > Authors

---

> > > ### Author Response · Authors · 2024-11-26
> > > **Request to review the rebuttal**
> > >
> > > Dear Reviewer zQzi,
> > >
> > > Thank you for your thoughtful review and the valuable insights you provided on our manuscript. We have carefully considered your feedback and have submitted a revised version of the paper. Your expertise has been crucial to our revisions, and we are keen to know your thoughts on the changes we've implemented.
> > >
> > > We understand that reviewing requires significant effort, and we truly appreciate the time you dedicate to this process. We hope you are able to review our rebuttal and share any additional comments, which will be essential in further enhancing the quality and clarity of our manuscript. If our clarifications and improvements address your concerns, we would greatly appreciate it if you could reconsider your evaluation.
> > >
> > > Yours sincerely,
> > >
> > > Authors

---

> ### Comment · Reviewer_zQzi · 2024-12-02
>
> Thank you for the detailed and extensive responses; I really appreciate them. For each considered benchmark, there exist methods that have significant performance results. Although I like that the paper has theoretical analysis, I believe that OOD detection is a practical task, and experimental soundness is more important. Since I found the results gap compared to existing methods to be minor, and the pipeline novelty incremental, I will maintain my score.

---

> > ### Author Response · Authors · 2024-12-02
> >
> > Thank you for your thoughtful and constructive feedback. We greatly appreciate your recognition of the theoretical analysis in the paper. We understand your concern that OOD detection is a practical task and that experimental soundness is paramount. However, we would like to address a few key points:
> >
> > >For each considered benchmark, there exist methods that have significant performance results.
> >
> > - As shown in Tables 2/3/4/5, while other methods achieve significant performance on specific benchmarks, no single method demonstrates consistently competitive performance across all tasks. In contrast, GROD stably excels across both NLP and CV tasks, representing a significant contribution.
> > - As summarized in the tables, GROD outperforms competing methods in five key metrics across all considered benchmarks, highlighting its stability and robustness:
> >
> > **On average (``Average" in Table 2/3/4):**
> > |Benchmark| Table 2 | Table 3 | Table 4 (PCA only) |
> > | ----------- | ----------- |----------- | ----------- |
> > |ID ACC (%)|97.31 (+0.33)|86.21 (+1.34)|90.71 (+0.22)|
> > |FPR@95 $\downarrow$ (%)|0.12 (-9.29)|29.44 (-6.53)|18.26 (+0.13)|
> > | AUROC (%)|99.98 (+2.20)|92.49 (+0.92)|95.24 (+0.06)|
> > | AUPR_IN (%)|99.97 (+2.27)|91.49 (+2.16)|95.54 (+0.25)|
> > | AUPR_OUT (%)|99.97 (+1.98)|92.66 (+0.60)|95.23 (+0.39)|
> >
> > In this table, we compare GROD’s results with the best-performing methods (excluding GROD) for each benchmark and evaluation metric. The value in parentheses represents the difference between the two results. When compared with the best performance across all outlier detection methods, GROD maintains a stable advantage. This demonstrates that GROD is not just competitive with individual methods, but outperforms the best of all methods across multiple tasks. In practical applications, this stability is critical, as practitioners prefer methods that reliably perform well across a range of datasets without the need for extensive trial and error on multiple OOD detection methods.
> >
> >
> > **Table 5 with two benchmarks and two backbones:** While GEN-L outperforms GROD on the IMDB dataset using GPT-2 without a CLS token, it is unstable and not competitive on other tasks. In all other cases, GROD consistently delivers SOTA performance.
> >
> > - Our method excels not only in **performance** across a wide range of benchmarks but also in **processing speed** and **versatility** across multiple modalities. GROD delivers competitive results while maintaining high computational efficiency and broad applicability, making it a strong choice for real-world deployments where both performance and practical efficiency are critical.
> >
> > >The pipeline novelty incremental
> >
> > We agree that some aspects of our approach may appear incremental, but the key novelty lies in integrating the PAC learning framework with practical OOD detection methods. Specifically, we have innovatively designed a theoretically guaranteed loss function and a synthetic outlier generation method, and simultaneously provided reliable explanations for all outlier synthesis strategies.
> >
> > While OOD detection is inherently a practical task, its goal of improving model generalization and safety urgently requires theoretical guarantees and interpretability, which our approach provides.
> >
> >  We hope that our response clarifies the importance of both the practical and theoretical aspects of our approach. Thank you again for your comments and the opportunity to address your concerns.

---

### Author Response · Authors · 2024-12-02
**General Response by Authors**

We sincerely thank all the reviewers for their time, effort, and insightful feedback on our paper. We are pleased that the reviewers broadly acknowledged the contributions of our work. Below, we summarize the key points raised by the reviewers and how we have addressed them in the revised manuscript:

- **Novelty** [PTbR, Hak4, fp5B]: We appreciate the recognition of our contribution in establishing the PAC learning framework for OOD detection. This theoretical foundation is a significant advancement, particularly in bridging gaps in the understanding of transformers' learnability for OOD tasks. The use of synthetic outliers generated through PCA and LDA projections has also been highlighted as an innovative approach. We believe this framework is a valuable addition to the literature on OOD detection.

- **Theoretical insights** [PTbR, zQzi, Hak4, fp5B, PBKN]: We are grateful for the recognition of the PAC learning framework as a solid theoretical contribution to OOD detection in transformers. This framework provides essential conditions for learnability and error boundary estimates, filling a crucial gap in the current understanding of transformer-based models for OOD tasks. We are confident that these insights will offer guidance and pave the way for future advancements in OOD detection frameworks.

- **Experiments and performance of GROD** [zQzi, PTbR, Hak4, fp5B, PBKN]: We appreciate the positive feedback on the experimental results demonstrating GROD's adaptability across different modalities. The comprehensive experiments and ablation studies validate the effectiveness of GROD, highlighting its ability to enhance the generalization capabilities of transformers. GROD achieved SOTA results in OOD detection for both NLP and CV tasks. Additionally, the computational efficiency of GROD has been well-received.

- **Clarity** [Hak4]: We are glad that the clarity of key concepts, such as the PAC learning framework and the GROD algorithm, has been acknowledged


We also thank all reviewers' insightful and constructive comments, which have been invaluable in further improving our paper. Below, we have incorporated several additional experimental results and made revisions based on the reviewers' constructive suggestions:

- **Various backbone models** [zQzi, PTbR]: We have added a decoder-only backbone network, **GPT-2**, for NLP tasks to validate the performance of models without CLS tokens. The results are presented in **Table 5** of the revised manuscript.

- **Outlier exposure methods for comparison** [zQzi, fp5B]: In response to the reviewers' suggestions, we included two **outlier exposure methods**, OE and MIXOE, for comparison. These results are shown in **Tables 2/3/4, and Fig. 2**. For NLP tasks in **Table 5**, the auxiliary outlier benchmark has not been standardized, which is unsuitable for implementing outlier exposure methods.

- **Ablation study of different outlier generation methods** [fp5B, PBKN]: We conducted an ablation study to evaluate different outlier generation methods. In this study, we replaced our original outlier generation method in GROD with Gaussian distribution, uniform random distribution, and random dimension selection for projections. The results, demonstrating the efficiency of our method, are included in **Table 6**.

- **Other supplements**: We updated the results for using CIFAR-100 as the ID dataset in **Table 3**. In certain cases, $\rm min \{ |\hat{I}|,[\frac{2B}{K\cdot num}]\}$ can be equal to 1, and we have accounted for this in the revised manuscript. We have also reported additional results for detecting far-OOD datasets in the comments, as requested by reviewers zQzi and PBKN.

In conclusion, we are grateful for the constructive feedback provided by the reviewers, which has been instrumental in strengthening our paper. We believe that GROD offers a significant contribution to the field of OOD detection, both in terms of its theoretical insights and practical performance. We hope that the revised manuscript meets the reviewers' expectations and look forward to the opportunity for our work to contribute meaningfully to the ongoing research in this area. Thank you once again for all your time and thoughtful consideration.

---

### Meta-Review · Area_Chair_Pgac · 2024-12-18

**Metareview:**

The paper introduces GROD, an approach to enhance transformers' OOD detection performance by incorporating synthesized OOD data. GROD leverages a Probably Approximately Correct (PAC) theory framework, proposing a learnable criterion for transformers that improves their ability to recognize OOD instances. By integrating OOD misclassification penalties into the loss function and generating synthetic outliers through PCA and LDA projections, GROD establishes a more robust boundary between in-distribution and OOD data. For the strengths, the paper establishes a PAC learning framework for OOD detection applied to transformers, providing necessary and sufficient conditions for learnability and error boundary estimates. The approach of generating synthetic outliers using PCA and LDA projections is innovative and contributes to the robustness of the model. GROD enhances the generalization capabilities of transformers, leading to improved performance on both ID and OOD data across different tasks and data types. The algorithm achieves SOTA results in OOD detection for both NLP and CV tasks, outperforming other prevalent methods. The paper includes extensive experiments and ablation studies that validate the effectiveness of GROD and provide insights into hyperparameter tuning. However, there are several points to be further improved. For example, while the authors propose a synthetic OOD generation approach, they do not include a comparison to Outlier Exposure (OE) methods. A comparison, especially with traditional OE using Gaussian noise, would be valuable in demonstrating GROD’s necessity and superiority. OE proposed using additional OOD data which is used to train/finetune a model to better OOD detection performance, a similar to the idea presented in this paper. While GROD’s OOD data generation is sophisticated, it is unclear if the benefits of PCA and LDA projections over simpler alternatives like Gaussian noise have been adequately evaluated. Including a comparison experiment would strengthen claims about the effectiveness of GROD’s approach in OOD data synthesis. Moreover, the GROD algorithm involves several hyperparameters (e.g., the scaling parameter for Mahalanobis distance, LDA cluster dimensions) that require fine-tuning. While GROD achieves a balance between computational efficiency and performance, its iterative processes, including OOD data synthesis and Mahalanobis distance calculation, may not scale well with significantly larger datasets (e.g., ImageNet) or models. This limitation could restrict its deployment in real-time applications where processing speed is crucial. Therefore, this paper cannot be accepted at ICLR this time, but the enhanced version is highly encouraged to submit other top-tier venues.

**Additional Comments On Reviewer Discussion:**

Reviewers keep the score after the rebuttal.

---

### Decision · Program_Chairs · 2025-01-22

Reject